# Latent Variable Representation for Reinforcement Learning

**Tongzheng Ren**[1,2,*]    **Chenjun Xiao**[3,*]    **Tianjun Zhang**[1,4]    **Na Li**[1,5]    **Zhaoran Wang**[6]
**Sujay Sanghavi**[2]    **Dale Schuurmans**[1,3]    **Bo Dai**[1,7]
[1]Google Research, Brain Team    [2]UT Austin    [3]University of Alberta    [4]UC Berkeley
[5]Harvard University    [6]Northwestern University    [7]Georgia Tech
tongzheng@utexas.edu, chenjun@ualberta.ca, bodai@google.com

## Abstract

Deep latent variable models have achieved significant empirical successes in model-based reinforcement learning (RL) due to their expressiveness in modeling complex transition dynamics. On the other hand, it remains unclear theoretically and empirically how latent variable models may facilitate learning, planning, and exploration to improve the sample efficiency of RL. In this paper, we provide a representation view of the latent variable models for state-action value functions, which allows both tractable variational learning algorithm and effective implementation of the optimism/pessimism principle in the face of uncertainty for exploration. In particular, we propose a computationally efficient planning algorithm with UCB exploration by incorporating kernel embeddings of latent variable models. Theoretically, we establish the sample complexity of the proposed approach in the online and offline settings. Empirically, we demonstrate superior performance over current state-of-the-art algorithms across various benchmarks.

## 1 Introduction

Reinforcement learning (RL) seeks an optimal policy that maximizes the expected accumulated rewards by interacting with an unknown environment sequentially. Most research in RL is based on the framework of Markov decision processes (MDPs) (Puterman, 2014). For MDPs with finite states and actions, there is already a clear understanding with sample and computationally efficient algorithms (Auer et al., 2008; Dann & Brunskill, 2015; Osband & Van Roy, 2014; Azar et al., 2017; Jin et al., 2018). However, the cost of these RL algorithms quickly becomes unacceptable for large or infinite state problems. Therefore, function approximation or parameterization is a major tool to tackle the curse of dimensionality. Based on the parametrized component to be learned, RL algorithms can roughly be classified into two categories: model-free and model-based RL, where the algorithms in the former class directly learn a value function or policy to maximize the cumulative rewards, while algorithms in the latter class learn a model to mimic the environment and the optimal policy is obtained by planning with the learned simulator.

Model-free RL algorithms exploit an end-to-end learning paradigm for policy and value function training, and have achieved empirical success in robotics (Peng et al., 2018), video-games (Mnih et al., 2013), and dialogue systems (Jiang et al., 2021), to name a few, thanks to flexible deep neural network parameterizations. The flexibility of such parameterizations, however, also comes with a cost in optimization and exploration. Specifically, it is well-known that temporal-difference methods become unstable or even divergent with general nonlinear function approximation (Boyan & Moore, 1994; Tsitsiklis & Van Roy, 1996). Uncertainty quantization for general nonlinear function approximators is also underdeveloped. Although there are several theoretically interesting model-free exploration algorithms with general nonlinear function approximators (Wang et al., 2020; Kong et al., 2021; Jiang et al., 2017), a computationally-friend exploration method for model-free RL is still missing.

Model-based RL algorithms, on the other hand, exploit more information from the environment during learning, and are therefore considered to be more promising in terms of sample efficiency (Wang et al., 2019). Equipped with powerful deep models, model-based RL can successfully reduce

---

*Equal Contribution. Project Website: https://rlrep.github.io/lvrep/

approximation error, and have demonstrated strong performance in practice (Hafner et al., 2019a;b; Wu et al., 2022), following with some theoretical justifications (Osband & Van Roy, 2014; Foster et al., 2021). However, the reduction of approximation error brings new challenges in planning and exploration, which have not been treated seriously from the empirical and theoretical aspects. Specifically, with general nonlinear models, the planning problem itself is already no longer tractable, and the problem becomes more difficult with an exploration mechanism introduced. While theoretical analysis typically assumes a planning oracle providing an optimal policy, some approximations are necessary in practice, including dyna-style planning (Chua et al., 2018; Luo et al., 2018), random shooting (Kurutach et al., 2018; Hafner et al., 2019a), and policy search with backpropagation through time (Deisenroth & Rasmussen, 2011; Heess et al., 2015). These may lead to sub-optimal policies, even with perfect models, wasting potential modeling power.

In sum, for both model-free and model-based algorithms, there has been insufficient work considering both statistical and computation tractability and efficiency in terms of learning, planning and exploration in a unified and coherent perspective for algorithm design. This raises the question:

*Is there a way to design a **provable** and **practical** algorithm to remedy both the statistical and computational difficulties of RL?*

Here, by "provable" we mean the statistical complexity of the algorithm can be rigorously characterized without explicit dependence on the number of states but instead the fundamental complexity of the parameterized representation space; while by "practical" we mean the learning, planning and exploration components in the algorithm are computationally tractable and can be implemented in real-world scenarios.

This work provides an affirmative answer to the question above by establishing the representation view of latent variable dynamics models through a connection to linear MDPs. Such a connection immediately provides a computationally tractable approach to planning and exploration in the *linear space* constructed by the flexible deep latent variable model. Such a latent variable model view also provides a *variational* learning method that remedies the intractbility of MLE for general linear MDPs (Agarwal et al., 2020; Uehara et al., 2022). Our main contributions consist of the following:

- We establish the representation view of latent variable dynamics models in RL, which naturally induces *Latent Variable Representation (LV-Rep)* for linearly representing the state-action value function, and paves the way for a practical variational method for representation learning (Section 3);
- We provide computation efficient algorithms to implement the principle of optimistm and pessimism in the face of uncertainty with the learned LV-Rep for online and offline RL (Section 3.1);
- We theoretically analyze the sample complexity of LV-Rep in both online and offline settings, which reveals the essential complexity beyond the cardinality of the latent variable (Section 4);
- We empirically demonstrate LV-Rep outperforms the state-of-the-art model-based and model-free RL algorithms on several RL benchmarks (Section 6)

## 2 PRELIMINARIES

In this section, we provide brief introduction to MDPs and linear MDP, which play important roles in the algorithm design and theoretical analysis. We also provide the required background knowledge on functional analysis in Appendix D.

### 2.1 MARKOV DECISION PROCESSES

We consider the infinite horizon discounted Markov decision process (MDP) specified by the tuple $\mathcal{M} = \langle \mathcal{S}, \mathcal{A}, T^*, r, \gamma, d_0 \rangle$, where $\mathcal{S}$ is the state space, $\mathcal{A}$ is a discrete action space, $T^* : \mathcal{S} \times \mathcal{A} \to \Delta(\mathcal{S})$ is the transition, $r : \mathcal{S} \times \mathcal{A} \to [0, 1]$ is the reward, $\gamma \in (0, 1)$ is the discount factor and $d_0 \in \Delta(\mathcal{S})$ is the initial state distribution. Following the standard convention (e.g. Jin et al., 2020), we assume $r(s, a)$ and $d_0$ are known to the agent. We aim to find the policy $\pi : \mathcal{S} \to \Delta(\mathcal{A})$, that maximizes the following discounted cumulative reward:

$$V_{T^*, r}^\pi := \mathbb{E}_{T^*, \pi} \left[ \sum_{i=0}^\infty \gamma^i r(s_i, a_i) \Big| s_0 \sim d_0 \right].$$

We define the state value function $V : \mathcal{S} \rightarrow \left[0, \frac{1}{1-\gamma}\right]$ and state-action value function $Q : \mathcal{S} \times \mathcal{A} \rightarrow \left[0, \frac{1}{1-\gamma}\right]$ following the standard notation:

$$Q_{T^*,r}^\pi(s,a) = \mathbb{E}_{T^*,\pi}\left[\sum_{i=0}^\infty \gamma^i r(s_i, a_i)\Big| s_0 = s, a_0 = a\right], \quad V_{T^*,r}^\pi(s) = \mathbb{E}_{a \sim \pi(\cdot|s)}\left[Q_{T^*,r}^\pi(s,a)\right],$$

It is straightforward to see that $V_{T^*,r}^\pi = \mathbb{E}_{s \sim d_0}\left[V_{T^*,r}^\pi(s)\right]$, as well as the following Bellman equation:

$$Q_{T^*,r}^\pi(s,a) = r(s,a) + \gamma \mathbb{E}_{s' \sim T^*(\cdot|s,a)}\left[V_{T^*,r}^\pi(s')\right].$$

We also define the discounted occupancy measure $d_{T^*}^\pi$ of policy $\pi$ as follows:

$$d_{T^*}^\pi(s,a) = \mathbb{E}_{T^*,\pi}\left[\sum_{i=0}^\infty \gamma^i \mathbf{1}_{s_i=s, a_i=a}\Big| s_0 \sim d_0\right].$$

By the definition of the discounted occupancy measure, we can see $V_{T^*,r}^\pi = \mathbb{E}_{(s,a) \sim d_{T^*}^\pi}\left[r(s,a)\right]$. Furthermore, with the property of the Markov chain, we can obtain

$$d_{T^*}^\pi(s,a) = (1-\gamma)d_0 \cdot \pi(a|s) + \gamma \mathbb{E}_{(\widetilde{s},\widetilde{a}) \sim d_{T^*}^\pi(s,a)}\left[T^*(s|\widetilde{s},\widetilde{a}) \times \pi(a|s)\right].$$

## 2.2 Linear MDP

In the tabular MDP, where the state space $|\mathcal{S}|$ is finite, there exist lots of work on sample- and computation-efficient RL algorithms (e.g. Azar et al., 2017; Jin et al., 2018). However, such methods can still be expensive when $|\mathcal{S}|$ becomes large or even infinite, which is quite common for in real-world applications. To address this issue, we would like to introduce function approximations into RL algorithms to alleviate the statistical and computational bottleneck. The linear MDP (Jin et al., 2020; Agarwal et al., 2020) is a promising subclass admits special structure for such purposes.

**Definition 1** (Linear MDP (Jin et al., 2020; Agarwal et al., 2020))**.** *An MDP is called a linear MDP if there exists* $\phi^* : \mathcal{S} \times \mathcal{A} \rightarrow \mathcal{H}$ *and* $\mu^* : \mathcal{S} \rightarrow \mathcal{H}$ *for some proper Hilbert space* $\mathcal{H}$, *such that* $T^*(s'|s,a) = \langle \phi^*(s,a), \mu^*(s') \rangle_\mathcal{H}$.

The complete definition of linear MDPs require $\phi^*$ and $\mu^*$ satisfy certain normalization conditions, which we defer to Section 4 for the ease of presentation. The most significant benefit for linear MDP is that, for any policy $\pi : \mathcal{S} \rightarrow \mathcal{A}$, $Q_{T^*,r}^\pi(s,a)$ is linear with respect to $[r(s,a), \phi^*(s,a)]$, thanks to the following observation:

$$Q_{T^*,r}^\pi(s,a) = r(s,a) + \gamma \mathbb{E}_{s' \sim T^*(\cdot|s,a)}\left[V_{T^*,r}^\pi(s')\right] = r(s,a) + \left\langle \phi^*(s,a), \int_\mathcal{S} \mu^*(s')V_{T^*,r}^\pi(s')ds' \right\rangle_\mathcal{H}. \tag{1}$$

Plenty of sample-efficient algorithms have been developed based on the linear MDP structure with known $\phi^*$ (*e.g.* Yang & Wang, 2020; Jin et al., 2020; Yang et al., 2020). This requirement limits their practical applications. In fact, in most cases, we do not have access to $\phi^*$ and we need to perform representation learning to obtain an estimate of $\phi^*$. However, the learning of $\phi$ relies on efficient exploration for the full-coverage data, while the design of exploration strategy relies on the accurate estimation of $\phi$. The coupling between exploration and learning induces extra difficulty.

Recently, Uehara et al. (2022) designed UCB-style exploration for iterative finite-dimension representation updates with theoretical guarantees. The algorithm requires the computaiton oracle for the maximum likelihood estimation (MLE) to the conditional density estimation,

$$\max_{\phi,\mu} \sum_{i=1}^n \log\langle \phi(s_i,a_i), \mu(s_i') \rangle_\mathcal{H}, \quad \text{s.t.} \quad \forall(s,a), \quad \left\langle \phi(s,a), \int_\mathcal{S} \mu(s')ds' \right\rangle_\mathcal{H} = 1, \tag{2}$$

which is difficult as we generally do not have specific realization of $(\phi, \mu)$ pairs to make the constraints hold for arbitrary $(s,a)$ pairs, and therefore, impractical for real-world applications.

## 3 Latent Variable Models as Linear MDPs

In this section, we first reveal the linear representation view of the transitions with a latent variable structure. This essential connection brings several benefits for learning, planning and exploration/exploitation. More specifically, the latent variable model view provides us a tractable variational learning scheme, while the linear representation view inspires computational-efficient planning and exploration/exploitation mechanism.

We focus on the transition operator $T^*: \mathcal{S} \times \mathcal{A} \to \Delta(\mathcal{S})$ with a latent variable structure, *i.e.*, there exist latent space $\mathcal{Z}$ and two conditional probability measure $p^*(z|s,a)$ and $p^*(s'|z)$, such that

$$T^*(s'|s,a) = \int_{\mathcal{Z}} p^*(z|s,a)p^*(s'|z)d\mu, \tag{3}$$

where $\mu$ is the Lebesgue measure on $\mathcal{Z}$ when $\mathcal{Z}$ is continuous and $\mu$ is the counting measure on $\mathcal{Z}$ when $\mathcal{Z}$ is discrete.

Assume that $p^*(\cdot|s,a) \in L_2(\mu)$, $p^*(s'|\cdot) \in L_2(\mu)$, we have the equivalent formulation of (3) as
$$T^*(s'|s,a) = \langle p^*(\cdot|s,a), p^*(s'|\cdot) \rangle_{L_2(\mu)},$$

which obviously demonstrates the linear MDP structure following Defintion 1, and immediately implies $\phi^*(s,a) = p_z^*(\cdot|s,a)$, and $\mu^*(s') = p^*(s'|\cdot)$. We call $p_z^*(\cdot|s,a)$ as *Latent Variable Representation (LV-Rep)*.

**Connection to Ren et al. (2022b).**   To provide a concrete example of LV-rep, we consider the stochastic nonlinear control model with Gaussian noise (Ren et al., 2022b), which is widely used in most of model-based RL algorithms. Such a model can be understood as a special case of LV-Rep. In Ren et al. (2022b), the transition operator is defined as

$$T^*(s'|s,a) = \left(2\pi\sigma^2\right)^{-d/2} \exp\left(-\|s' - f^*(s,a)\|^2 / (2\sigma^2)\right) = \langle p^*(\cdot|s,a), p^*(s'|\cdot) \rangle_{L_2(\mu)}, \tag{4}$$

where $p^*(z|s,a) \propto \exp\left(-2\|z - f^*(s,a)\|^2 / \sigma^2\right)$ and $p^*(s'|z) \propto \exp\left(-2\|z - s'\|^2 / \sigma^2\right)$, both following the Gaussian distributions. The proposed LV-Rep can exploit more general distributions beyond Gaussian for $p^*(\cdot|s,a)$ and $p^*(s'|z)$, that introduces more flexibility in transition modeling.

Our definition of LV-Rep is more general than the original definition (Definition 2) in Agarwal et al. (2020), which assumes $|\mathcal{Z}|$ is finite. As shown by Agarwal et al. (2020), block MDPs (Du et al., 2019; Misra et al., 2020) with finite latent state space $\mathcal{Z}$ have a latent variable representation where $\mathcal{S}$ corresponds to the set of observation, $\mathcal{Z}$ corresponds to the set of latent state, and $p^*(z'|s,a)$ is a composition of deterministic $p(z|s)$ and a transition $p(z'|z,a)$. Agarwal et al. (2020) also remarks that, compared with the latent variable representation, the original low-rank representation relaxes the simplex constraint on the $p^*(z|s,a)$, and thus, can be more compact with fewer dimensions. However, the ambient dimension may not be a proper measure of the representation complexity. As we will show in Section 4, even we work on the infinite $\mathcal{Z}$, as long as $p(z|s,a) \in \mathcal{H}_k$ and $k$ satisfies standard regularity conditions, we can still perform sample-efficient learning. A proper measure of the representation complexity is still an open problem to the whole community.

The LV-Rep with $p^*(\cdot|s,a)$ and $p^*(s'|\cdot)$ naturally satisfies the distribution requirements, which brings the benefits of efficient sampling and learning.

**Efficient Simulation from LV-Rep.**   Specifically, we can easily draw samples from the learned model $\hat{T}(s'|s,a) = \int_{\mathcal{Z}} \hat{p}(z|s,a)\hat{p}(s'|z)d\mu$ by first sampling $z_i \sim \hat{p}(z|s,a)$, then sampling $s_i' \sim p(s'|z_i)$, without the need to call other complicated samplers, *e.g.*, MCMC, for the general unnormalized transition operator in linear MDPs. Such a property is important for computation-efficient planning on the learned model.

**Variational Learning of LV-Rep.**   Another significant benefit of the LV-Rep is that, we can leverage the variational method to obtain a tractable surrogate objective of MLE, which is also known as the evidence lower bound (ELBO) (Kingma et al., 2019), that can be derived as follows:

$$\log T(s'|s,a) = \log \int p^*(z|s,a)\, p^*(s'|z)dz = \log \int \frac{p^*(z|s,a)\, p^*(s'|z)}{q(z|s,a,s')} q(z|s,a,s')dz$$
$$= \max_{q \in \Delta(\mathcal{Z})} \mathbb{E}_{z \sim q(\cdot|s,a,s')} \left[\log p^*(s'|z)\right] - D_{\mathrm{KL}}\left(q(z|s,a,s')\|p^*(z|s,a)\right), \tag{5}$$

where $q(z|s,a,s')$ is an auxiliary distribution. The last equality comes from Jensen's inequality, and the equality only holds when $q(z|s,a,s') = p(z|s,a,s') \propto p(z|s,a)\, p(s'|z)$.

Compared with the standard MLE used in (Agarwal et al., 2020; Uehara et al., 2022), maximizing the ELBO is more computation-efficient, as it avoids the computation of integration at any time. Meanwhile, if the family of variational distribution $q$ is sufficient flexible that contains the optimal $p(z|s,a,s')$ for any possible $(p(z|s,a), p(s'|z))$ pair, then maximizing the ELBO is equivalent to perform MLE, *i.e.*, they share the same solution,

---

**Algorithm 1** Online Exploration with LV-Rep

---
1: **Input:** Model class $\mathcal{P} = \{(p(z|s,a), p(s'|z))\}$, $\mathcal{Q} = \{q(z|s,a,s')\}$, Iteration $N$.
2: **Initialize** $\pi_0(s) = \mathcal{U}(\mathcal{A})$ where $\mathcal{U}(\mathcal{A})$ denotes the uniform distribution on $\mathcal{A}$; $\mathcal{D}_0 = \emptyset$; $\mathcal{D}'_0 = \emptyset$.
3: **for** episode $n = 1, \cdots, N$ **do**
4:    Collect the transition $(s, a, s', a', \tilde{s})$ where $s \sim d_{T^*}^{\pi_{n-1}}$, $a \sim \mathcal{U}(\mathcal{A})$, $s' \sim T^*(\cdot|s,a)$, $a' \sim \mathcal{U}(\mathcal{A})$, $\tilde{s} \sim T^*(\cdot|s',a')$. $\mathcal{D}_n = \mathcal{D}_{n-1} \cup \{s,a,s'\}$, $\mathcal{D}'_n = \mathcal{D}'_{n-1} \cup \{s',a',\tilde{s}\}$.
5:    Learn the latent variable model $\hat{p}_n(z|s,a)$ with $\mathcal{D}_n \cup \mathcal{D}'_n$ via maximizing the ELBO in (5), and obtain the learned model $\hat{T}_n$.
6:    Set the exploration bonus $\hat{b}_n(s,a)$ as (7).
7:    Update policy $\pi_n = \arg\max_\pi V^\pi_{\hat{T}_n, r+\hat{b}_n}$.
8: **end for**
9: **Return** $\pi_1, \cdots, \pi_N$.

---

### 3.1 REINFORCEMENT LEARNING WITH LV-REP

As the transition operator is linear with respect to LV-Rep, the state-action function for arbitrary policy can be linearly represented by LV-Rep. Once the LV-Rep is learned, we can execute planning and exploration in the linear space formed by LV-Rep. Due to the space limit, we mainly consider online exploration setting, and the offline policy optimization is explained in Appendix B.

**Practical Parameterization of $Q$ function.** With the linear factorization of dynamics through latent variable models (1), we have

$$Q^\pi_{T^*,r}(s,a) = r(s,a) + \gamma \mathbb{E}_{p^*(z|s,a)}[w^\pi(z)], \tag{6}$$

where $w^\pi(z) = \int_{\mathcal{S}} p^*(z|s') V^\pi_{T^*,r}(s') ds'$ can be viewed as a value function of the latent state. When the latent variable is in finite dimension, *i.e.*, $|\mathcal{Z}|$ is finite, we have $w = [w(z)]_{z \in \mathcal{Z}} \in \mathbb{R}^{|\mathcal{Z}|}$, and the expectation $\mathbb{E}_{p^*(z|s,a)}[w^\pi(z)]$ can be computed exactly by enumerating over $\mathcal{Z}$.

However, when $\mathcal{Z}$ is not a finite set, generally we can not exactly compute the expectation, which makes the representation of $Q$ function through $p^*(z|s,a)$ hard. Particularly, under our normalization condition Assumption 2 shown later, we have $w^\pi \in \mathcal{H}_k$ where $\mathcal{H}_k$ is a reproducing kernel Hilbert space with kernel $k$. When $k$ admits a random feature representation (see Definition 13), we can then express $w^\pi$ as:

$$w^\pi(z) = \int_\Xi \widetilde{w}^\pi(\xi) \psi(z;\xi) dP(\xi),$$

where the concrete $P(\xi)$ depends on the kernel $k$. Plug this representation of $w^\pi(z)$ into (6), we obtain the approximated representation of $Q^\pi_{T^*,r}(s,a)$ as:

$$\begin{aligned}
Q^\pi_{T^*,r}(s,a) &= r(s,a) + \gamma \int_{\mathcal{Z}} w^\pi(z) p^*(z|s,a) d\mu \\
&= r(s,a) + \gamma \int_{\mathcal{Z}} \int_\Xi \widetilde{w}(\xi) \psi(z;\xi) dP(\xi) \cdot p^*(z|s,a) d\mu \\
&\approx r(s,a) + \frac{\gamma}{m} \sum_{i \in [m]} \widetilde{w}(\xi_i) \psi(z_i;\xi_i),
\end{aligned}$$

which shows that we can approximate $Q^\pi_{T^*r}(s,a)$ with a linear function on top of the random feature $\varphi(s,a) = [\psi(z_i;\xi_i)]_{i \in [m]}$ where $z_i \sim p^*(z|s,a)$ and $\xi_i \sim P(\xi)$. This can be viewed as a two-layer neural network with fixed first layer weight $\xi_i$ and activation $\psi$ and trainable second layer weight $\widetilde{w} = [\widetilde{w}(\xi_i)]_{i=1}^m \in \mathbb{R}^m$.

**Planning and Exploration with LV-Rep.** Following the idea of REP-UCB (Uehara et al., 2022), we introduce an additional bonus to implement the principle of optimism in the face of uncertainty. We use the standard elliptical potential for the upper confidence bound, which can be computed efficiently as below,

$$\hat{\varphi}_n(s,a) = [\psi(z_i;\xi_i)]_{i \in [m]}, \quad \text{where} \quad \{z_i\}_{i \in [m]} \sim \hat{p}_n(z|s,a), \quad \{\xi_i\}_{i \in [m]} \sim P(\xi),$$

$$\hat{b}_n(s,a) = \alpha_n \hat{\varphi}_n(s,a) \hat{\Sigma}_n^{-1} \hat{\varphi}_n(s,a), \quad \text{with} \quad \hat{\Sigma}_n = \sum_{(s_i,a_i) \in \mathcal{D}_n} \hat{\varphi}_n(s_i,a_i) \hat{\varphi}_n(s_i,a_i)^\top + \lambda I, \tag{7}$$

where $\alpha_n$ and $\lambda$ are some constants, and $\mathcal{D}_n$ is the collected dataset.

The planning can be then completed by Bellman recursion with bonus, *i.e.*,

$$Q^\pi(s,a) = r(s,a) + \hat{b}_n(s,a) + \gamma \mathbb{E}_T[V^\pi(s')]. \tag{8}$$

We can exploit the augmented feature $[r(s,a), \varphi(s,a), \hat{b}_n(s,a)]$ to linearly represented $Q^\pi$ after bonus introduced. However, there will be an extra $\mathcal{O}(m^2)$ due to the bonus in feature. Therefore, we consider a two-layer MLP upon $\varphi$ to parametrize $Q(s,a) = w_0 r(s,a) + \widetilde{w}_1^\top \varphi(s,a) + \widetilde{w}_2^\top \sigma(\widetilde{w}_3^\top \varphi(s,a))$, where $\sigma(\cdot)$ is a nonlinear activation function, used for complement the effect of the nonlinear $\hat{b}_n$. We finally conduct approximate dynamic programming style algorithm (*e.g.* Munos & Szepesvári, 2008) with the $Q$ parameterization.

**The Complete Algorithm.** We show the complete algorithm for the online exploration with LV-Rep in Algorithm 1. Our algorithm follows the standard protocol for sequential decision making. In each episode, the agent first executes the exploratory policy obtained from the last episode and collects the data (Line 4). The data are later used for training the latent variable model by maximizing the ELBO defined in equation 5 (Line 5). With the newly learned $\hat{p}_n(z|s,a)$, we add the exploration bonus defined in equation 7 to the reward (Line 6), and obtain the new exploratory policy by planning on the learned model with the exploration bonus (Line 7), that will be used in the next episode. Note that, in Line 4, we requires to sample $s \sim d_{T^*}^{\pi_{n-1}}$, which can be obtained by starting from $s_0 \sim d_t$, executing $\pi_{n-1}$, stopping with probability $1 - \gamma$ at each time step $t \geq 0$ and returning $s_t$. LV-Rep can also be used for offline exploitation, and we defer the corresponding algorithm to Appendix B.

## 4 THEORETICAL ANALYSIS

In this section, we provide the theoretical analysis of representation learning with LV-Rep. Before we start, we introduce the following two assumptions, that are widely used in the community (e.g. Agarwal et al., 2020; Uehara et al., 2022).

**Assumption 1** (Finite Candidate Class with Realizability). $|\mathcal{P}| < \infty$ and $(p^*(z|s,a), p^*(s'|z)) \in \mathcal{P}$. *Meanwhile, for all* $(p(z|s,a), p(s'|z)) \in \mathcal{P}$, $p(z|s,a,s') \in \mathcal{Q}$.

*Remark* 1. The assumption on $\mathcal{P}$ is widely used in the community (e.g. Agarwal et al., 2020; Uehara et al., 2022), while the assumption on $\mathcal{Q}$ is to guarantee the estimator obtained by maximizing the ELBO defined in equation 5 is identical to the estimator obtained by MLE. We would like to remark that, the extension to other data-independent function class complexity (e.g. Rademacher complexity (Bartlett & Mendelson, 2002)) can be straightforward with a refined non-asymptotic generalization bound of MLE.

**Assumption 2** (Normalization Conditions). $\forall P \in \mathcal{P}, (s,a) \in \mathcal{S} \times \mathcal{A}, \|p(\cdot|s,a)\|_{\mathcal{H}_k} \leq 1$. *Furthermore,* $\forall g : \mathcal{S} \to \mathbb{R}$ *such that* $\|g\|_\infty \leq 1$, *we have* $\left\| \int_\mathcal{S} p(s'|\cdot)g(s')ds' \right\|_{\mathcal{H}_k} \leq C$.

*Remark* 2. Our assumptions on normalization conditions is substantially different from standard linear MDPs. Specifically, standard linear MDPs assume that the representation $\phi(s,a)$ and $\mu(s')$ are of finite dimension $d$, with $\|\phi(s,a)\|_2 \leq 1$ and $\forall \|g\|_\infty \leq 1$, $\left\| \int_\mathcal{S} \mu(s')g(s')ds' \right\|_2 \leq d$. When $|\mathcal{Z}|$ is finite, as $\|f\|_{L_2(\mu)} \leq \|f\|_{\mathcal{H}_k}$, our normalization conditions are more general than the counterparts of the standard linear MDPs and we can use the identical normalization conditions as the standard linear MDPs. However, when $|\mathcal{Z}|$ is infinite, if we assume $\|p(z|s,a)\|_{L_2(\mu)} \leq 1$, we cannot provide a sample complexity bound without polynomial dependency on $|\mathcal{P}|$, which can be unsatisfactory. Furthermore, we would like to note that, the assumption $\int_\mathcal{S} p(s'|z)g(s')ds' \in \mathcal{H}_k$ is mild, which is necessary for justifying the estimation from the approximate dynamic programming algorithm.

**Theorem 1** (PAC Guarantee for Online Exploration, Informal). *Assume the reproducing kernel* $k$ *satisfies the regularity conditions in Appendix E.1. If we properly choose the exploration bonus* $\hat{b}_n(s,a)$, *we can obtain an* $\varepsilon$-*optimal policy with probability at least* $1 - \delta$ *after we interact with the environments for* $N = \text{poly}\left(C, |\mathcal{A}|, (1-\gamma)^{-1}, \varepsilon, \log(|\mathcal{P}|/\delta)\right)$ *episodes.*

*Remark* 3. Although $|\mathcal{Z}|$ may not be finite, we can still obtain a sample complexity independent w.r.t. $|\mathcal{S}|$, while has polynomial dependency on $C$, $|\mathcal{A}|$, $(1-\gamma)^{-1}$ and $\varepsilon$ and $\log |\mathcal{P}|$ with the assumption that $p(\cdot|s,a) \in \mathcal{H}_k$ and some standard regularity conditions for the kernel $k$. This means that we do not really need to assume a discrete $\mathcal{Z}$ with finite cardinality, but only need to properly control the complexity of the representation class, by either the ambient dimension $|\mathcal{Z}|$, or some "effective dimension" that can be derived from the eigendecay of the kernel $k$ (see Appendix E.1 for the details). The formal statement for Theorem 1 and the proof is deferred to Appendix E.2. We also provide the PAC guarantee for offline exploitation with LV-Rep in Appendix E.3.

*Remark* 4. Our proof strategy is based on the analysis of REP-UCB (Uehara et al., 2022). However, there are substantial differences between our analysis and the analysis of REP-UCB, as the representation we consider can be infinite-dimensional, and hence the analysis of REP-UCB, which assumes that the feature is finite-dimensional, cannot be directly applied in our case. As we mentioned, to address the infinite dimension issue, we assume the representation $p(z|s,a) \in \mathcal{H}_k$ and prove two novel lemmas, one for the concentration of bonus (Lemma 17) and one for the ellipsoid potential bound (Lemma 19) when the representation lies in the RKHS. We further note that, different from the work on the kernelized bandit and kernelized MDP (Srinivas et al., 2010; Valko et al., 2013; Yang et al., 2020) that assume the reward function and $Q$ function lies in some RKHS, we assume the condition density of the latent random variable lies in the RKHS and the $Q$ function is the $L_2(\mu)$ inner product of two functions in RKHS. As a result, the techniques used in their work cannot be directly adapted to our setting, and their regret bounds depend on the alternative notions of maximum information gain and effective dimension of the specific kernel, which can be implied by the eigendecay conditions we assume in Appendix E.1 (see Yang et al. (2020) for the details).

## 5    RELATED WORK

There are several other theoretically grounded representation learning methods under the assumption of linear MDP. However, most of these work either consider more restricted model or totally ignore the computation issue. Du et al. (2019); Misra et al. (2020) focused on the representation learning in block MDPs, which is a special case of linear MDP (Agarwal et al., 2020), and proposed to learn the representation via the regression. However, both of them used policy-cover based exploration that need to maintain large amounts of policies in the training phase, which induces a significant computation bottleneck. Uehara et al. (2022) and Zhang et al. (2022b) exploit UCB upon learned representation to resolve this issue. However, their algorithms depend on some computational oracles, *i.e.*, MLE for unnormalized conditional distribution in (2) or a $\max - \min - \max$ optimization solver motivated from Modi et al. (2021), respectively, that can be hard to implement in practice.

A variety of recent work have been proposed to replace the computational oracle with more tractable estimators. For example, Ren et al. (2022b) exploited representation with the structure of Gaussian noise in nonlinear stochastic control problem with arbitrary dynamics, which restricts the flexibility. Zhang et al. (2022a); Qiu et al. (2022) proposed to use a contrastive learning approach as an alternative. However, similar to other contrastive learning approach, both of their methods require the access to a negative distribution supported on the whole state space, and their performance highly depends on the quality of the negative distribution. Ren et al. (2022a) designed a new objective based on the idea of the spectral decomposition. But the solution for their objective is not necessarily to be a valid distribution, and the generalization bound is worse than the MLE when the state space is finite.

Algorithmically, many representation learning methods have been developed for different purposes, such as state extractor from vision-based features (Laskin et al., 2020a;b; Kostrikov et al., 2020), bi-simulation (Ferns et al., 2004; Gelada et al., 2019; Zhang et al., 2020), successor feature (Dayan, 1993; Barreto et al., 2017; Kulkarni et al., 2016), spectral representation from transition operator decomposition (Mahadevan & Maggioni, 2007; Wu et al., 2018; Duan et al., 2019), contrastive learning (Oord et al., 2018; Nachum & Yang, 2021; Yang et al., 2020), and so on. However, most of these methods are designed for state-only feature, ignoring the action dependency, and learning from pre-collected datasets, without taking the planning and exploration in to account and ignoring the coupling between representation learning and exploratin. Therefore, there is no rigorously theoretical characterization provided.

We would like to emphasize that the proposed LV-Rep is the algorithm which achieves both statistical efficiency theoretically and computational tractability empirically. For more related work on model-based RL, please refer to Appendix A.

## 6    EXPERIMENTS

We extensively test our algorithm on the Mujoco (Todorov et al., 2012) and DeepMind Control Suite (Tassa et al., 2018). Before presenting the experiment results, we first discuss some details towards a practical implementation of LV-Rep.

Table 1: Performance on various MuJoCo control tasks. All the results are averaged across 4 random seeds and a window size of 10K. Results marked with * is adopted from MBBL. LV-Rep-C and LV-Rep-D use continuous and discrete latent variable model respectively. LV-Rep achieves strong performance compared with baselines.

| | | HalfCheetah | Reacher | Humanoid-ET | Pendulum | I-Pendulum |
|---|---|---|---|---|---|---|
| Model-Based RL | ME-TRPO* | 2283.7±900.4 | -13.4±5.2 | 72.9±8.9 | **177.3±1.9** | -126.2±86.6 |
| | PETS-RS* | 966.9±471.6 | -40.1±6.9 | 109.6±102.6 | 167.9±35.8 | -12.1±25.1 |
| | PETS-CEM* | 2795.3±879.9 | -12.3±5.2 | 110.8±90.1 | 167.4±53.0 | -20.5±28.9 |
| | Best MBBL | 3639.0±1135.8 | **-4.1±0.1** | 1377.0±150.4 | **177.3±1.9** | **0.0±0.0** |
| Model-Free RL | PPO* | 17.2±84.4 | -17.2±0.9 | 451.4±39.1 | 163.4±8.0 | -40.8±21.0 |
| | TRPO* | -12.0±85.5 | -10.1±0.6 | 289.8±5.2 | 166.7±7.3 | -27.6±15.8 |
| | SAC* (3-layer) | 4000.7±202.1 | -6.4±0.5 | **1794.4±458.3** | 168.2±9.5 | -0.2±0.1 |
| Representation RL | DeepSF | 4180.4±113.8 | -16.8±3.6 | 168.6±5.1 | 168.6±5.1 | -0.2±0.3 |
| | SPEDE | 4210.3±92.6 | -7.2±1.1 | 886.9±95.2 | 169.5±0.6 | 0.0±0.0 |
| | **LV-Rep-C** | **5557.6±439.5** | **-5.8±0.3** | 1086±278.2 | 167.1±3.1 | **0.0±0.0** |
| | **LV-Rep-D** | **4616.5±261.5** | **-6.0±0.2** | 1359.2 ±198.6 | 170.2 ± 4.2 | **0.0±0.0** |

| | | Ant-ET | Hopper-ET | S-Humanoid-ET | CartPole | Walker-ET |
|---|---|---|---|---|---|---|
| Model-Based RL | ME-TRPO* | 42.6±21.1 | 1272.5±500.9 | -154.9±534.3 | 160.1±69.1 | -1609.3±657.5 |
| | PETS-RS* | 130.0±148.1 | 205.8±36.5 | 320.7±182.2 | 195.0±28.0 | 312.5±493.4 |
| | PETS-CEM* | 81.6±145.8 | 129.3±36.0 | 355.1±157.1 | 195.5±3.0 | 260.2±536.9 |
| | Best MBBL | 275.4±309.1 | 1272.5±500.9 | **1084.3±77.0** | **200.0±0.0** | 312.5±493.4 |
| Model-Free RL | PPO* | 80.1±17.3 | 758.0±62.0 | 454.3±36.7 | 86.5±7.8 | 306.1±17.2 |
| | TRPO* | 116.8±47.3 | 237.4±33.5 | 281.3±10.9 | 47.3±15.7 | 229.5±27.1 |
| | SAC* (3-layer) | 2012.7±571.3 | 1815.5±655.1 | 834.6±313.1 | **199.4±0.4** | **2216.4±678.7** |
| Representation RL | DeepSF | 768.1±44.1 | 548.9±253.3 | 533.8±154.9 | 194.5±5.8 | 165.6±127.9 |
| | SPEDE | 806.2±60.2 | 732.2±263.9 | 986.4±154.7 | 138.2±39.5 | 501.6±204.0 |
| | **LV-Rep-C** | **2511.8±460.0** | **2204.8±496.0** | 963.1±45.1 | **200.7±0.2** | **2523.5±333.9** |
| | **LV-Rep-D** | **2436.0±603.1** | **2187.5±453.6** | 956.8± 87.5 | **198.5 ± 2.0** | **2209.0±589.2** |

## 6.1 IMPLEMENTATION DETAILS

As discussed, the latent variable representation is learned by minimizing the ELBO (5). We consider two practical implementations. The first one applies a continuous latent variable model, where the distributions are approximated using Gaussian with parameterized mean and variance, similarly to (Hafner et al., 2019b). We call this method LV-Rep-C. The second implementation considers using a discrete sparse latent variable model (Hafner et al., 2019b), which we call LV-Rep-D. As discussed in line 7 of Algorithm 1, we apply a planning algorithm with the learned latent representation to improve the policy. We use *Soft Actor Critic (SAC)* (Haarnoja et al., 2018) as our planner, where the critic is parameterized as shown in (6). In practice, we find that it is beneficial to have more updates for the latent variable model than critic. We also use a target network for the latent variable model to stabilize training.

## 6.2 DENSE-REWARD MUJOCO BENCHMARKS

We first conduct experiments on dense-reward Mujoco locomotion control tasks, which are commonly used test domains for both model-free and model-based RL algorithms. We compare LV-Rep with model-based algorithms, including ME-TRPO (Kurutach et al., 2018), PETS (Chua et al., 2018), and the best model-based results from (Wang et al., 2019), among 9 baselines (Luo et al., 2018; Deisenroth & Rasmussen, 2011; Heess et al., 2015; Clavera et al., 2018; Nagabandi et al., 2018; Tassa et al., 2012; Levine & Abbeel, 2014), as well as model-free algorithms, including PPO (Schulman et al., 2017), TRPO (Schulman et al., 2015) and SAC (Haarnoja et al., 2018).

We compare all algorithms after running $200K$ environment steps. Table 1 presents all experiment results, where all results are averaged over 4 random seeds. In practice we found LV-Rep-C provides comparable or better performance (see Figure 1 for example), so that we report its result for LV-Rep in the table. We present the best model-based RL performance for comparison. The results clearly show that LV-Rep provides significant better or comparable performance compared to all model-based algorithms. In particular, in the most challenging domains such as Walker and Ant, most model-based methods completely fail the task, while LV-Rep achieves the state-of-the-art performance in contrast. Furthermore, LV-Rep show dominant performance in all domains comparing to two representative representation learning based RL methods, Deep Successor Feature (DeepSF) (Barreto et al., 2017) and SPEDE (Ren et al., 2022b). LV-Rep also achieves better performance than the strongest model-free algorithm SAC in most challenging domains except Humanoid.

Table 2: Performance of on various Deepmind Suite Control tasks. All the results are averaged across four random seeds and a window size of 10K. Comparing with SAC, our method achieves even better performance on sparse-reward tasks.

|  |  | cheetah_run | walker_run | humanoid_run | hopper_hop |
|---|---|---|---|---|---|
| Model-Based RL | Dreamer | 542.0 ± 27.7 | 337.7±67.2 | 1.0±0.2 | 46.1±17.3 |
| Model-Free RL | PPO | 227.7±57.9 | 51.6±1.5 | 1.1±0.0 | 0.7±0.8 |
|  | SAC | 453.4±57.9 | 488.5±40.2 | 1.1±0.1 | 10.8±6.6 |
| Representation RL | DeepSF | 295.3±43.5 | 27.9±2.2 | 0.9±0.1 | 0.3±0.1 |
|  | Proto RL | 305.5±37.9 | 433.5±56.8 | 0.3±0.6 | 1.0±0.2 |
|  | **LV-Rep** | **639.3±24.5** | **724.2±37.8** | **11.8±6.8** | **72.9±40.6** |

Finally, we provide learning curves of LV-Rep-C and LV-Rep-D in comparison to SAC in Figure 1, which clearly shows that comparing to the SOTA model-free baseline SAC, LV-Rep enjoys great sample efficiency in these tasks.

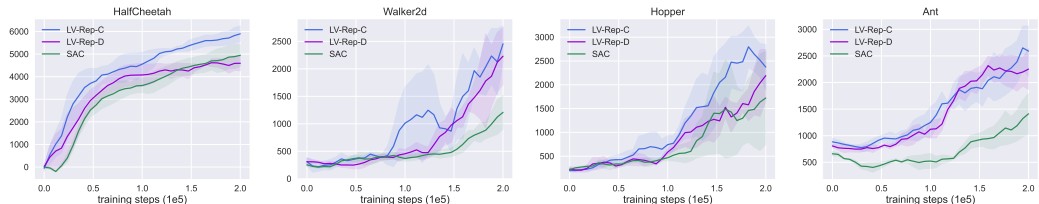

Figure 1: We show the learning curves in Mujoco control compared to the baseline algorithms. The $x$-axis shows the training iterations and $y$-axis shows the performance. All plots are averaged over 4 random seeds. The shaded area shows the standard error. We only compare to SAC as it has the best overall performance in all baseline methods.

## 6.3 SPARSE-REWARD DEEPMIND CONTROL SUITE

In this experiment we show the effectiveness of our proposed methods in sparse reward problems. We compare LV-Rep with the state-of-the-art model-free RL methods including SAC and PPO. Since the proposed LV-Rep significantly dominates all the model-based RL algorithms in MuJoCo from Wang et al. (2019), we consider a different model-based RL method, *i.e.*, Dreamer (Hafner et al., 2019b), and add another representation-based RL methods, *i.e.*, Proto-RL (Yarats et al., 2021), besides DeepSF (Barreto et al., 2017).

We compare all algorithms after running 200K environment steps across 4 random seeds. Results are presented in Table 2. We report the result of LV-Rep-C for LV-Rep as it gives better empirical performance. We can clearly observe that LV-Rep dominates the performance across all domains. In relatively dense-reward problems, *cheetah-run* and *walker-run*, LV-Rep outperforms all baselines by a large margin. Remarkably, for sparse reward problems, *hopper-hop* and *humanoid-run*, LV-Rep provides reasonable results while other methods do not even start learning.

We also plot the learning curves of LV-Rep with all competitors in Figure 2. This shows that LV-Rep outperforms other baselines in terms of both sample efficiency and final performance.

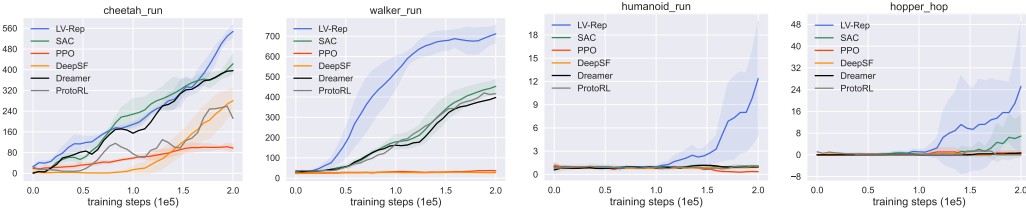

Figure 2: We show the results in DeepMind Control Suite compared to the baseline algorithms. The $x$-axis shows the training iterations and $y$-axis shows the performance. All plots are averaged over 4 random seeds. The shaded area shows the standard error.

## 7 CONCLUSION

In this paper, we reveal the representation view of latent variable dynamics model, which induces the *Latent Variable Representation (LV-Rep)*. Based on the LV-Rep, a new provable and practical algorithm for reinforcement learning is proposed, achieving the balance between flexibility and efficiency in terms of statistical complexity, with tractable learning, planning and exploration. We provide rigorous theoretical analysis, which is applicable for LV-Rep with both finite- and infinite-dimension and comprehensive empirical justification, which demonstrates the superior performances.

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

## A  MORE RELATED WORK

Our method is also closely related to the model-based reinforcement learning. These methods maintain an estimation of the dynamics and reward from the data, and extract the optimal policy via planning modules. The major differences among these methods are in terms of **i**), model parameterization and learning objectives, and **ii**), the approximated algorithms used for planning.

Specifically, Gaussian processes are exploited in (Deisenroth & Rasmussen, 2011). A stochastic deep dynamics with restricts Gaussian noise assumption is widely used (Heess et al., 2015; Kurutach et al., 2018; Chua et al., 2018; Clavera et al., 2018; Nagabandi et al., 2018). Hafner et al. (2019a;b; 2020); Lee et al. (2020) recently exploits recurrent latent state space model, but focused on Partially Observable MDP setting, which is beyond the scope of our paper. Different approximated planning algorithms, including Dyna-style, shooting, and policy search with backpropagation through time, have been tailored in these methods. Please refer to (Wang et al., 2019) for detailed discussion.

As we discussed in Section 1, these algorithms did not balance the flexibility in modeling and tractability in planning and exploration, which may lead to sub-optimal performances. While the proposed LV-Rep not only is more flexible beyond Gaussian noise assumption, but also lead to provable and tractable learning, planning, and exploration, and thus, achieving better empirical performances.

## B  ALGORITHMS AND THEORETICAL ANALYSIS FOR OFFLINE EXPLOITATION

---

**Algorithm 2** Offline Exploitation with LV-Rep
___
 1: **Input:** Model class $\mathcal{P} = \{(p(z|s, a), p(s'|z))\}$, $\mathcal{Q} = \{q(z|s, a, s')\}$, Offline Dataset $\mathcal{D}_n$.
 2: Learn the latent variable model $\hat{p}(z|s, a)$ with $\mathcal{D}_n$ via maximizing the ELBO defined in equation 5, and obtain the learned model $\hat{T}$.
 3: Set the exploitation penalty $\hat{b}(s, a)$ as equation 7.
 4: Obtain policy $\hat{\pi} = \arg\max_{\pi} V_{\hat{T}, r - \hat{b}}^{\pi}$.
 5: **Return** $\hat{\pi}$.

---

In this section, we show the algorithms for offline exploitation. For offline exploitation, we have the access to a offline dataset, which we assume is collected from the stationary distribution of the fixed behavior policy $\pi_b$, which we will denote as $\rho$. And we are not allowed to interact with the environments to collect new data. The only difference between the algorithms for offline exploitation and online exploration is that, as we do not have access to the new data from the environment, we cannot further explore the state-action pair that the offline dataset do not cover. Hence, we need to penalize the visitation to the unseen state action pair to avoid the risky behavior.

## C  IMPLEMENTATION DETAILS

Our algorithm is implemented using Pytorch. For DeepMind control, we use an open source implementation as our SAC baseline (Yarats & Kostrikov, 2020). As discussed in Section 6.1, we find it is beneficial to have more updates for the latent variable model than critic in practice. We use a parameter *feature-updates-per-step* that decides how many updates are performed for the latent variable model at each training step. For all Mujoco and DeepMind Control experiments, we tune this parameter from $\{1, 3, 5, 10, 20\}$ and report the best result. Finally, in Table 3, we list all other hyperparameters and network architecture we use for our experiments.

For evaluation in Mujoco, in each evaluation (every 5K steps) we test our algorithm for 10 episodes. We average the results over the last 4 evaluations and 4 random seeds. For Dreamer and Proto-RL, we change their network from CNN to 3-layer MLP and disable the image data augmentation part (since we test on the state space). We tune some of their hyperparameters (e.g., exploration steps in Proto-RL) and report the best number across our runs.

Table 3: Hyperparameters used for LV-Rep in all the environments in MuJoCo and DM Control Suite.

|  | Hyperparameter Value |
| --- | --- |
| Actor lr | 0.0003 |
| Model lr | 0.0003 |
| Actor Network Size (MuJoCo) | (256, 256) |
| Actor Network Size (DM Control) | (1024, 1024) |
| LV-Rep Feature Embedding Dim (MuJoCo) | 256 |
| LV-Rep Feature Embedding Dim (DM Control) | 1024 |
| ERP Embedding Network Size (DM Control) | (1024, 1024, 1024) |
| Critic Network Size (MuJoCo) | (256, 256, 1) |
| Critic Network Size (DM Control) | (1024, 1024, 1) |
| Discount | 0.99 |
| Critic Target Update Tau | 0.005 |
| Latent Variable Target Update Tau | 0.005 |
| Batch Size | 256 |

## D  TECHNICAL BACKGROUNDS

In this section, we introduce several important concepts from functional analysis that will be repeatedly used in our theoretical analysis. We start from the concept of the $\mathbb{R}$-vector space.

**Definition 2** ($\mathbb{R}$-vector space (Steinwart & Christmann, 2008))**.** *An $\mathbb{R}$-vector space is defined as a triple $(E, +, \cdot)$, where $E$ is a non-empty set, $+ : E \times E \to E$ and $\cdot : \mathbb{R} \times E \to E$ satisfies the following properties:*

- $\forall x, y, z \in E, (x + y) + z = x + (y + z)$.

- $\forall x, y \in E, x + y = y + x$.

- $\exists$ *an element* $0 \in E$, *such that* $\forall x \in E, x + 0 = x$.

- $\forall x \in E, \exists -x \in E$, *such that* $x + (-x) = 0$.

- $\forall \alpha, \beta \in \mathbb{R}, x \in E, (\alpha\beta) \cdot x = \alpha \cdot (\beta \cdot x)$.

- $\forall x \in E, 1 \cdot x = x$.

- $\forall \alpha, \beta \in \mathbb{R}, x \in E, (\alpha + \beta) \cdot x = \alpha \cdot x + \beta \cdot x$.

- $\forall \alpha \in \mathbb{R}, x, y \in E, \alpha \cdot (x + y) = \alpha \cdot x + \alpha \cdot y$.

*The $\cdot$ denotes the scalar multiplication will be omitted if there will be no confusion.*

**Definition 3** (Norm and Banach Space (Steinwart & Christmann, 2008))**.** *Let $E$ be a $\mathbb{R}$-vector space. A map $\| \cdot \| : E \to [0, \infty)$ is a norm on $E$ if*

- $\|x\| = 0 \Leftrightarrow x = 0$.

- $\forall \alpha \in \mathbb{R}, x \in E, \|\alpha x\| = \alpha \|x\|$.

- $\forall x, y \in E, \|x + y\| \leq \|x\| + \|y\|$.

*In this case, the pair $(E, \| \cdot \|)$ is called a Banach space, and we use $E$ to denote the Banach space for simplicity if there will be no confusion.*

**Definition 4** (Bounded Linear Operator (Steinwart & Christmann, 2008))**.** *Let $E$ and $F$ be two Banach spaces. A map $S : E \to F$ is a bounded linear operator if*

- $\forall x, y \in E, S(x + y) = Sx + Sy$.

- $\forall \alpha \in \mathbb{R}, x \in E, S(\alpha x) = \alpha(Sx)$.

- $S0 = 0$.

*Furthermore, $S$ satisfies the following properties*

- $\exists c \in [0, \infty]$, such that $\forall x \in E$, $\|Sx\|_F \leq c\|x\|_E$.

*Note that, all of the bounded linear operator itself can define an $\mathbb{R}$-vector space, and we can define an operator norm of $S$ as $\|S\|_{\mathrm{op}} := \sup_{x \in \mathcal{B}_E} \|Sx\|_F$, where $\mathcal{B}_E = \{x \in E : \|x\|_E \leq 1\}$ is the unit ball of $E$.*

**Definition 5** (Compact Operator (Steinwart & Christmann, 2008)). *A bounded linear operator $S : E \to F$ is compact if the closure of $S\mathcal{B}_E$ is compact in $F$.*

**Definition 6** (Inner Product and Hilbert Space (Steinwart & Christmann, 2008)). *A map $\langle \cdot, \cdot \rangle : \mathcal{H} \times \mathcal{H} \to \mathbb{R}$ on a $\mathbb{R}$-vector space is an inner product if*

- $\forall x, y, z \in \mathcal{H}$, $\langle x + y, z \rangle = \langle x, z \rangle + \langle y, z \rangle$.

- $\forall \alpha \in \mathbb{R}, x, y \in \mathcal{H}$, $\langle \alpha x, y \rangle = \alpha \langle x, y \rangle$.

- $\forall x, y \in \mathcal{H}$, $\langle x, y \rangle = \langle y, x \rangle$.

- $\forall x \in \mathcal{H}$, $\langle x, x \rangle \geq 0$, and $\langle x, x \rangle = 0 \Leftrightarrow x = 0$.

*If the norm induced by the inner product $\|x\|_{\mathcal{H}} := \sqrt{\langle x, x \rangle}$ is complete, the pair $(\mathcal{H}, \langle \cdot, \cdot \rangle)$ is called a Hilbert space. We sometimes use $\mathcal{H}$ to denote the Hilbert space and use $\langle \cdot, \cdot \rangle_{\mathcal{H}}$ to distinguish between different inner products. Note that, the inner product satisfies the following Cauchy-Schwartz inequality:*

$$\forall x, y \in \mathcal{H}, \quad |\langle x, y \rangle_{\mathcal{H}}| \leq \|x\|_{\mathcal{H}} \|y\|_{\mathcal{H}}.$$

**Definition 7** ((Self-)Adjoint Operator (Steinwart & Christmann, 2008)). *Let $H_1$ and $H_2$ be two Hilbert spaces, For the operator $S : \mathcal{H}_1 \to \mathcal{H}_2$, the adjoint operator $S^* : \mathcal{H}_2 \to \mathcal{H}_1$ is defined by*

$$\forall x \in \mathcal{H}_1, y \in \mathcal{H}_2, \quad \langle Sx, y \rangle_{\mathcal{H}_2} = \langle x, S^*y \rangle_{\mathcal{H}_1}.$$

*Furthermore, $S$ is a self-adjoint operator, if $S : \mathcal{H}_1 \to \mathcal{H}_1$, and*

$$\forall x, y \in \mathcal{H}_1, \quad \langle Sx, y \rangle_{\mathcal{H}_1} = \langle x, Sy \rangle_{\mathcal{H}_1}.$$

*For self-adjoint operator $S$, if $\langle Sx, x \rangle \geq 0$, $S$ is called a positive semi-definite operator, and if $\langle Sx, x \rangle > 0$, $S$ is called a positive definite operator.*

*Remark* 5. Note that, the definition of the adjoint operator can be generalized to Banach spaces. But adjoint operators for Hilbert spaces are sufficient for our purposes. So we omit the definition of the adjoint operators on Banach spaces.

**Definition 8** (Orthonormal System and Orthonormal Basis (Steinwart & Christmann, 2008)). *For the Hilbert space $\mathcal{H}$, the family $\{e_i\}_{i \in I}, e_i \in \mathcal{H}$ is an orthonormal system if $\langle e_i, e_i \rangle = 1$, and $\langle e_i, e_j \rangle = 0$ if $i \neq j$. Furthermore, if the closure of the linear span of $\{e_i\}_{i \in I}$ equals to $\mathcal{H}$, it is an orthonormal basis. Note that, each Hilbert space $H$ has an orthonormal basis, and if $\mathcal{H}$ is separable, $\mathcal{H}$ has a countable orthonormal basis. Furthermore, $\forall x \in H$, we have*

$$x = \sum_{i \in I} \langle x, e_i \rangle e_i.$$

**Theorem 2** (Spectral Theorem (Steinwart & Christmann, 2008)). *Let $\mathcal{H}$ be a Hilbert space and $T : \mathcal{H} \to \mathcal{H}$ is compact and self-adjoint. Then their exists at most countable $\{\mu_i(T)\}_{i \in I}$ converging to 0 such that $|\mu_1(T)| \geq |\mu_2(T)| \geq \cdots > 0$ and an orthonormal system $\{e_i\}_{i \in I}$, such that*

$$\forall x \in \mathcal{H}, \quad Tx = \sum_{i \in I} \mu_i(T) \langle x, e_i \rangle_{\mathcal{H}} e_i.$$

*Here $\{\mu_i(T)\}_{i \in I}$ can be viewed as the set of eigen-value of $T$, as $Te_i = \mu_i(T)$.*

**Definition 9** (Trace and Trace class (Steinwart & Christmann, 2008)). *For a compact and self-adjoint operator $T$, if $\sum_{i=1}^{\infty} \mu_i(T) < \infty$, we say $T$ is nuclear or of trace class, and define the nuclear norm and the trace as:*

$$\|T\|_* = \mathrm{Tr}(T) = \sum_{i \in I} \mu_i(T).$$

**Definition 10** (Hilbert-Schmidt Operator (Steinwart & Christmann, 2008)). *Let $\mathcal{H}_1, \mathcal{H}_2$ be two Hilbert spaces. An operator $S : \mathcal{H}_1 \to \mathcal{H}_2$ is Hilbert-Schmidt if*

$$\|S\|_{\mathrm{HS}} := \left( \sum_{i \in I} \|Se_i\|_{\mathcal{H}_2}^2 \right)^{1/2} < \infty,$$

*where $\{e_i\}_{i \in I}$ is an arbitrary orthonormal basis of $\mathcal{H}_1$. Furthermore, the set of Hilbert-Schmidt operators defined on $\mathcal{H} \to \mathcal{H}$ where $\mathcal{H}$ is a Hilbert space is indeed a Hilbert space with the following inner product:*

$$\langle T_1, T_2 \rangle_{\mathrm{HS}(H)} = \sum_{i \in I} \langle T_1 e_i, T_2 e_i \rangle_{\mathcal{H}}, \quad T_1, T_2 \in \mathrm{HS}(\mathcal{H}),$$

*where $\{e_i\}_{i \in I}$ is an arbitrary orthonormal basis of $\mathcal{H}$.*

**Definition 11** ($L^2(\mu)$ space). *Let $(\mathcal{X}, \mathcal{A}, \mu)$ be a measure space. The $L^2(\mu)$ space is defined as the Hilbert space consists of square-integrable function with respect to $\mu$, with inner product*

$$\langle f, g \rangle_{L_2(\mu)} := \int_{\mathcal{X}} fg d\mu,$$

*and the norm*

$$\|f\|_{L_2(\mu)} := \left( \int_{\mathcal{X}} f^2 d\mu \right)^{1/2}.$$

*Throughout the paper, $\mu$ is specified as the Lebesgue measure for continuous $\mathcal{X}$ and the counting measure for discrete $\mathcal{X}$. Specifically, when $\mathcal{X}$ is discrete, we can represent $f$ as a sequence $[f(x)]_{x \in \mathcal{X}}$, and the corresponding $L_2(\mu)$ inner product and $L_2(\mu)$ norm is identical to the $\ell^2$ inner product and $\ell^2$ norm, which is defined as*

$$\langle f, g \rangle_{l^2} = \sum_{x \in \mathcal{X}} f(x)g(x), \quad \|f\|_{l^2} = \left( \sum_{x \in \mathcal{X}} f^2(x) \right)^{1/2},$$

*that is closely related to the inner product and norm of the Euclidean space.*

**Definition 12** (Kernel and Reproducing Kernel Hilbert Space (RKHS) (Aronszajn, 1950; Paulsen & Raghupathi, 2016)). *A function $k : \mathcal{X} \times \mathcal{X} \to \mathbb{R}$ is a kernel on non-empty set $\mathcal{X}$, if there exists a Hilbert space $\mathcal{H}$ and a feature map $\phi : \mathcal{X} \to \mathcal{H}$, such that $\forall x, x' \in \mathcal{X}$, $k(x, x') = \langle \phi(x), \phi(x') \rangle_{\mathcal{H}}$. Furthermore, if $\forall n \geq 1$, $\{a_i\}_{i \in [n]} \subset \mathbb{R}$ and mutually distinct $\{x_i\}_{i \in [n]}$,*

$$\sum_{i \in [n]} \sum_{j \in [n]} a_i a_j k(x_i, x_j) \geq 0,$$

*the kernel $k$ is said to be positive semi-definite. And if the inequality is strict, the kernel $k$ is said to be positive definite.*

*Given the kernel $k$, the Hilbert space $\mathcal{H}_k$ consists of $\mathbb{R}$-valued function on non-empty set $\mathcal{X}$ is said to be a reproducing kernel Hilbert space associated with $k$ if the following two conditions hold:*

- *$\forall x \in \mathcal{X}$, $k(x, \cdot) \in \mathcal{H}_k$.*

- *Reproducing Property: $\forall x \in \mathcal{X}$, $f \in \mathcal{H}_k$, $f(x) = \langle f, k(x, \cdot) \rangle_{\mathcal{H}_k}$.*

*Here $k$ is also called the reproducing kernel of $\mathcal{H}_k$. The RKHS norm of $f \in \mathcal{H}_k$ is defined as $\|f\|_{\mathcal{H}_k} := \sqrt{\langle f, f \rangle_{\mathcal{H}_k}}$.*

Some of the well-known kernels include:

- Linear Kernel: $k(x, x') = x^\top x'$, where $x, x' \in \mathbb{R}^d$;
- Polynomial Kernel: $k(x, x') = (1 + x^\top x')^n$, where $x, x' \in \mathbb{R}^d$, $n \in \mathbb{N}^+$.
- Gaussian Kernel: $k(x, x') = \exp\left(-\frac{\|x - x'\|_2^2}{\sigma^2}\right)$, where $x, x' \in \mathbb{R}^d$, $\sigma > 0$ is the scale parameter.
- Matérn Kernel: $k(x, x') = \frac{2^{1-\nu}}{\Gamma(\nu)} r^\nu B_\nu(r)$, where $x, x' \in \mathbb{R}^d$, $\nu > 0$ is the smoothness parameter, $l > 0$ is the scale parameter, $r = \frac{\sqrt{2\nu}}{l} \|x - x'\|$, $\Gamma(\cdot)$ is the Gamma function and $B_\nu(\cdot)$ is the modified Bessel function of the second kind.

**Theorem 3** (Mercer's Theorem (Riesz & Nagy, 2012; Steinwart & Scovel, 2012)). *Let $(\mathcal{X}, \mathcal{A}, \mu)$ be a measure space with compact support $\mathcal{X}$ and strictly positive Borel measure $\phi$. $k$ is a continuous positive definite kernel defined on $\mathcal{X} \times \mathcal{X}$. Then there exists at most countable $\{\mu_i\}_{i \in I}$ with $\mu_1 \geq \mu_2 \geq \cdots > 0$ and $\{e_i\}_{i \in I}$ where $\{e_i\}_{i \in I}$ is the set of orthonormal basis of $L_2(\mu)$, such that*

$$\forall x, x' \in \mathcal{X}, \quad k(x, x') = \sum_{i \in I} \mu_i e_i(x) e_i(x'),$$

*where the convergence is absolute and uniform.*

*Remark* 6 (Spectral Characterization of RKHS). With the represener property, we know that

$$\sum_{i \in I} \mu_i e_i(x) e_i(\cdot) = k(x, \cdot) \in \mathcal{H}_k.$$

Note that, for $\beta$-finite spectrum, we can choose $I$ such that $\mu_i > 0$. If we define the inner product

$$\left\langle \sum_{i \in I} a_i e_i(\cdot), \sum_{i \in I} b_i e_i(\cdot) \right\rangle_{\mathcal{H}_k} = \sum_{i \in I} \frac{a_i b_i}{\mu_i},$$

then we have the reproducing property

$$\left\langle \sum_{i \in I} a_i e_i(\cdot), k(x, \cdot) \right\rangle_{\mathcal{H}_k} = \left\langle \sum_{i \in I} a_i e_i(\cdot), \sum_{i \in I} \mu_i e_i(x) e_i(\cdot) \right\rangle_{\mathcal{H}_k} = \sum_{i \in I} a_i e_i(x).$$

With the spectral characterization, we know that

$$\left\| \sum_{i \in I} a_i e_i(\cdot) \right\|_{\mathcal{H}_k} = \sum_{i \in I} \frac{a_i^2}{\mu_i} \geq \frac{\sum_{i \in I} a_i^2}{\mu_1} = \frac{\left\| \sum_{i \in I} a_i e_i(\cdot) \right\|_{L_2(\mu)}}{\mu_1}.$$

Hence, we know $\forall f \in \mathcal{H}_k$, $f \in L_2(\mu)$. Furthermore, note that

$$k(x, x) = \left\langle \sum_{i \in I} \mu_i e_i(x) e_i(\cdot), \sum_{i \in I} \mu_i e_i(x) e_i(\cdot) \right\rangle_{\mathcal{H}_k} = \sum_{i \in I} \mu_i e_i^2(x).$$

Hence,

$$\sum_{i \in I} \mu_i = \int_{\mathcal{X}} \sum_{i \in I} \mu_i e_i^2(x) d\mu = \int_{\mathcal{X}} k(x, x) d\mu.$$

The following Hilbert-Schmidt integral operator is useful in our analysis:

$$T_k : L_2(\mu) \to L_2(\mu), \quad T_k f = \int_{\mathcal{X}} k(x, x') f(x') d\mu.$$

Obviously, $T_k$ is self-adjoint. Use the fact that $k(x, x') = \sum_{i \in I} \mu_i e_i(x) e_i(x')$, we know $T_k e_i = \mu_i e_i$, which means $e_i$ is the eigenfunction of $T_k$ with the corresponding eigenvalue as $\mu_i$.

With the spectral characterization of $T_k$, we can define the power operator $T_k$, by

$$T_k^\tau f : L_2(\mu) \to L_2(\mu), \quad T_k^\tau f = \sum_{i \in I} \mu_i^\tau \langle f, e_i \rangle e_i.$$

And these power operators are all self-adjoint. Note that, $\|f\|_{\mathcal{H}_k} = \langle f, T_k^{-1} f \rangle_{L_2(\mu)}$. Throughout the paper, we work on the $L_2(\mu)$ space, and all of the operators are defined on $L_2(\mu) \to L_2(\mu)$. As $\mathcal{H}_k \subset L_2(\mu)$, all of these operators can also operator on the elements from $\mathcal{H}_k$.

The power RKHS induced by the following kernel will be helpful in our analysis:

$$\forall x, x' \in \mathcal{X}, \widetilde{k}(x, x') = \sum_{i \in I} \mu_i^2 e_i(x) e_i(x').$$

And it is straightforward to see $\|f\|_{H_{\widetilde{k}}} = \langle f, T_k^{-2} f \rangle_{L_2(\mu)}$, which will be useful in the proof.

**Definition 13** (Kernel with Random Feature Representation). *A kernel $k : \mathcal{X} \times \mathcal{X} \to \mathbb{R}$ is said to have a random feature representation if there exists a function $\psi : \mathcal{X} \times \Xi \to \mathbb{R}$ and a probability measure $P$ over $\Xi$ such that*

$$k(x, x') = \int_{\Xi} \psi(x; \xi) \psi(x'; \xi) dP(\xi).$$

*We then show that, $\mathcal{H}_k$ coincides with the following $\mathbb{R}$-valued function space*

$$\left\{ f : \mathcal{X} \to \mathbb{R} \,\middle|\, f(x) = \int_{\Xi} \widetilde{f}(\xi) \psi(x; \xi) dP(\xi), \widetilde{f} \in L_2(P) \right\},$$

*with the inner product defined as $\langle f, g \rangle_{\mathcal{H}_k} = \int_{\Xi} \widetilde{f}(\xi) \widetilde{g}(\xi) dP(\xi)$. Note that, $\widetilde{k}(x, \cdot) = \psi(x; \xi)$. Hence, it is straightforward to show that $\forall x \in \mathcal{X}, k(x, \cdot) \in \mathcal{H}_k$. Furthermore, we have*

$$f(x) = \int_{\Xi} \widetilde{f}(\xi) \psi(x; \xi) dP(\xi) = \int_{\Xi} \widetilde{f}(\xi) \widetilde{k}(x, \cdot) dP(\xi) = \langle f, k(x, \cdot) \rangle_{\mathcal{H}_k},$$

*which shows the reproducing property. As a result, we obtain an equivalent representation of the RKHS $\mathcal{H}_k$, which means $\forall f \in \mathcal{H}_k$, we can obtain a random feature representation.*

Examples of such kernel $k$ includes the Gaussian kernel and the Matérn kernel. See Rahimi & Recht (2007); Dai et al. (2014); Choromanski et al. (2018) for the details.

**Definition 14** ($\varepsilon$-net and $\varepsilon$-covering number and $i$-th (dyadic) entropy number (Steinwart & Christmann, 2008)). *Let $(T, d)$ be a metric space. $S \subset T$ is an $\varepsilon$-net for $\varepsilon > 0$, if $\forall t \in T$, $\exists s \in S$, such that $d(s, t) \leq \varepsilon$. Furthermore, the $\varepsilon$-covering number $\mathcal{N}(T, d, \varepsilon)$ is defined as as the minimum cardinality of the $\varepsilon$-net for $T$ under metric $d$, and the $i$-th entropy number $e_i(T, d)$ is the minimum $\varepsilon$ that there exists an $\varepsilon$ cover of cardinality $2^{i-1}$.*

## E MAIN PROOF

### E.1 TECHNICAL CONDITIONS

**Assumption 3** (Regularity Conditions for Kernel). *$\mathcal{Z}$ is a compact metric space with the Lebesgue measure $\mu$ if $\mathcal{Z}$ is continuous, and $\int_{\mathcal{Z}} k(z, z)d\mu \leq 1$.*

*Remark* 7. Assumption 3 is mainly for the ease of presentation. The assumption that $\mathcal{Z}$ is compact when $\mathcal{Z}$ is continuous can be relaxed to $\mathcal{Z}$ is a general domain but requires much more involved techniques from e.g. Steinwart & Scovel (2012). Furthermore, with Mercer's theorem (see Theorem 3 and Remark 6 for the details), we know $\sum_{i \in I} \mu_i = \int_{\mathcal{Z}} k(z, z)d\mu \leq 1$. As $\forall i \in I, \mu_i > 0$, we know $\mu_1 \leq 1$, and $\|f\|_{L_2(\mu)} \leq \|f\|_{\mathcal{H}_k}$ without any other absolute constant, that can keep the eventual result clean. We can relax the assumption $\int_{\mathcal{Z}} k(z, z)d\mu \leq 1$ to $\int_{\mathcal{Z}} k(z, z)d\mu \leq c$ for some positive constant $c > 1$, at the cost of additional terms at most $\text{poly}(c)$ in the sample complexity.

**Assumption 4** (Eigendecay Conditions for Kernel). *For the reproducing kernel $k$, we assume $\mu_i$, the $i$-th eigenvalue of the operator $T_k : L_2(\mu) \to L_2(\mu)$, $T_k f = \int_{\mathcal{Z}} f(z')k(z, z')d\mu(z')$, satisfies one of the following conditions:*

- *$\beta$-finite spectrum: $\mu_i = 0$, $\forall i > \beta$, where $\beta$ is a positive integer.*
- *$\beta$-polynomial decay: $\mu_i \leq C_0 i^{-\beta}$, where $C_0$ is an absolute constant and $\beta > 1$.*
- *$\beta$-exponential decay: $\mu_i \leq C_1 \exp(-C_2 i^\beta)$, where $C_1$ and $C_2$ are absolute constants and $\beta > 0$.*

*For ease of presentation, we use $C_{\text{poly}}$ to denote constants appeared in the analysis of $\beta$-polynomial decay that only depends on $C_0$ and $\beta$, and $C_{\text{exp}}$ to denote constants appeared in the analysis of $\beta$-exponential decay that only depends on $C_1$, $C_2$ and $\beta$. Both of them can be varied step by step.*

*Remark* 8. We remark that, most of the existing kernels satisfy one of these eigendecay conditions. Specifically, as discussed in Seeger et al. (2008); Yang et al. (2020), the linear kernel and the polynomial kernel satisfy the $\beta$-finite spectrum condition, the Matérn kernel satisfies the $\beta$-polynomial decay and the Gaussian kernel satisfies the $\beta$-exponential decay. Furthermore, for discrete $\mathcal{Z}$, we can directly observe that it corresponds to the case of $\beta$-finite spectrum with $\beta \leq |\mathcal{Z}|$.

### E.2 PROOF FOR THE ONLINE SETTING

**Theorem 4** (PAC Guarantee for Online Exploration, Formal). *If we choose the bonus $\hat{b}_n(s, a)$ as:*

$$\hat{b}_n(s, a) = \min \left\{ \alpha_n \|\hat{p}_n(\cdot|s, a)\|_{L_2(\mu), \hat{\Sigma}_{n, \hat{p}_n}^{-1}}, 2 \right\},$$

*where*

$$\hat{\Sigma}_{n, \hat{p}_n} : L_2(\mu) \to L_2(\mu), \quad \hat{\Sigma}_{n, \hat{p}_n} := \sum_{(s_i, a_i) \in \mathcal{D}_n} \left[ \hat{p}_n(z|s_i, a_i)\hat{p}_n(z|s_i, a_i)^\top \right] + \lambda T_k^{-1},$$

*$\|f\|_{L_2(\mu), \Sigma} := \sqrt{\langle f, \Sigma f \rangle_{L_2(\mu)}}$ for self-adjoint operator $\Sigma$, $\lambda$ for different eigendecay conditions is given by:*

- *$\beta$-finite spectrum: $\lambda = \Theta(\beta \log N + \log(N|\mathcal{P}|/\delta))$*
- *$\beta$-polynomial decay: $\lambda = \Theta(C_{\text{poly}} N^{1/(1+\beta)} + \log(N|\mathcal{P}|/\delta))$;*

- $\beta$-exponential decay: $\lambda = \Theta(C_{\exp}(\log N)^{1/\beta} + \log(N|\mathcal{P}|/\delta))$;

and $\alpha_n = \Theta\left(\frac{\gamma}{1-\gamma}\sqrt{|\mathcal{A}|\log(n|\mathcal{P}|/\delta) + \lambda C}\right)$, then with probability at least $1 - \delta$, After interacting with the environments for $N$ episodes where

- $N = \Theta\left(\frac{C\beta^3|\mathcal{A}|^2\log(|\mathcal{P}|/\delta)}{(1-\gamma)^4\varepsilon^2}\log^3\left(\frac{C\beta^3|\mathcal{A}|^2\log(|\mathcal{P}|/\delta)}{(1-\gamma)^4\varepsilon^2}\right)\right)$ for $\beta$-finite spectrum;

- $N = \Theta\left(C_{\mathrm{poly}}\left(\frac{|\mathcal{A}|\sqrt{C\log(|\mathcal{P}|/\delta)}}{(1-\gamma)^2\varepsilon}\log^{3/2}\left(\frac{\sqrt{C}|\mathcal{A}|\log(|\mathcal{P}|/\delta)}{(1-\gamma)^2\varepsilon}\right)\right)^{\frac{2(1+\beta)}{\beta-1}}\right)$ for $\beta$-polynomial decay;

- $N = \Theta\left(\frac{C_{\exp}C|\mathcal{A}|^2\log(|\mathcal{P}|/\delta)}{(1-\gamma)^4\varepsilon^2}\log^{\frac{3+2\beta}{\beta}}\left(\frac{C|\mathcal{A}|^2\log(|\mathcal{P}|/\delta)}{(1-\gamma)^4\varepsilon^2}\right)\right)$ for $\beta$-exponential decay;

we can obtain an $\varepsilon$-optimal policy with probability at least $1 - \delta$.

**Notation** Following the notation of Uehara et al. (2022), we define

$$\rho_n(s) := \frac{1}{n}\sum_{i=1}^{n-1} d^{\pi_i}_{T^*}(s),$$

and with slight abuse of notation, we overload the above notation and define

$$\rho_n(s,a) := \frac{1}{n}\sum_{i=1}^{n-1} d^{\pi_i}_{T^*}(s,a).$$

Furthermore, we define $\rho'_n(s')$ as the marginal distribution of $s'$ for the following joint distribution
$$(s, a, s') \sim \rho_n(s) \times \mathcal{U}(\mathcal{A}) \times T^*(s'|s,a).$$
Finally, we define the following operators in the space of $L_2(\mu) \to L_2(\mu)$:

$$\Sigma_{\rho_n \times \mathcal{U}(\mathcal{A}),\phi} = n\mathbb{E}_{s\sim\rho_n, a\sim\mathcal{U}(\mathcal{A})}\left[\phi(s,a)\phi^\top(s,a)\right] + \lambda T_k^{-1}$$
$$\Sigma_{\rho_n,\phi} = n\mathbb{E}_{(s,a)\sim\rho_n}\left[\phi(s,a)\phi^\top(s,a)\right] + \lambda T_k^{-1}$$

Note that, by the spectral theorem, if $\left\|T^{-1/2}x'\right\|_{L_2(\mu)} \leq \infty$ for $x' \in L_2(\mu)$, we have the following Cauchy-Schwartz inequality for weighted $L_2(\mu)$ norm: $\forall x \in L_2(\mu)$,

$$\langle x, x'\rangle_{L_2(\mu)} = \left\langle T^{1/2}x, T^{-1/2}x'\right\rangle_{L_2(\mu)} \leq \|x\|_{L_2(\mu),T}\|x'\|_{L_2(\mu),T^{-1}}.$$

**Lemma 5** (One Step Back Inequality for the Learned Model). *Assume $g : \mathcal{S} \times \mathcal{A} \to \mathbb{R}$ such that $\|g\|_\infty \leq B$, then conditioning on the event that the following MLE generalization bound holds:*

$$\mathbb{E}_{s\sim\rho_n, a\sim\mathcal{U}(\mathcal{A})}\left[\|\hat{T}(s,a) - T^*(s,a)\|_1\right] \leq \zeta_n,$$

$\forall \pi$, we have

$$\left|\mathbb{E}_{(s,a)\sim d^\pi_{\hat{T}_n}}[g(s,a)]\right|$$
$$\leq \gamma\mathbb{E}_{(\widetilde{s},\widetilde{a})\sim d^\pi_{\hat{T}_n}}\|\hat{p}_n(\cdot|\widetilde{s},\widetilde{a})\|_{L_2(\mu),\Sigma^{-1}_{\rho_n \times \mathcal{U}(\mathcal{A}),\hat{p}_n}}\sqrt{n|\mathcal{A}|\mathbb{E}_{s\sim\rho'_n, a\sim\mathcal{U}(\mathcal{A})}[g^2(s,a)] + \lambda B^2 C + nB^2\zeta_n}$$
$$+ \sqrt{(1-\gamma)|\mathcal{A}|\mathbb{E}_{s\sim\rho_n, a\sim\mathcal{U}(\mathcal{A})}[g^2(s,a)]}.$$

*Proof.* We start from the following equality:

$$\mathbb{E}_{(s,a)\sim d^\pi_{\hat{T}_n}}[g(s,a)] = \gamma\mathbb{E}_{(\widetilde{s},\widetilde{a})\sim d^\pi_{\hat{T}_n}, s\sim\hat{T}_n(\cdot|\widetilde{s},\widetilde{a}), a\sim\pi(\cdot|s)}[g(s,a)] + (1-\gamma)\mathbb{E}_{s\sim d_0, a\sim\pi(\cdot|s)}[g(s,a)],$$
(9)

which is obtained by the property of the stationary distribution. For the second term, with Jensen's inequality and an importance sampling step, we have that

$$(1-\gamma)\mathbb{E}_{s\sim d_0, a\sim\pi(\cdot|s)}[g(s,a)] \leq \sqrt{(1-\gamma)|\mathcal{A}|\mathbb{E}_{s\sim\rho_n, a\sim\mathcal{U}(\mathcal{A})}[g^2(s,a)]}.$$

Now we consider the first term. With Cauchy-Schwartz inequality of $L_2(\mu)$ inner product, we have that

$$\gamma \mathbb{E}_{(\widetilde{s},\widetilde{a})\sim d^\pi_{\hat{T}_n}, s\sim \hat{T}_n(\cdot|\widetilde{s},\widetilde{a}), a\sim\pi(\cdot|s)}[g(s,a)]$$

$$=\gamma \mathbb{E}_{(\widetilde{s},\widetilde{a})\sim d^\pi_{\hat{T}_n}} \left\langle \hat{p}_n(\cdot|\widetilde{s},\widetilde{a}), \int_\mathcal{S} \sum_{a\in\mathcal{A}} \hat{p}_n(s|\cdot)\pi(a|s)g(s,a)ds \right\rangle_{L_2(\mu)}$$

$$\leq \gamma \mathbb{E}_{(\widetilde{s},\widetilde{a})\sim d^\pi_{\hat{T}_n}} \|\hat{p}_n(\cdot|\widetilde{s},\widetilde{a})\|_{L_2(\mu),\Sigma^{-1}_{\rho_n\times\mathcal{U}(\mathcal{A}),\hat{p}_n}} \left\| \int_\mathcal{S} \sum_{a\in\mathcal{A}} \hat{p}_n(s|\cdot)\pi(a|s)g(s,a)ds \right\|_{L_2(\mu),\Sigma_{\rho_n\times\mathcal{U}(\mathcal{A}),\hat{p}_n}}$$

Note that

$$\left\| \int_\mathcal{S} \sum_{a\in\mathcal{A}} \hat{p}_n(s|\cdot)\pi(a|s)g(s,a)ds \right\|^2_{L_2(\mu),\Sigma_{\rho_n\times\mathcal{U}(\mathcal{A}),\hat{p}_n}}$$

$$=n\mathbb{E}_{\widetilde{s}\sim\rho_n,\widetilde{a}\sim\mathcal{U}(\mathcal{A})} \left\{ \mathbb{E}_{s\sim\hat{T}_n(\cdot|\widetilde{s},\widetilde{a}), a\sim\pi(\cdot|s)}[g(s,a)] \right\}^2 + \lambda \left\| \int_\mathcal{S} \sum_{a\in\mathcal{A}} \hat{p}_n(s|\cdot)\pi(a|s)g(s,a)ds \right\|_{\mathcal{H}_k}$$

$$\leq n\mathbb{E}_{\widetilde{s}\sim\rho_n,\widetilde{a}\sim\mathcal{U}(\mathcal{A})} \left\{ \mathbb{E}_{s\sim T^*(\cdot|\widetilde{s},\widetilde{a}), a\sim\pi(\cdot|s)}[g(s,a)] \right\}^2 + \lambda B^2 C + nB^2\zeta_n$$

$$\leq n\mathbb{E}_{\widetilde{s}\sim\rho_n,\widetilde{a}\sim\mathcal{U}(\mathcal{A}), s\sim T^*(\cdot|\widetilde{s},\widetilde{a}), a\sim\pi(\cdot|s)}\left[g^2(s,a)\right] + \lambda B^2 C + nB^2\zeta_n$$

$$\leq n|\mathcal{A}|\mathbb{E}_{\widetilde{s}\sim\rho_n,\widetilde{a}\sim\mathcal{U}(\mathcal{A}), s\sim T^*(\cdot|\widetilde{s},\widetilde{a}), a\sim\mathcal{U}(\mathcal{A})}\left[g^2(s,a)\right] + \lambda B^2 C + nB^2\zeta_n$$

$$= n|\mathcal{A}|\mathbb{E}_{s\sim\rho'_n, a\sim\mathcal{U}(\mathcal{A})}\left[g^2(s,a)\right] + \lambda B^2 C + nB^2\zeta_n$$

Substitute back, we obtain the desired result. $\qquad\square$

**Lemma 6** (One Step Back Inequality for the True Model). *Assume $g : \mathcal{S}\times\mathcal{A} \to \mathbb{R}$ such that $\|g\|_\infty \leq B$, then*

$$\mathbb{E}_{(s,a)\sim d^\pi_{T^*}}[g(s,a)] \leq \mathbb{E}_{(\widetilde{s},\widetilde{a})\sim d^\pi_{T^*}} \|p^*(\cdot|\widetilde{s},\widetilde{a})\|_{L_2(\mu),\Sigma^{-1}_{\rho_n,p^*}} \sqrt{n\gamma|\mathcal{A}|\mathbb{E}_{s\sim\rho_n,a\sim\mathcal{U}(\mathcal{A})}\left[g^2(s,a)\right] + \lambda\gamma^2 B^2 C}$$

$$+ \sqrt{(1-\gamma)|\mathcal{A}|\mathbb{E}_{s\sim\rho_n,a\sim\mathcal{U}(\mathcal{A})}[g^2(s,a)]}.$$

*Proof.* By the property of the stationary distribution, we have

$$\mathbb{E}_{(s,a)\sim d^\pi_{T^*}}[g(s,a)] = \gamma\mathbb{E}_{(\widetilde{s},\widetilde{a})\sim d^\pi_{T^*}, s\sim T^*(\cdot|\widetilde{s},\widetilde{a}), a\sim\pi(\cdot|s)}[g(s,a)] + (1-\gamma)\mathbb{E}_{s\sim d_0, a\sim\pi(\cdot|s)}[g(s,a)]. \tag{10}$$

For the second term, we still use the following upper bound:

$$(1-\gamma)\mathbb{E}_{s\sim d_0, a\sim\pi(\cdot|s)}[g(s,a)] \leq \sqrt{(1-\gamma)|\mathcal{A}|\mathbb{E}_{s\sim\rho_n,a\sim\mathcal{U}(\mathcal{A})}[g^2(s,a)]}.$$

For the first term, with the Cauchy-Schwartz inequality, we have

$$\gamma\mathbb{E}_{(\widetilde{s},\widetilde{a})\sim d^\pi_{T^*}, s\sim T^*(\cdot|\widetilde{s},\widetilde{a}), a\sim\pi(\cdot|s)}[g(s,a)]$$

$$=\gamma\mathbb{E}_{(\widetilde{s},\widetilde{a})\sim d^\pi_{T^*}} \left\langle p^*(\cdot|\widetilde{s},\widetilde{a}), \int_\mathcal{S} \sum_{a\in\mathcal{A}} p^*(s|\cdot)\pi(a|s)g(s,a)ds \right\rangle_{L_2(\mu)}$$

$$\leq\gamma\mathbb{E}_{(\widetilde{s},\widetilde{a})\sim d^\pi_{T^*}} \|p^*(\cdot|\widetilde{s},\widetilde{a})\|_{L_2(\mu),\Sigma^{-1}_{\rho_n,p^*}} \left\| \int_\mathcal{S} \sum_{a\in\mathcal{A}} p^*(s|\cdot)\pi(a|s)g(s,a)ds \right\|_{L_2(\mu),\Sigma^2_{\rho_n,p^*}}.$$

Note that

$$\left\| \int_\mathcal{S} \sum_{a\in\mathcal{A}} p^*(s|\cdot)\pi(a|s)g(s,a)ds \right\|^2_{L_2(\mu),\Sigma^2_{\rho_n,p^*}}$$

$$=n\mathbb{E}_{(\widetilde{s},\widetilde{a})\sim\rho_n} \left\{ \mathbb{E}_{s\sim T^*(\cdot|\widetilde{s},\widetilde{a}), a\sim\pi(\cdot|s)}[g(s,a)] \right\}^2 + \lambda \left\| \int_\mathcal{S} \sum_{a\in\mathcal{A}} \hat{p}_n(s|\cdot)\pi(a|s)g(s,a)ds \right\|_{\mathcal{H}_k}$$

$$\leq n\mathbb{E}_{(\widetilde{s},\widetilde{a})\sim\rho_n, s\sim T^*(\cdot|\widetilde{s},\widetilde{a}), a\sim\pi(\cdot|s)}\left[g^2(s,a)\right] + \lambda B^2 C$$

$$\leq n|\mathcal{A}|\mathbb{E}_{(\widetilde{s},\widetilde{a})\sim\rho_n, s\sim T^*(\cdot|\widetilde{s},\widetilde{a}), a\sim\mathcal{U}(\mathcal{A})}\left[g^2(s,a)\right] + \lambda B^2 C$$

$$=\frac{n|\mathcal{A}|}{\gamma}\mathbb{E}_{s\sim\rho_n,a\sim\mathcal{U}(\mathcal{A})}\left[g^2(s,a)\right]+\lambda B^2 C$$

Substitute back, we obtain the desired result. $\qquad\square$

**Lemma 7** (Almost Optimism at the Initial Distribution). *Consider an episode $n\in[N]$, if we set $\zeta_n=\Theta\left(\frac{\log(n|\mathcal{P}|/\delta)}{n}\right)$ (such that the MLE generalization bound holds by Lemma 16), $\lambda$ for different eigendecay condition as follows:*

- *$\beta$-finite spectrum: $\lambda=\Theta(\beta\log N+\log(N|\mathcal{P}|/\delta))$*

- *$\beta$-polynomial decay: $\lambda=\Theta(C_{\mathrm{poly}}N^{1/(1+\beta)}+\log(N|\mathcal{P}|/\delta))$;*

- *$\beta$-exponential decay: $\lambda=\Theta(C_{\exp}(\log N)^{1/\beta}+\log(N|\mathcal{P}|/\delta))$;*

*and $\alpha_n=\Theta\left(\frac{\gamma}{1-\gamma}\sqrt{|\mathcal{A}|\log(n|\mathcal{P}|/\delta)+\lambda C}\right)$, the following events hold with probability at least $1-\delta$:*

$$\forall n\in[N],\quad\forall\pi,\quad V_{\hat{T}_n,r+\hat{b}_n}^{\pi}-V_{T^*,r}^{\pi}\geq-\sqrt{\frac{|\mathcal{A}|\zeta_n}{(1-\gamma)^3}}.$$

*Proof.* With Lemma 17 and a union bound over $\mathcal{P}$, we know using the chosen $\lambda$, $\forall\hat{T}_n\in\mathcal{P}$, with probability at least $1-\delta$,

$$\|\hat{p}_n(\cdot|s,a)\|_{L_2(\mu),\hat{\Sigma}_{n,\hat{p}_n}^{-1}}=\Theta\left(\|\hat{p}_n(\cdot|s,a)\|_{L_2(\mu),\Sigma_{\rho_n\times\mathcal{U}(\mathcal{A}),\hat{p}_n}}^{-1}\right).$$

With Lemma 18, we have that

$$(1-\gamma)\left(V_{\hat{T}_n,r+\hat{b}_n}^{\pi}-V_{T^*,r}^{\pi}\right)$$

$$=\mathbb{E}_{(s,a)\sim d_{\hat{T}_n}^{\pi}}\left[b_n(s,a)+\gamma\mathbb{E}_{\hat{T}_n(s'|s,a)}\left[V_{T,r}^{\pi}(s')\right]-\gamma\mathbb{E}_{P(s'|s,a)}\left[V_{T,r}^{\pi}(s')\right]\right]$$

$$\gtrsim\mathbb{E}_{(s,a)\sim d_{\hat{T}_n}^{\pi}}\left[\min\left\{\alpha_n\|\hat{p}_n(\cdot|s,a)\|_{L_2(\mu),\Sigma_{\rho_n\times\mathcal{U}(\mathcal{A}),\hat{p}_n}^{-1}},2\right\}+\gamma\mathbb{E}_{\hat{T}_n(s'|s,a)}\left[V_{T^*,r}^{\pi}(s')\right]-\gamma\mathbb{E}_{T^*(s'|s,a)}\left[V_{T^*,r}^{\pi}(s')\right]\right].$$

Denote $f_n(s,a)=\mathrm{TV}(T^*(s'|s,a),\hat{T}_n(s'|s,a))$ with $\|f_n\|_\infty\leq2$, with Hölder's inequality, we have that

$$\left|\mathbb{E}_{(s,a)\sim d_{\hat{T}_n}^{\pi}}\left[\mathbb{E}_{\hat{T}_n(s'|s,a)}\left[V_{T,r}^{\pi}(s')\right]-\mathbb{E}_{T^*(s'|s,a)}\left[V_{T,r}^{\pi}(s')\right]\right]\right|\leq\mathbb{E}_{(s,a)\sim d_{\hat{T}_n}^{\pi}}\left[\frac{f_n(s,a)}{1-\gamma}\right].$$

With Lemma 5, we have that

$$\mathbb{E}_{(s,a)\sim d_{\hat{T}_n}^{\pi}}\left[\frac{f_n(s,a)}{1-\gamma}\right]$$

$$\leq\mathbb{E}_{(\tilde{s},\tilde{a})\sim d_{\hat{T}_n}^{\pi}}\|\hat{p}_n(\cdot|s,a)\|_{L_2(\mu),\Sigma_{\rho_n\times\mathcal{U}(\mathcal{A}),\hat{p}_n}^{-1}}\sqrt{\frac{n\gamma^2|\mathcal{A}|\mathbb{E}_{s\sim\rho_n',a\sim\mathcal{U}(\mathcal{A})}\left[f_n^2(s,a)\right]}{(1-\gamma)^2}+\frac{4\lambda\gamma^2 C}{(1-\gamma)^2}+\frac{4n\gamma^2\zeta_n}{(1-\gamma)^2}}$$

$$+\sqrt{\frac{|\mathcal{A}|\mathbb{E}_{s\sim\rho_n,a\sim\mathcal{U}(\mathcal{A})}\left[f_n^2(s,a)\right]}{1-\gamma}}$$

$$\leq\mathbb{E}_{(\tilde{s},\tilde{a})\sim d_{\hat{T}_n}^{\pi}}\|\hat{p}_n(\cdot|s,a)\|_{L_2(\mu),\Sigma_{\rho_n\times\mathcal{U}(\mathcal{A}),\hat{p}_n}^{-1}}\sqrt{\frac{n\gamma^2|\mathcal{A}|\zeta_n}{(1-\gamma)^2}+\frac{4\lambda\gamma^2 C}{(1-\gamma)^2}+\frac{4\gamma^2 n\zeta_n}{(1-\gamma)^2}}+\sqrt{\frac{|\mathcal{A}|\zeta_n}{1-\gamma}}$$

Note that, we set $\alpha_n$ such that

$$\sqrt{\frac{n\gamma^2|\mathcal{A}|\zeta_n}{(1-\gamma)^2}+\frac{4\lambda\gamma^2 C}{(1-\gamma)^2}+\frac{4n\gamma^2\zeta_n}{(1-\gamma)^2}}\lesssim\alpha_n,$$

which concludes the proof. $\qquad\square$

**Lemma 8** (Regret). *With probability at least $1-\delta$, we have that*

- *For $\beta$-finite spectrum, we have*
$$\sum_{n=1}^{N} V_{T^*,r}^{\pi^*} - V_{T^*}^{\pi_n} \lesssim \frac{\beta^{3/2}|\mathcal{A}|\sqrt{CN}\log(N|\mathcal{P}|/\delta)}{(1-\gamma)^2}.$$

- *For $\beta$-polynomial decay, we have*
$$\sum_{n=1}^{N} V_{T^*,r}^{\pi^*} - V_{T^*}^{\pi_n} \lesssim \frac{C_{\mathrm{poly}}\sqrt{C}|\mathcal{A}|N^{\frac{1}{2}+\frac{1}{2(1+\beta)}}\log(N|\mathcal{P}|/\delta)}{(1-\gamma)^2}.$$

- *For $\beta$-exponential decay, we have*
$$\sum_{n=1}^{N} V_{T^*,r}^{\pi^*} - V_{T^*}^{\pi_n} \lesssim \frac{C_{\exp}|\mathcal{A}|\sqrt{CN}(\log N)^{\frac{3+2\beta}{2\beta}}\log(N|\mathcal{P}|/\delta)}{(1-\gamma)^2}.$$

*Proof.* With Lemma 7 and Lemma 18, we have that

$$V_{T^*,r}^{\pi^*} - V_{T^*,r}^{\pi_n}$$

$$\leq V_{\hat{T}_n,r+b_n}^{\pi^*} + \sqrt{\frac{|\mathcal{A}|\zeta_n}{(1-\gamma)^3}} - V_{T^*,r}^{\pi_n}$$

$$\leq V_{\hat{T}_n,r+b_n}^{\pi_n} + \sqrt{\frac{|\mathcal{A}|\zeta_n}{(1-\gamma)^3}} - V_{T^*,r}^{\pi_n}$$

$$\leq \frac{1}{1-\gamma}\mathbb{E}_{(s,a)\sim d_{T^*}^{\pi_n}}\left[b_n(s,a) + \gamma\mathbb{E}_{\hat{T}_n(s'|s,a)}[V_{\hat{T}_n,r+b_n}^{\pi_n}(s')] - \gamma\mathbb{E}_{T^*(s'|s,a)}[V_{\hat{T}_n,r+b_n}^{\pi_n}(s')]\right] + \sqrt{\frac{|\mathcal{A}|\zeta_n}{(1-\gamma)^3}}.$$

Applying Lemma 6 and note that $b_n = O(1)$, we have that

$$\mathbb{E}_{(s,a)\sim d_{T^*}^{\pi_n}}[b_n(s,a)]$$

$$\lesssim \mathbb{E}_{(s,a)\sim d_{T^*}^{\pi_n}}\left[\min\left\{\alpha_n\|\hat{p}_n(\cdot|s,a)\|_{L_2(\mu),\Sigma_{\rho_n\times\mathcal{U}(\mathcal{A}),\hat{p}_n}^{-1}}, 2\right\}\right]$$

$$\lesssim \mathbb{E}_{(\tilde{s},\tilde{a})\sim d_{T^*}^{\pi_n}}\|p^*(\cdot|\tilde{s},\tilde{a})\|_{\Sigma_{\rho_n,p^*}^{-1}}\sqrt{n\gamma|\mathcal{A}|\alpha_n^2\mathbb{E}_{s\sim\rho_n,a\sim\mathcal{U}(\mathcal{A})}\left[\|\hat{p}_n(\cdot|s,a)\|_{L_2(\mu),\Sigma_{\rho_n\times\mathcal{U}(\mathcal{A}),\hat{p}_n}^{-1}}^2\right]} + \lambda\gamma^2 C$$

$$+ \sqrt{(1-\gamma)|\mathcal{A}|\alpha_n^2\mathbb{E}_{s\sim\rho_n,a\sim\mathcal{U}(\mathcal{A})}\left[\|\hat{p}_n(\cdot|s,a)\|_{L_2(\mu),\Sigma_{\rho_n\times\mathcal{U}(\mathcal{A}),\hat{p}_n}^{-1}}^2\right]}.$$

Note that,

$$n\mathbb{E}_{s\sim\rho_n,a\sim\mathcal{U}(\mathcal{A})}\left[\|\hat{p}_n(\cdot|s,a)\|_{L_2(\mu),\Sigma_{\rho_n\times\mathcal{U}(\mathcal{A}),\hat{p}_n}^{-1}}^2\right]$$

$$= \mathrm{Tr}\left(n\mathbb{E}_{s\sim\rho_n,a\sim\mathcal{U}(\mathcal{A})}\left[\hat{p}_n(\cdot|s,a)\hat{p}_n(\cdot|s,a)^\top\right]\left(n\mathbb{E}_{s\sim\rho_n,a\sim\mathcal{U}(\mathcal{A})}\left[\hat{p}_n(\cdot|s,a)\hat{p}_n(\cdot|s,a)^\top\right] + \lambda T_k^{-1}\right)^{-1}\right)$$

$$= \mathrm{Tr}\left(n\mathbb{E}_{s\sim\rho_n,a\sim\mathcal{U}(\mathcal{A})}\left[T_k^{1/2}\hat{p}_n(\cdot|s,a)\hat{p}_n(\cdot|s,a)^\top T_k^{1/2}\right]\left(n\mathbb{E}_{s\sim\rho_n,a\sim\mathcal{U}(\mathcal{A})}\left[T_k^{1/2}\hat{p}_n(\cdot|s,a)\hat{p}_n(\cdot|s,a)^\top T_k^{1/2}\right] + \lambda I\right)^{-1}\right)$$

$$\leq \log\det\left(I + \frac{n}{\lambda}\mathbb{E}_{s\sim\rho_n,a\sim\mathcal{U}(\mathcal{A})}\left[T_k^{1/2}\hat{p}_n(\cdot|s,a)^\top\hat{p}_n(\cdot|s,a)T_k^{1/2}\right]\right),$$

where the first equality is due to the definition of Hilbert-Schmidt inner product and the expectation operator is a linear operator, the second equality is due to the fact that $\mathrm{Tr}(A(A+B)^{-1}) = \mathrm{Tr}\left((B^{-1/2}AB^{-1/2})(I + B^{-1/2}AB^{-1/2})^{-1}\right)$ for positive semi-definite operator $A$ and positive definite operator $B$, and the last inequality is due to the fact that if $A$ has the eigensystem $\{\mu_i, e_i\}$, then $A(A+\lambda I)^{-1}$ has the eigensystem $\{\frac{\mu_i}{\mu_i+\lambda}, e_i\}$, and $\frac{x}{1+x} \leq \log(1+x)$. Here $\det$ denotes the Fredholm determinant. Note that, if $x \in \mathcal{B}_{\mathcal{H}_k}$, $T_k^{1/2}x \in \mathcal{B}_{\mathcal{H}_{\tilde{k}}}$, and we know $\mathbb{E}_{s\sim\rho_n,a\sim\mathcal{U}(\mathcal{A})}\left[T_k^{1/2}\hat{p}_n(\cdot|s,a)^\top\hat{p}_n(\cdot|s,a)T_k^{1/2}\right]$ is in the trace class and the Fredholm determinant is well-defined. Invoking Lemma 19, we have that

- For $\beta$-finite spectrum, as $\lambda = \Theta(\beta\log N + \log(N|\mathcal{P}|/\delta))$, we have $n/\lambda = O(n)$
$$\log\det\left(I + \frac{n}{\lambda}\mathbb{E}_{s\sim\rho_n,a\sim\mathcal{U}(\mathcal{A})}\left[T_k^{1/2}\hat{p}_n(\cdot|s,a)^\top\hat{p}_n(\cdot|s,a)T_k^{1/2}\right]\right) = O(\beta\log n),$$

which means
$$\mathbb{E}_{(s,a)\sim d_{T^*}^{\pi_n}}[b_n(s,a)] \lesssim \sqrt{\gamma|\mathcal{A}|\alpha_n^2\beta\log(n) + \lambda\gamma^2 C} \cdot \mathbb{E}_{(\widetilde{s},\widetilde{a})\sim d_{T^*}^{\pi_n}} \|p^*(\cdot|\widetilde{s},\widetilde{a})\|_{\Sigma_{\rho_n,p^*}^{-1}}$$
$$+ \sqrt{\frac{(1-\gamma)|\mathcal{A}|\alpha_n^2\beta\log(n)}{n}}.$$

- For $\beta$-polynomial decay, as $\lambda = \Theta(C_{\mathrm{poly}}N^{1/(1+\beta)} + \log(N|\mathcal{P}|/\delta))$ and $n \leq N$, we have $n/\lambda = O\left(C_{\mathrm{poly}}n^{\frac{\beta}{1+\beta}}\right)$ and
$$\log\det\left(I + \frac{n}{\lambda}\mathbb{E}_{s\sim\rho_n, a\sim\mathcal{U}(\mathcal{A})}\left[T_k^{1/2}\hat{p}_n(\cdot|s,a)^\top \hat{p}_n(\cdot|s,a)T_k^{1/2}\right]\right) = O\left(C_{\mathrm{poly}}n^{\frac{1}{2(1+\beta)}}\log(n)\right),$$
This leads to
$$\mathbb{E}_{(s,a)\sim d_{T^*}^{\pi_n}}[b_n(s,a)] \lesssim \sqrt{\gamma|\mathcal{A}|C_{\mathrm{poly}}\alpha_n^2 n^{\frac{1}{2(1+\beta)}}\log n + \lambda\gamma^2 C} \cdot \mathbb{E}_{(\widetilde{s},\widetilde{a})\sim d_{T^*}^{\pi_n}} \|p^*(\cdot|\widetilde{s},\widetilde{a})\|_{\Sigma_{\rho_n,p^*}^{-1}}$$
$$+ \sqrt{(1-\gamma)|\mathcal{A}|C_{\mathrm{poly}}n^{-1+\frac{1}{2(1+\beta)}}\log(n)\alpha_n^2}.$$

- For $\beta$-exponential decay, as $\lambda = \Theta\left(C_{\exp}(\log N)^{1/\beta} + \log(N|\mathcal{P}|/\delta)\right)$, we have $n/\lambda = O\left(C_{\exp}n\right)$ and
$$\log\det\left(I + \frac{n}{\lambda}\mathbb{E}_{s\sim\rho_n, a\sim\mathcal{U}(\mathcal{A})}\left[T_k^{1/2}\hat{p}_n(\cdot|s,a)^\top \hat{p}_n(\cdot|s,a)T_k^{1/2}\right]\right) = O\left(C_{\exp}(\log n)^{1+1/\beta}\right),$$
This leads to
$$\mathbb{E}_{(s,a)\sim d_{T^*}^{\pi_n}}[b_n(s,a)] \lesssim \sqrt{\gamma|\mathcal{A}|C_{\exp}(\log n)^{1+1/\beta}\alpha_n^2 + \lambda\gamma^2 C} \cdot \mathbb{E}_{(\widetilde{s},\widetilde{a})\sim d_{T^*}^{\pi_n}} \|p^*(\cdot|\widetilde{s},\widetilde{a})\|_{\Sigma_{\rho_n,p^*}^{-1}}$$
$$+ \sqrt{\frac{(1-\gamma)|\mathcal{A}|C_{\exp}(\log n)^{1+1/\beta}\alpha_n^2}{n}}.$$

For the remaining terms, denote $f_n(s,a) = \mathrm{TV}(T^*(s'|s,a), \hat{T}_n(s')|s,a)$ with $\|f_n\|_\infty \leq 2$. With Hölder's inequality, we have
$$\left|\mathbb{E}_{(s,a)\sim d_{T^*}^{\pi_n}}\left[\mathbb{E}_{\hat{T}_n(s'|s,a)}\left[V_{\hat{T}_n,r+b_n}^{\pi_n}(s')\right] - \mathbb{E}_{T^*(s'|s,a)}\left[V_{\hat{T}_n,r+b_n}^{\pi_n}(s')\right]\right]\right| \lesssim \mathbb{E}_{(s,a)\sim d_{T^*}^{\pi_n}}\left[\frac{f_n(s,a)}{1-\gamma}\right].$$
With Lemma 6, we have that
$$\mathbb{E}_{(s,a)\sim d_{T^*}^{\pi_n}}\left[\frac{f_n(s,a)}{1-\gamma}\right] \leq \mathbb{E}_{(\widetilde{s},\widetilde{a})\sim d_{T^*}^{\pi_n}} \|p^*(\cdot|\widetilde{s},\widetilde{a})\|_{L_2(\mu),\Sigma_{\rho_n,p^*}^{-1}} \sqrt{\frac{n\gamma|\mathcal{A}|\mathbb{E}_{s\sim\rho_n, a\sim\mathcal{U}(\mathcal{A})}[f_n^2(s,a)]}{(1-\gamma)^2} + \frac{4\lambda\gamma^2 C}{(1-\gamma)^2}}$$
$$+ \sqrt{\frac{|\mathcal{A}|\mathbb{E}_{s\sim\rho_n, a\sim\mathcal{U}(\mathcal{A})}[f_n^2(s,a)]}{1-\gamma}}$$
$$\leq \mathbb{E}_{(\widetilde{s},\widetilde{a})\sim d_{T^*}^{\pi}} \|p^*(\cdot|\widetilde{s},\widetilde{a})\|_{L_2(\mu),\Sigma_{\rho_n,p^*}^{-1}} \cdot \sqrt{\frac{n\gamma|\mathcal{A}|\zeta_n}{(1-\gamma)^2} + \frac{4\lambda\gamma^2 C}{(1-\gamma)^2}} + \sqrt{\frac{|\mathcal{A}|\zeta_n}{1-\gamma}}$$
$$\lesssim \alpha_n\mathbb{E}_{(\widetilde{s},\widetilde{a})\sim d_{T^*}^{\pi}} \|p^*(\cdot|\widetilde{s},\widetilde{a})\|_{L_2(\mu),\Sigma_{\rho_n,p^*}^{-1}} + \sqrt{\frac{|\mathcal{A}|\zeta_n}{1-\gamma}}$$

Combine with the previous results and take the dominating terms out, we have that

- For $\beta$-finite spectrum,
$$V_{T^*,r}^{\pi^*} - V_{T^*}^{\pi_n}$$
$$\lesssim \frac{1}{1-\gamma}\sqrt{\gamma|\mathcal{A}|\alpha_n^2\beta\log n + \lambda\gamma^2 C} \cdot \mathbb{E}_{(\widetilde{s},\widetilde{a})\sim d_{T^*}^{\pi_n}} \|p^*(\cdot|\widetilde{s},\widetilde{a})\|_{\Sigma_{\rho_n,p^*}^{-1}}$$
$$+ \sqrt{\frac{|\mathcal{A}|\alpha_n^2\beta\log n}{(1-\gamma)n}} + \sqrt{\frac{|\mathcal{A}|\zeta_n}{(1-\gamma)^3}}.$$

- For $\beta$-polynomial decay,
$$V_{T^*,r}^{\pi^*} - V_{T^*}^{\pi_n}$$

$$\lesssim \frac{1}{1-\gamma}\sqrt{\gamma|\mathcal{A}|C_{\text{poly}}\alpha_n^2 n^{\frac{1}{2(1+\beta)}}\log n + \lambda\gamma^2 C} \cdot \mathbb{E}_{(\widetilde{s},\widetilde{a})\sim d_{T^*}^{\pi_n}}\left\|p^*(\cdot|\widetilde{s},\widetilde{a})\right\|_{\Sigma_{\rho_n,p^*}^{-1}}$$
$$+ \sqrt{\frac{|\mathcal{A}|C_{\text{poly}}\alpha_n^2 n^{-1+\frac{1}{2(1+\beta)}}\log n}{1-\gamma}} + \sqrt{\frac{|\mathcal{A}|\zeta_n}{(1-\gamma)^3}}.$$

- For $\beta$-exponential decay,

$$V_{T^*,r}^{\pi^*} - V_{T^*}^{\pi_n}$$
$$\lesssim \frac{1}{1-\gamma}\sqrt{\gamma|\mathcal{A}|C_{\exp}\alpha_n^2(\log n)^{1+1/\beta} + \lambda\gamma^2 C} \cdot \mathbb{E}_{(\widetilde{s},\widetilde{a})\sim d_{T^*}^{\pi_n}}\left\|p^*(\cdot|\widetilde{s},\widetilde{a})\right\|_{\Sigma_{\rho_n,p^*}^{-1}}$$
$$+ \sqrt{\frac{|\mathcal{A}|C_{\exp}\alpha_n^2(\log n)^{1+1/\beta}}{(1-\gamma)n}} + \sqrt{\frac{|\mathcal{A}|\zeta_n}{(1-\gamma)^3}}.$$

Finally, with Cauchy-Schwartz inequality, we have

$$\sum_{n=1}^{N}\mathbb{E}_{(\widetilde{s},\widetilde{a})\sim d_{T^*}^{\pi_n}}\left\|p^*(\cdot|\widetilde{s},\widetilde{a})\right\|_{L_2(\mu),\Sigma_{\rho_n,p^*}^{-1}} \leq \sqrt{N\sum_{n=1}^{N}\mathbb{E}_{(\widetilde{s},\widetilde{a})\sim d_{T^*}^{\pi_n}}\left\langle p^*(\cdot|\widetilde{s},\widetilde{a}),\Sigma_{\rho_n,p^*}^{-1}p^*(\cdot|\widetilde{s},\widetilde{a})\right\rangle_{L_2(\mu)}}.$$

Note that

$$\mathbb{E}_{(\widetilde{s},\widetilde{a})\sim d_{T^*}^{\pi_n}}\left\langle p^*(\cdot|\widetilde{s},\widetilde{a}),\Sigma_{\rho_n,p^*}^{-1}p^*(\cdot|\widetilde{s},\widetilde{a})\right\rangle_{L_2(\mu)}$$
$$=\mathbb{E}_{(\widetilde{s},\widetilde{a})\sim d_{T^*}^{\pi_n}}\left\langle p^*(\cdot|\widetilde{s},\widetilde{a}),\left(n\mathbb{E}_{(s,a)\sim\rho_n}\left[p^*(\cdot|s,a)p^*(\cdot|s,a)^\top\right] + \lambda T_k^{-1}\right)^{-1}p^*(\cdot|\widetilde{s},\widetilde{a})\right\rangle_{L_2(\mu)}$$
$$=\mathbb{E}_{(\widetilde{s},\widetilde{a})\sim d_{T^*}^{\pi_n}}\left\langle T_k^{1/2}p^*(\cdot|\widetilde{s},\widetilde{a}),\left(n\mathbb{E}_{(s,a)\sim\rho_n}\left[T_k^{1/2}p^*(\cdot|s,a)p^*(\cdot|s,a)^\top T_k^{1/2}\right] + \lambda I\right)^{-1}T_k^{1/2}p^*(\cdot|\widetilde{s},\widetilde{a})\right\rangle_{L_2(\mu)}$$
$$=\text{Tr}\left(\left(\frac{n}{\lambda}\mathbb{E}_{(s,a)\sim\rho_n}\left[T_k^{1/2}p^*(\cdot|s,a)p^*(\cdot|s,a)^\top T_k^{1/2}\right] + I\right)^{-1}, \frac{\mathbb{E}_{(\widetilde{s},\widetilde{a})\sim d_{T^*}^{\pi_n}}\left[T_k^{1/2}p^*(\cdot|\widetilde{s},\widetilde{a})p^*(\cdot|\widetilde{s},\widetilde{a})T_k^{1/2}\right]}{\lambda}\right)$$
$$\leq\log\det\left(\left(\frac{n}{\lambda}\mathbb{E}_{(s,a)\sim\rho_n}\left[T_k^{1/2}p^*(\cdot|s,a)p^*(\cdot|s,a)^\top T_k^{1/2}\right] + I\right)\right)$$
$$- \log\det\left(\left(\frac{n-1}{\lambda}\mathbb{E}_{(s,a)\sim\rho_{n-1}}\left[T_k^{1/2}p^*(\cdot|s,a)p^*(\cdot|s,a)^\top T_k^{1/2}\right] + I\right)\right),$$

where in the last inequality, we use the fact that $\log\det(X)$ is concave with positive definite operators $X$ and $\frac{d\log\det(X)}{dX} = (X^\top)^{-1}$.

Telescoping and applying Lemma 19, we have that:

- For $\beta$-finite spectrum: as $N/\lambda = O(N)$, we have
$$\sum_{n=1}^{N}\mathbb{E}_{(\widetilde{s},\widetilde{a})\sim d_{T^*}^{\pi_n}}\left\langle p^*(\cdot|\widetilde{s},\widetilde{a}),\Sigma_{\rho_n,p^*}^{-1}p^*(\cdot|\widetilde{s},\widetilde{a})\right\rangle_{L_2(\mu)} = O(\beta\log N).$$

- For $\beta$-polynomial decay: as $\lambda = \Theta(C_{\text{poly}}N^{1/(1+\beta)} + \log(N|\mathcal{P}|/\delta))$, we have $N/\lambda = O\left(C_{\text{poly}}N^{\frac{\beta}{1+\beta}}\right)$ and
$$\sum_{n=1}^{N}\mathbb{E}_{(\widetilde{s},\widetilde{a})\sim d_{T^*}^{\pi_n}}\left\langle p^*(\cdot|\widetilde{s},\widetilde{a}),\Sigma_{\rho_n,p^*}^{-1}p^*(\cdot|\widetilde{s},\widetilde{a})\right\rangle_{L_2(\mu)} = O\left(C_{\text{poly}}N^{\frac{1}{2(1+\beta)}}\log N\right).$$

- For $\beta$-exponential decay:
$$\sum_{n=1}^{N}\mathbb{E}_{(\widetilde{s},\widetilde{a})\sim d_{T^*}^{\pi_n}}\left\langle p^*(\cdot|\widetilde{s},\widetilde{a}),\Sigma_{\rho_n,p^*}^{-1}p^*(\cdot|\widetilde{s},\widetilde{a})\right\rangle_{L_2(\mu)} = O\left(C_{\exp}(\log N)^{1+1/\beta}\right).$$

Hence, after we substitute $\alpha_n$ and $\lambda$ back and take the dominating term out, we can conclude that:

- For $\beta$-finite spectrum, we have

$$\sum_{n=1}^{N} V_{T^*,r}^{\pi^*} - V_{T^*}^{\pi_n} \lesssim \frac{\beta^{3/2}|\mathcal{A}| \log N \sqrt{CN \log(N|\mathcal{P}|/\delta)}}{(1-\gamma)^2}.$$

- For $\beta$-polynomial decay, we have

$$\sum_{n=1}^{N} V_{T^*,r}^{\pi^*} - V_{T^*}^{\pi_n} \lesssim \frac{C_{\text{poly}}|\mathcal{A}|N^{\frac{1}{2}+\frac{1}{1+\beta}} \log N \sqrt{C \log(N|\mathcal{P}|/\delta)}}{(1-\gamma)^2}.$$

- For $\beta$-exponential decay, we have

$$\sum_{n=1}^{N} V_{T^*,r}^{\pi^*} - V_{T^*}^{\pi_n} \lesssim \frac{C_{\text{exp}}|\mathcal{A}| \sqrt{CN \log(N|\mathcal{P}|/\delta)} (\log N)^{\frac{3+2\beta}{2\beta}}}{(1-\gamma)^2}.$$

This finishes the proof. $\qquad\square$

**Theorem 9** (PAC Guarantee for Online Setting). *After interacting with the environments for $N$ episodes where*

- $N = \Theta\left( \frac{C\beta^3|\mathcal{A}|^2 \log(|\mathcal{P}|/\delta)}{(1-\gamma)^4 \varepsilon^2} \log^3\left( \frac{C\beta^3|\mathcal{A}|^2 \log(|\mathcal{P}|/\delta)}{(1-\gamma)^4 \varepsilon^2} \right) \right)$ *for $\beta$-finite spectrum;*

- $N = \Theta\left( C_{\text{poly}} \left( \frac{|\mathcal{A}|\sqrt{C \log(|\mathcal{P}|/\delta)}}{(1-\gamma)^2 \varepsilon} \log^{3/2}\left( \frac{\sqrt{C}|\mathcal{A}| \log(|\mathcal{P}|/\delta)}{(1-\gamma)^2 \varepsilon} \right) \right)^{\frac{2(1+\beta)}{\beta-1}} \right)$ *for $\beta$-polynomial*

  *decay;*

- $N = \Theta\left( \frac{C_{\text{exp}}C|\mathcal{A}|^2 \log(|\mathcal{P}|/\delta)}{(1-\gamma)^4 \varepsilon^2} \log^{\frac{3+2\beta}{\beta}}\left( \frac{C|\mathcal{A}|^2 \log(|\mathcal{P}|/\delta)}{(1-\gamma)^4 \varepsilon^2} \right) \right)$ *for $\beta$-exponential decay;*

*we can obtain an $\varepsilon$-optimal policy with high probability.*

*Proof.* Note that, $\log \log x = O(\log x)$. We consider the case with different eigendecay conditions separately.

- For $\beta$-finite spectrum, by taking the output policy as the uniform policy over $\{\pi_i\}_{i\in[n]}$, we can obtain a policy with the sub-optimality gap

$$O\left( \frac{\beta^{3/2}|\mathcal{A}| \log N \sqrt{C \log(N|\mathcal{P}|/\delta)}}{(1-\gamma)^2 \sqrt{N}} \right).$$

  Take $N = \Theta\left( \frac{C\beta^3|\mathcal{A}|^2 \log(|\mathcal{P}|/\delta)}{(1-\gamma)^4 \varepsilon^2} \log^3\left( \frac{C\beta^3|\mathcal{A}|^2 \log(|\mathcal{P}|/\delta)}{(1-\gamma)^4 \varepsilon^2} \right) \right)$, we can see the sub-optimality gap is smaller than $\varepsilon$, which finishes the proof for $\beta$-finite spectrum.

- For $\beta$-polynomial decay, by taking the output policy as the uniform policy over $\{\pi_i\}_{i\in[n]}$, we can obtain a policy with the sub-optimality gap

$$O\left( \frac{C_{\text{poly}}N^{\frac{\beta-1}{2(1+\beta)}} \log N \sqrt{C \log(N|\mathcal{P}|/\delta)}}{(1-\gamma)^2} \right).$$

  Take $N = \Theta\left( C_{\text{poly}} \left( \frac{|\mathcal{A}|\sqrt{C \log(|\mathcal{P}|/\delta)}}{(1-\gamma)^2 \varepsilon} \log^{3/2}\left( \frac{\sqrt{C}|\mathcal{A}| \log(|\mathcal{P}|/\delta)}{(1-\gamma)^2 \varepsilon} \right) \right)^{\frac{2(1+\beta)}{\beta-1}} \right)$, we can see

  the sub-optimality gap is smaller than $\varepsilon$, which finishes the proof for $\beta$-exponential decay.

- For $\beta$-exponential decay, by taking the output policy as the uniform policy over $\{\pi_i\}_{i\in[n]}$, we can obtain a policy with the sub-optimality gap

$$O\left( \frac{C_{\text{exp}}|\mathcal{A}| \sqrt{C \log(N|\mathcal{P}|/\delta)} (\log N)^{\frac{3+2\beta}{2\beta}}}{(1-\gamma)^2 \sqrt{N}} \right).$$

Take $N = \Theta\left(\frac{C_{\exp}C|\mathcal{A}|^2\log(|\mathcal{P}|/\delta)}{(1-\gamma)^4\varepsilon^2}\log^{\frac{3+2\beta}{\beta}}\left(\frac{C|\mathcal{A}|^2\log(|\mathcal{P}|/\delta)}{(1-\gamma)^4\varepsilon^2}\right)\right)$, we can see the sub-optimality gap is smaller than $\varepsilon$, which finishes the proof for $\beta$-exponential decay.

As a result, we finish the proof for the PAC guarantee. $\qquad\square$

### E.3 Proof for the Offline Setting

Similar to the online exploration case, we can obtain the upper bound of the statistical error for $\hat{\pi}$, which is stated in the following:

**Theorem 10** (PAC Guarantee for Offline Exploitation). *Define* $\omega := \max_{s,a}\pi_b^{-1}(a|s)$, *and*

$$C_\pi^* := \sup_{x \in L_2(\mu)} \frac{\mathbb{E}_{(s,a)\sim d_{T^*}^\pi}\left[\langle p^*(\cdot|s,a), x\rangle_{L_2(\mu)}\right]^2}{\mathbb{E}_{(s,a)\sim\rho}\left[\langle p^*(\cdot|s,a), x\rangle_{L_2(\mu)}\right]^2}.$$

*If the penalty and its corresponding parameters are identical to the bonus we define in Theorem 4, then with probability at least $1-\delta$, for any competitor policy $\pi$ including non-Markovian history-dependent policy, we have*

- *For $\beta$-finite spectrum, we have*

$$V_{T^*,r}^\pi - V_{T^*,r}^{\hat{\pi}} \lesssim \frac{\omega\beta^{3/2}\log n}{(1-\gamma)^2}\sqrt{\frac{CC_\pi^*\log(|\mathcal{P}|/\delta)}{n}}$$

- *For $\beta$-polynomial decay, we have*

$$V_{T^*,r}^\pi - V_{T^*,r}^{\hat{\pi}} \lesssim \frac{C_{\text{poly}}\omega n^{\frac{1-\beta}{2(1+\beta)}}\log n\sqrt{CC_\pi^*\log(|\mathcal{P}|/\delta)}}{(1-\gamma)^2}$$

- *For $\beta$-exponential decay, we have*

$$V_{T^*,r}^\pi - V_{T^*,r}^{\hat{\pi}} \lesssim \frac{C_{\exp}\omega(\log n)^{\frac{3+2\beta}{2\beta}}}{(1-\gamma)^2}\sqrt{\frac{CC_\pi^*\log(|\mathcal{P}|/\delta)}{n}}$$

We start by showing that $C_\pi^*$ can be viewed as a measure of the offline data quality, which can be demonstrated by the following lemma, that was first introduced in Chang et al. (2021):

**Lemma 11** (Distribution Shift Lemma). *For any positive definite operator $\Lambda : L_2(\mu) \to L_2(\mu)$, we have that*

$$\mathbb{E}_{(s,a)\sim d_{T^*}^\pi}\langle p^*(\cdot|s,a), \Lambda p^*(\cdot|s,a)\rangle_{L_2(\mu)} \le C_\pi^*\mathbb{E}_{(s,a)\sim\rho}\langle p^*(\cdot|s,a), \Lambda p^*(\cdot|s,a)\rangle_{L_2(\mu)}.$$

*Proof.* We denote the eigendecomposition of $\Lambda$ as $\Lambda = U\Sigma U$ where $\{\sigma_i, u_i\}$ is the eigensystem of $\Lambda$. Then we have

$$\begin{aligned}
&\mathbb{E}_{(s,a)\sim d_{T^*}^\pi}\langle p^*(\cdot|s,a), \Lambda p^*(\cdot|s,a)\rangle_{L_2(\mu)}\\
=&\sum_{i\in I}\sigma_i\mathbb{E}_{(s,a)\sim d_{T^*}^\pi}\langle u_i, p^*(\cdot|s,a)^\top\rangle_{L_2(\mu)}^2\\
\le&C_\pi^*\sum_{i\in I}\sigma_i\mathbb{E}_{(s,a)\sim\rho}\langle u_i, p^*(\cdot|s,a)^\top\rangle_{L_2(\mu)}^2\\
=&C_\pi^*\mathbb{E}_{(s,a)\sim\rho}\langle p^*(\cdot|s,a), \Lambda p^*(\cdot|s,a)\rangle_{L_2(\mu)},
\end{aligned}$$

which finishes the proof. $\qquad\square$

We also define the $\Sigma_{\rho,\phi} : L_2(\mu) \to L_2(\mu)$:

$$\Sigma_{\rho,\phi} := n\mathbb{E}_{(s,a)\sim\rho}\left[\phi(s,a)\phi^\top(s,a)\right] + \lambda T_k^{-1},$$

where $\rho$ is the stationary distribution of $\pi_b$.

**Lemma 12** (One Step Back Inequality for the Learned Model in Offline Setting). *Assume $g : \mathcal{S}\times\mathcal{A}\to\mathbb{R}$, such that $\|g\|_\infty \le B$. Then conditioning on the following generalization bound:*

$$\mathbb{E}_{(s,a)\sim\rho}\|\hat{T}(s,a) - T^*(s,a)\|_1^2 \le \zeta,$$

*we have that* $\forall \pi$

$$\left| \mathbb{E}_{(s,a) \sim d_{\hat{T}}^{\pi}}[g(s,a)] \right|$$

$$\leq \gamma \mathbb{E}_{(\tilde{s},\tilde{a}) \sim d_{\hat{T}}^{\pi}} \|\hat{p}(\cdot|\tilde{s},\tilde{a})\|_{L_2(\mu),\Sigma_{\rho,\hat{p}}^{-1}} \sqrt{n\omega\gamma \mathbb{E}_{(s,a) \sim \rho}\left[g^2(s,a)\right] + \lambda B^2 C + n B^2 \zeta}$$

$$+ \sqrt{(1-\gamma)\omega \mathbb{E}_{(s,a) \sim \rho}[g^2(s,a)]}.$$

*Proof.* We still start from the following inequality:

$$\mathbb{E}_{(s,a) \sim d_{\hat{T}}^{\pi}}[g(s,a)] = \gamma \mathbb{E}_{(\tilde{s},\tilde{a}) \sim d_{\hat{T}}^{\pi}, s \sim \hat{p}_n(\cdot|\tilde{s},\tilde{a}), a \sim \pi(\cdot|s)}[g(s,a)] + (1-\gamma) \mathbb{E}_{s \sim d_0, a \sim \pi(\cdot|s)}[g(s,a)].$$

For the second term, with Jensen's inequality and an importance sampling step, we have that

$$(1-\gamma) \mathbb{E}_{s \sim d_0, a \sim \pi(\cdot|s)}[g(s,a)] \leq \sqrt{(1-\gamma)\omega \mathbb{E}_{(s,a) \sim \rho}[g^2(s,a)]}.$$

For the first term, with Cauchy-Schwartz inequality of $L_2(\mu)$ inner product, we have that

$$\gamma \mathbb{E}_{(\tilde{s},\tilde{a}) \sim d_{\hat{T}}^{\pi}, s \sim \hat{T}(\cdot|\tilde{s},\tilde{a}), a \sim \pi(\cdot|s)}[g(s,a)]$$

$$= \gamma \mathbb{E}_{(\tilde{s},\tilde{a}) \sim d_{\hat{T}}^{\pi}} \left\langle \hat{p}(\cdot|\tilde{s},\tilde{a}), \int_{\mathcal{S}} \sum_{a \in \mathcal{A}} \hat{p}(s|\cdot)\pi(a|s)g(s,a)ds \right\rangle_{L_2(\mu)}$$

$$\leq \gamma \mathbb{E}_{(\tilde{s},\tilde{a}) \sim d_{\hat{T}}^{\pi}} \|\hat{p}(\cdot|\tilde{s},\tilde{a})\|_{L_2(\mu),\Sigma_{\rho,\hat{p}}^{-1}} \left\| \int_{\mathcal{S}} \sum_{a \in \mathcal{A}} \hat{p}(s|\cdot)\pi(a|s)g(s,a)ds \right\|_{L_2(\mu),\Sigma_{\rho,\hat{p}}}.$$

Note that

$$\left\| \int_{\mathcal{S}} \sum_{a \in \mathcal{A}} \hat{p}(s|\cdot)\pi(a|s)g(s,a)ds \right\|^2_{L_2(\mu),\Sigma_{\rho,\hat{p}}}$$

$$= n \mathbb{E}_{(\tilde{s},\tilde{a}) \sim \rho} \left\{ \mathbb{E}_{s \sim \hat{T}(\cdot|\tilde{s},\tilde{a}), a \sim \pi(\cdot|s)}[g(s,a)] \right\}^2 + \lambda \left\| \int_{\mathcal{S}} \sum_{a \in \mathcal{A}} \hat{p}(s|\cdot)\pi(a|s)g(s,a)ds \right\|_{\mathcal{H}_k}$$

$$\leq n \mathbb{E}_{(\tilde{s},\tilde{a}) \sim \rho} \left\{ \mathbb{E}_{s \sim T^*(\cdot|\tilde{s},\tilde{a}), a \sim \pi(\cdot|s)}[g(s,a)] \right\}^2 + \lambda B^2 C + n B^2 \zeta$$

$$\leq n \mathbb{E}_{(\tilde{s},\tilde{a}) \sim \rho} \left\{ \mathbb{E}_{s \sim T^*(\cdot|\tilde{s},\tilde{a}), a \sim \pi(\cdot|s)}[g^2(s,a)] \right\} + \lambda B^2 C + n B^2 \zeta$$

$$\leq n \omega \mathbb{E}_{(\tilde{s},\tilde{a}) \sim \rho} \left\{ \mathbb{E}_{s \sim T^*(\cdot|\tilde{s},\tilde{a}), a \sim \pi_b(\cdot|s)}[g^2(s,a)] \right\} + \lambda B^2 C + n B^2 \zeta$$

$$\leq \frac{n\omega}{\gamma} \mathbb{E}_{(s,a) \sim \rho}[g^2(s,a)] + \lambda B^2 C + n B^2 \zeta.$$

Substitute back, we have the desired result. $\qquad \square$

**Lemma 13** (One Step Back Inequality for the True Model in Offline Setting). *Assume* $g : \mathcal{S} \times \mathcal{A} \to \mathbb{R}$, *such that* $\|g\|_\infty \leq B$. *Then we have that* $\forall \pi$,

$$\left| \mathbb{E}_{(s,a) \sim d_{T^*}^{\pi}}[g(s,a)] \right|$$

$$\leq \gamma \mathbb{E}_{(\tilde{s},\tilde{a}) \sim d_{T^*}^{\pi}} \|p^*(\cdot|\tilde{s},\tilde{a})\|_{L_2(\mu),\Sigma_{\rho,\hat{p}}^{-1}} \sqrt{n\omega\gamma \mathbb{E}_{(s,a) \sim \rho}\left[g^2(s,a)\right] + \lambda \gamma^2 B^2 C}$$

$$+ \sqrt{(1-\gamma)\omega \mathbb{E}_{(s,a) \sim \rho}[g^2(s,a)]}.$$

*Proof.* The proof for this lemma is nearly identical to the previous lemma, and we omit it for simplicity. $\qquad \square$

**Lemma 14** (Almost Pessimism at the Initial Distribution). *If we set* $\zeta = \Theta\left(\frac{\log(|\mathcal{P}|/\delta)}{n}\right)$, $\lambda$ *for different eigendecay condition as follows:*

- *$\beta$-finite spectrum:* $\lambda = \Theta(\beta \log n + \log(|\mathcal{P}|/\delta))$

- *$\beta$-polynomial decay:* $\lambda = \Theta(C_{\text{poly}} n^{1/(1+\beta)} + \log(|\mathcal{P}|/\delta))$;

- *$\beta$-exponential decay:* $\lambda = \Theta(C_{\exp}(\log n)^{1/\beta} + \log(|\mathcal{P}|/\delta))$;

*and* $\alpha = \Theta\left(\frac{\gamma}{1-\gamma}\sqrt{\omega\log(|\mathcal{P}|/\delta) + \lambda\gamma^2 C}\right)$, *the following events hold with probability at least* $1 - \delta$:

$$\forall \pi, \quad V_{\hat{T},r-b}^\pi - V_{T^*,r}^\pi \le \sqrt{\frac{\omega\zeta}{(1-\gamma)^3}}.$$

*Proof.* Note that, with the proof of Lemma 17 and a union bound over $\mathcal{P}$ (but not over $n$), we know using the chosen $\lambda$, $\forall \hat{T} \in \mathcal{P}$, with probability at least $1 - \delta$,

$$\|\hat{p}(\cdot|s,a)\|_{L_2(\mu),\hat{\Sigma}_{n,\hat{p}}^{-1}} = \Theta\left(\|\hat{p}(\cdot|s,a)\|_{L_2(\mu),\Sigma_{\rho,\hat{p}}}^{-1}\right).$$

With Lemma 18, we have that

$$(1-\gamma)\left(V_{\hat{T},r-b}^\pi - V_{T^*,r}^\pi\right)$$

$$= \mathbb{E}_{(s,a)\sim d_{\hat{T}}^\pi}\left[-b(s,a) + \gamma\mathbb{E}_{\hat{T}(s'|s,a)}\left[V_{T^*,r}^\pi(s')\right] - \gamma\mathbb{E}_{T^*(s'|s,a)}\left[V_{T^*,r}^\pi(s')\right]\right]$$

$$\lesssim \mathbb{E}_{(s,a)\sim d_{\hat{T}}^\pi}\left[-\min\left\{\alpha\|\hat{p}(\cdot|s,a)\|_{L_2(\mu),\Sigma_{\rho,\hat{p}}^{-1}}, 2\right\} + \gamma\mathbb{E}_{\hat{T}(s'|s,a)}\left[V_{T^*,r}^\pi(s')\right] - \gamma\mathbb{E}_{T^*(s'|s,a)}\left[V_{T^*,r}^\pi(s')\right]\right]$$

Denote $f(s,a) = \mathrm{TV}(\hat{T}(s,a), T^*(s,a))$, we know $\|f\|_\infty \le 2$. With Hölder's inequality, we can obtain that

$$\left|\mathbb{E}_{\hat{T}(s'|s,a)}\left[V_{T^*,r}^\pi(s')\right] - \mathbb{E}_{T^*(s'|s,a)}\left[V_{T^*,r}^\pi(s')\right]\right| \le \mathbb{E}_{(s,a)\sim d_{\hat{T}}^\pi}\left[\frac{f(s,a)}{1-\gamma}\right].$$

With Lemma 12, we have that

$$\mathbb{E}_{(s,a)\sim d_{\hat{T}}^\pi}\left[\frac{f(s,a)}{1-\gamma}\right]$$

$$\le \mathbb{E}_{(\widetilde{s},\widetilde{a})\sim d_{\hat{T}}^\pi}\|\hat{p}(\cdot|\widetilde{s},\widetilde{a})\|_{L_2(\mu),\Sigma_{\rho,\hat{p}}^{-1}}\sqrt{\frac{n\omega\gamma^2\mathbb{E}_{(s,a)\sim\rho}[f^2(s,a)]}{(1-\gamma)^2} + \frac{4\lambda\gamma^2 C}{(1-\gamma)^2} + \frac{4n\gamma^2\zeta}{(1-\gamma)^2}}$$

$$+ \sqrt{\frac{\omega\mathbb{E}_{(s,a)\sim\rho}[f^2(s,a)]}{1-\gamma}}$$

$$\le \mathbb{E}_{(\widetilde{s},\widetilde{a})\sim d_{\hat{T}}^\pi}\|\hat{p}(\cdot|\widetilde{s},\widetilde{a})\|_{L_2(\mu),\Sigma_{\rho,\hat{p}}^{-1}}\sqrt{\frac{n\omega\gamma^2\zeta_n}{(1-\gamma)^2} + \frac{4\lambda\gamma^2 C}{(1-\gamma)^2} + \frac{4n\gamma^2\zeta}{(1-\gamma)^2}} + \sqrt{\frac{\omega\zeta}{1-\gamma}}.$$

With the choice of $\alpha$, we can conclude the proof. $\qquad\square$

**Theorem 15** (PAC Guarantee for Offline Setting). *With probability at least* $1 - \delta$, *for any competitor policy* $\pi$ *including non-Markovian history-dependent policy, we have*

- *For $\beta$-finite spectrum, we have*

$$V_{T^*,r}^\pi - V_{\hat{T}^*,r}^{\hat{\pi}} \lesssim \frac{\omega\beta^{3/2}\log n}{(1-\gamma)^2}\sqrt{\frac{CC_\pi^*\log(|\mathcal{P}|/\delta)}{n}}$$

- *For $\beta$-polynomial decay, we have*

$$V_{T^*,r}^\pi - V_{T^*,r}^{\hat{\pi}} \lesssim \frac{C_{\mathrm{poly}}\omega n^{\frac{1-\beta}{2(1+\beta)}}\log n\sqrt{CC_\pi^*\log(|\mathcal{P}|/\delta)}}{(1-\gamma)^2}$$

- *For $\beta$-exponential decay, we have*

$$V_{T^*,r}^\pi - V_{T^*,r}^{\hat{\pi}} \lesssim \frac{C_{\exp}\omega(\log n)^{\frac{3+2\beta}{2\beta}}}{(1-\gamma)^2}\sqrt{\frac{CC_\pi^*\log(|\mathcal{P}|/\delta)}{n}}$$

*Proof.* With Lemma 14 and Lemma 18, we have that

$$V_{T^*,r}^\pi - V_{\hat{T}^*,r}^{\hat{\pi}}$$

$$\le V_{T^*,r}^\pi - V_{\hat{T},r-b}^{\hat{\pi}} + \sqrt{\frac{\omega\zeta}{(1-\gamma)^3}}$$

$$\leq V_{T^*,r}^\pi - V_{\hat{T},r-b}^\pi + \sqrt{\frac{\omega\zeta}{(1-\gamma)^3}}$$

$$\leq \frac{1}{1-\gamma}\mathbb{E}_{(s,a)\sim d_{T^*}^\pi}\left[b(s,a) + \gamma\mathbb{E}_{T^*(s'|s,a)}\left[V_{T^*,r}^\pi(s')\right] - \gamma\mathbb{E}_{\hat{T}(s'|s,a)}\left[V_{T^*,r}^\pi(s')\right]\right] + \sqrt{\frac{\omega\zeta}{(1-\gamma)^3}}.$$

As $b = O(1)$, with Lemma 13, we have that

$$\mathbb{E}_{(s,a)\sim d_{T^*}^\pi}[b(s,a)]$$

$$\lesssim \mathbb{E}\left[\min\left\{\alpha\left\|\hat{p}(\cdot|s,a)\right\|_{L_2(\mu),\Sigma_{\rho,\hat{p}}^{-1}}, 2\right\}\right]$$

$$\lesssim \mathbb{E}_{(\widetilde{s},\widetilde{a})\sim d_{T^*}^\pi}\left\|p^*(\cdot|\widetilde{s},\widetilde{a})\right\|_{L_2(\mu),\Sigma_{\rho,p^*}^{-1}} \sqrt{n\omega\gamma\alpha^2\mathbb{E}_{(s,a)\sim\rho}\left[\|\hat{p}(\cdot|s,a)\|_{L_2(\mu),\Sigma_{\rho,\hat{p}}^{-1}}^2\right] + \lambda\gamma^2 C}$$

$$+ \sqrt{(1-\gamma)\omega\alpha^2\mathbb{E}_{(s,a)\sim\rho}\left[\|\hat{p}(\cdot|s,a)\|_{L_2(\mu),\Sigma_{\rho,\hat{p}}^{-1}}^2\right]}.$$

With the reasoning similar to the proof in Lemma 8, we have that

- For $\beta$-finite spectrum,

$$\mathbb{E}_{(s,a)\sim\rho}\left[\|\hat{p}(\cdot|s,a)\|_{L_2(\mu),\Sigma_{\rho,\hat{p}}^{-1}}^2\right] = O\left(\beta\log n\right),$$

which leads to

$$\mathbb{E}_{(s,a)\sim d_{T^*}^\pi}[b(s,a)] \lesssim \mathbb{E}_{(\widetilde{s},\widetilde{a})\sim d_{T^*}^\pi}\left\|p^*(\cdot|\widetilde{s},\widetilde{a})\right\|_{L_2(\mu),\Sigma_{\rho,p^*}^{-1}} \sqrt{\omega\gamma\beta\alpha^2\log n + \lambda\gamma^2 C}$$

$$+ \sqrt{\frac{(1-\gamma)\omega\beta\alpha^2\log n}{n}}.$$

- For $\beta$-polynomial decay,

$$\mathbb{E}_{(s,a)\sim\rho}\left[\|\hat{p}(\cdot|s,a)\|_{L_2(\mu),\Sigma_{\rho,\hat{p}}^{-1}}^2\right] = O\left(C_{\text{poly}}n^{\frac{1}{2(1+\beta)}}\log n\right),$$

which leads to

$$\mathbb{E}_{(s,a)\sim d_{T^*}^\pi}[b(s,a)] \lesssim \mathbb{E}_{(\widetilde{s},\widetilde{a})\sim d_{T^*}^\pi}\left\|p^*(\cdot|\widetilde{s},\widetilde{a})\right\|_{L_2(\mu),\Sigma_{\rho,p^*}^{-1}} \sqrt{\omega\gamma C_{\text{poly}}\alpha^2 n^{\frac{1}{2(1+\beta)}}\log n + \lambda\gamma^2 C}$$

$$+ \sqrt{(1-\gamma)\omega C_{\text{poly}}\alpha^2 n^{-1+\frac{1}{2(1+\beta)}}\log n}.$$

- For $\beta$-exponential decay,

$$\mathbb{E}_{(s,a)\sim\rho}\left[\|\hat{p}(\cdot|s,a)\|_{L_2(\mu),\Sigma_{\rho,\hat{p}}^{-1}}^2\right] = O\left(C_{\exp}(\log n)^{1+1/\beta}\right),$$

which leads to

$$\mathbb{E}_{(s,a)\sim d_{T^*}^\pi}[b(s,a)] \lesssim \mathbb{E}_{(\widetilde{s},\widetilde{a})\sim d_{T^*}^\pi}\left\|p^*(\cdot|\widetilde{s},\widetilde{a})\right\|_{L_2(\mu),\Sigma_{\rho,p^*}^{-1}} \sqrt{\omega\gamma C_{\exp}\alpha^2(\log n)^{1+1/\beta} + \lambda\gamma^2 C}$$

$$+ \sqrt{\frac{(1-\gamma)\omega\alpha^2 C_{\exp}(\log n)^{1+1/\beta}}{n}}.$$

Furthermore, denote $f(s,a) = \text{TV}(\hat{T}(s,a), T^*(s,a))$, we have $\|f\|_\infty \leq 2$. With Hölder's inequality, we have

$$\left|\mathbb{E}_{(s,a)\sim d_{T^*}^\pi}\left[\mathbb{E}_{T^*(s'|s,a)}\left[V_{T^*,r}^\pi(s')\right] - \mathbb{E}_{\hat{T}(s'|s,a)}\left[V_{T^*,r}^\pi(s')\right]\right]\right| \leq \mathbb{E}_{(s,a)\sim d_{T^*}^\pi}\left[\frac{f(s,a)}{1-\gamma}\right].$$

With Lemma 13, we have

$$\mathbb{E}_{(s,a)\sim d_{T^*}^\pi}\left[\frac{f(s,a)}{1-\gamma}\right] \leq \mathbb{E}_{(\widetilde{s},\widetilde{a})\sim d_{T^*}^\pi}\left\|p^*(\cdot|\widetilde{s},\widetilde{a})\right\|_{L_2(\mu),\Sigma_{\rho,p^*}^{-1}} \sqrt{\frac{n\omega\gamma\mathbb{E}_{(s,a)\sim\rho}[f^2(s,a)]}{(1-\gamma)^2} + \frac{4\lambda\gamma^2 C}{(1-\gamma)^2}}$$

$$+ \sqrt{\frac{\omega\mathbb{E}_{(s,a)\sim\rho}[f^2(s,a)]}{1-\gamma}}$$

$$\leq \mathbb{E}_{(\widetilde{s},\widetilde{a})\sim d_{T^*}^\pi}\left\|p^*(\cdot|\widetilde{s},\widetilde{a})\right\|_{L_2(\mu),\Sigma_{\rho,p^*}^{-1}} \cdot \sqrt{\frac{n\omega\gamma\zeta}{(1-\gamma^2)} + \frac{4\lambda\gamma^2 C}{(1-\gamma)^2}} + \sqrt{\frac{\omega\zeta}{(1-\gamma)}}$$

$$\lesssim \alpha_n \mathbb{E}_{(\widetilde{s},\widetilde{a}) \sim d^\pi_{T^*}} \|p^*(\cdot|\widetilde{s},\widetilde{a})\|_{L_2(\mu),\Sigma^{-1}_{\rho,p^*}} + \sqrt{\frac{\omega\zeta}{1-\gamma}}.$$

Combine with the previous results and take the dominating terms out, we have that

- For $\beta$-finite spectrum,
$$V^\pi_{T^*,r} - V^{\hat{\pi}}_{T^*,r}$$
$$\lesssim \frac{1}{1-\gamma} \mathbb{E}_{(\widetilde{s},\widetilde{a}) \sim d^\pi_{T^*}} \|p^*(\cdot|\widetilde{s},\widetilde{a})\|_{L_2(\mu),\Sigma^{-1}_{\rho,p^*}} \sqrt{\omega\gamma\beta\alpha^2 \log n + \lambda\gamma^2 C}$$
$$+ \sqrt{\frac{\omega\beta\alpha^2 \log n}{(1-\gamma)n}} + \sqrt{\frac{\omega\zeta}{(1-\gamma)^3}}.$$

- For $\beta$-polynomial decay,
$$V^\pi_{T^*,r} - V^{\hat{\pi}}_{T^*}$$
$$\lesssim \frac{1}{1-\gamma} \mathbb{E}_{(\widetilde{s},\widetilde{a}) \sim d^\pi_{T^*}} \|p^*(\cdot|\widetilde{s},\widetilde{a})\|_{L_2(\mu),\Sigma^{-1}_{\rho,p^*}} \sqrt{\omega\gamma C_{\text{poly}}\alpha^2 n^{\frac{1}{2(1+\beta)}} \log n + \lambda\gamma^2 C}$$
$$+ \sqrt{\frac{\omega C_{\text{poly}}\alpha^2 n^{-1+\frac{1}{2(1+\beta)}} \log n}{1-\gamma}} + \sqrt{\frac{\omega\zeta}{(1-\gamma)^3}}.$$

- For $\beta$-exponential decay,
$$V^\pi_{T^*,r} - V^{\hat{\pi}}_{T^*,r}$$
$$\lesssim \frac{1}{1-\gamma} \mathbb{E}_{(\widetilde{s},\widetilde{a}) \sim d^\pi_{T^*}} \|p^*(\cdot|\widetilde{s},\widetilde{a})\|_{L_2(\mu),\Sigma^{-1}_{\rho,p^*}} \sqrt{\omega\gamma C_{\exp}\alpha^2 (\log n)^{1/\beta} \log\log n + \lambda\gamma^2 C}$$
$$+ \sqrt{\frac{\omega C_{\exp}\alpha^2 (\log n)^{1+1/\beta}}{(1-\gamma)n}} + \sqrt{\frac{\omega\zeta}{(1-\gamma)^3}}.$$

We now deal with the term $\mathbb{E}_{(\widetilde{s},\widetilde{a}) \sim d^\pi_{T^*}} \|p^*(\cdot|\widetilde{s},\widetilde{a})\|_{L_2(\mu),\Sigma^{-1}_{\rho,p^*}}$. With Lemma 11, we know

$$\mathbb{E}_{(\widetilde{s},\widetilde{a}) \sim d^\pi_{T^*}} \|p^*(\cdot|\widetilde{s},\widetilde{a})\|_{L_2(\mu),\Sigma^{-1}_{\rho,p^*}}$$
$$\leq \sqrt{\mathbb{E}_{(\widetilde{s},\widetilde{a}) \sim d^\pi_{T^*}} \|p^*(\cdot|\widetilde{s},\widetilde{a})\|^2_{L_2(\mu),\Sigma^{-1}_{\rho,p^*}}}$$
$$\leq \sqrt{C\mathbb{E}_{(\widetilde{s},\widetilde{a}) \sim \rho} \|p^*(\cdot|\widetilde{s},\widetilde{a})\|^2_{L_2(\mu),\Sigma^{-1}_{\rho,p^*}}}.$$

Applying the identical method used in the proof of Lemma 8, we have that:

- For $\beta$-finite spectrum, we have
$$\mathbb{E}_{(\widetilde{s},\widetilde{a}) \sim \rho} \left[ \|p^*(\cdot|\widetilde{s},\widetilde{a})\|^2_{L_2(\mu),\Sigma^{-1}_{\rho,p^*}} \right] = O\left(\beta \log n\right).$$

- For $\beta$-polynomial decay, we have
$$\mathbb{E}_{(\widetilde{s},\widetilde{a}) \sim \rho} \left[ \|p^*(\cdot|\widetilde{s},\widetilde{a})\|^2_{L_2(\mu),\Sigma^{-1}_{\rho,p^*}} \right] = O\left(C_{\text{poly}} n^{\frac{1}{2(1+\beta)}} \log n\right).$$

- For $\beta$-exponential decay, we have
$$\mathbb{E}_{(\widetilde{s},\widetilde{a}) \sim \rho} \left[ \|p^*(\cdot|\widetilde{s},\widetilde{a})\|^2_{L_2(\mu),\Sigma^{-1}_{\rho,p^*}} \right] = O\left(C_{\exp} (\log n)^{1+1/\beta}\right).$$

Substitute $\alpha$ and $\lambda$ back, we have that:

- For $\beta$-finite spectrum, we have
$$V^\pi_{T^*,r} - V^{\hat{\pi}}_{T^*,r} \lesssim \frac{\omega\beta^{3/2} \log n}{(1-\gamma)^2} \sqrt{\frac{CC^*_\pi \log(|\mathcal{P}|/\delta)}{n}}$$

- For $\beta$-polynomial decay, we have

$$V_{T^*,r}^{\pi} - V_{T^*,r}^{\hat{\pi}} \lesssim \frac{C_{\text{poly}}\omega n^{\frac{1-\beta}{2(1+\beta)}} \log n \sqrt{CC_\pi^* \log(|\mathcal{P}|/\delta)}}{(1-\gamma)^2}$$

- For $\beta$-exponential decay, we have

$$V_{T^*,r}^{\pi} - V_{T^*,r}^{\hat{\pi}} \lesssim \frac{C_{\text{exp}}\omega(\log n)^{\frac{3+2\beta}{2\beta}}}{(1-\gamma)^2}\sqrt{\frac{CC_\pi^* \log(|\mathcal{P}|/\delta)}{n}}$$

This finishes the proof. $\square$

## F    AUXILLARY LEMMAS

We first state the MLE generalization bound from (Agarwal et al., 2020). Note that, when Assumption 1 holds, the MLE generalization bound only depends on the complexity of $\mathcal{P}$.

**Lemma 16** (MLE Generalization Bound (Agarwal et al., 2020)). *For a fixed episode $n$, with probability at least $1 - \delta$, we have that*

$$\mathbb{E}_{s\sim 0.5\rho_n+0.5\rho_n', a\sim\mathcal{U}(\mathcal{A})}\left[\left\|\hat{T}_n(s,a) - T^*(s,a)\right\|_1^2\right] \leq \frac{\log(|\mathcal{P}|/\delta)}{n}.$$

*With a union bound, with probability at least $1 - \delta$, we have that*

$$\forall n \in \mathbb{N}^+, \mathbb{E}_{s\sim 0.5\rho_n+0.5\rho_n', a\sim\mathcal{U}(\mathcal{A})}\left[\left\|\hat{T}_n(s,a) - T^*(s,a)\right\|_1^2\right] \leq \frac{\log(n|\mathcal{P}|/\delta)}{n}$$

**Lemma 17** (Concentration of the Bonuses). *Let $\mu_i$ be the conditional distribution of $\phi$ given the sampled $\phi_1, \cdots, \phi_{i-1}$, define $\Sigma : L_2(\mu) \to L_2(\mu)$, $\Sigma_n := \frac{1}{n}\sum_{i\in[n]}\mathbb{E}_{\phi\sim\mu_i}\phi\phi^\top$. Assume $\|\phi\|_{\mathcal{H}_k} \leq 1$ for any realization of $\phi$. If $\lambda$ satisfies the following conditions for each eigendecay condition:*

- *$\beta$-finite spectrum: $\lambda = \Theta(\beta \log N + \log(N/\delta))$;*

- *$\beta$-polynomial decay: $\lambda = \Theta(C_{\text{poly}}N^{1/(1+\beta)} + \log(N/\delta))$;*

- *$\beta$-exponential decay: $\lambda = \Theta(C_{\text{exp}}(\log N)^{1/\beta} + \log(N/\delta))$, where $C_3$ is a constant depends on $C_1$ and $C_2$;*

*then there exists absolute constant $c_1$ and $c_2$, such that $\forall x \in \mathcal{H}_k$, the following event holds with probability at least $1 - \delta$:*

$$\forall n \in [N], \quad c_1 \left\langle x, \left(n\Sigma_n + \lambda T_k^{-1}\right)x\right\rangle_{L_2(\mu)} \leq \left\langle x, \left(\sum_{i\in[n]}\phi_i\phi_i^\top + \lambda T_k^{-1}\right)x\right\rangle_{L_2(\mu)},$$

*and* $\left\langle x, \left(\sum_{i\in[n]}\phi_i\phi_i^\top + \lambda T_k^{-1}\right)x\right\rangle_{L_2(\mu)} \leq c_2 \left\langle x, \left(n\Sigma_n + \lambda T_k^{-1}\right)x\right\rangle_{L_2(\mu)}.$

*In the same event above, the following event must hold as well:*

$$\forall n \in [N], \quad \frac{1}{c_2}\left\langle x, \left(n\Sigma_n + \lambda T_k^{-1}\right)^{-1}x\right\rangle_{L_2(\mu)} \leq \left\langle x, \left(\sum_{i\in[n]}\phi_i\phi_i^\top + \lambda T_k^{-1}\right)^{-1}x\right\rangle_{L_2(\mu)},$$

*and* $\left\langle x, \left(\sum_{i\in[n]}\phi_i\phi_i^\top + \lambda T_k^{-1}\right)^{-1}x\right\rangle_{L_2(\mu)} \leq \frac{1}{c_1}\left\langle x, \left(n\Sigma_n + \lambda T_k^{-1}\right)^{-1}x\right\rangle_{L_2(\mu)}$

*Proof.* Note that, $\|T_k^{-1/2}\phi\|_{L_2(\mu)} = \|\phi\|_{\mathcal{H}_k} \leq 1$. Hence, the operator norm of operators $\widetilde{\Sigma}_n := T_k^{-1/2}\Sigma_n T_k^{-1/2}$ that maps from $L_2(\mu)$ to $L_2(\mu)$ are upper bounded by 1. For notation simplicity, we

define $\widetilde{\phi} := T_k^{-1/2}\phi$ and $\widetilde{\mu}_i$ denotes the conditional distribution of $\widetilde{\phi}$ given the sampled $\widetilde{\phi}_1, \cdots, \widetilde{\phi}_{i-1}$. Note that $\forall x \in \mathcal{H}_k$, $T_k^{1/2}x \in \mathcal{H}_{\widetilde{k}}$. We now prove the following equivalent form of the claim: $\forall x \in \mathcal{H}_{\widetilde{k}}, \forall n \geq 1,$

$$c_1 \left\langle x, \left(n\widetilde{\Sigma}_n + \lambda T_k^{-2}\right) x \right\rangle_{L_2(\mu)} \leq \left\langle x, \left(\sum_{i\in[n]} \widetilde{\phi}\widetilde{\phi}^\top + \lambda T_k^{-2}\right) x \right\rangle_{L_2(\mu)} \leq c_2 \left\langle x, \left(n\widetilde{\Sigma}_n + \lambda T_k^{-2}\right) x \right\rangle_{L_2(\mu)}$$

It is sufficient to consider $x$ with $\|x\|_{\mathcal{H}_{\widetilde{k}}} = 1$. Note that, we have

$$\left\langle x, \widetilde{\phi}\widetilde{\phi}^\top x \right\rangle_{L_2(\mu)} = \left\langle x, \widetilde{\phi} \right\rangle_{L_2(\mu)}^2 \leq \|x\|_{L_2(\mu)}^2.$$

Denote $\widetilde{\Sigma}^i := \mathbb{E}_{\phi\sim\mu_i}\widetilde{\phi}\widetilde{\phi}^\top$. We have

$$\mathrm{Var}_{\phi\sim\mu_i}\left[\langle x, \widetilde{\phi}\rangle_{L_2(\mu)}^2\right] \leq \|x\|_{L_2(\mu)}^2 \mathbb{E}_{\widetilde{\phi}\sim\widetilde{\mu}_i}\left[\langle x, \widetilde{\phi}\rangle_{L_2(\mu)}^2\right] = \|x\|_{L_2(\mu)}^2 \langle x, \widetilde{\Sigma}^i x\rangle_{L_2(\mu)}$$

we can invoke a Bernstein-style martingale concentration inequality (Lemma 45, Zanette et al., 2021), and obtain that with probability at least $1 - \delta$

$$\left| \frac{1}{n} \sum_{i\in[n]} \left[\langle x, \widetilde{\phi}_i\rangle_{L_2(\mu)}^2\right] - \langle x, \widetilde{\Sigma}x\rangle_{L_2(\mu)} \right| \leq c\left( \sqrt{\frac{\|x\|_{L_2(\mu)}^2 \langle x, \widetilde{\Sigma}_n x\rangle_{L_2(\mu)} \log(2/\delta)}{n}} + \frac{\|x\|_{L_2(\mu)}^2 \log(2/\delta)}{3n} \right),$$

where $c$ is an absolute constant. We then show that, if we have $\lambda = \Omega(\log(1/\delta))$, we have that

$$c\left( \sqrt{\frac{\|x\|_{L_2(\mu)}^2 \langle x, \widetilde{\Sigma}x\rangle_{L_2(\mu)} \log(2/\delta)}{n}} + \frac{\|x\|_{L_2(\mu)^2} \log(2/\delta)}{3n} \right) \leq C\left( \langle x, \widetilde{\Sigma}x\rangle_{L_2(\mu)} + \frac{\lambda\|x\|_{L_2(\mu)}^2}{n} \right).$$

where $C < 1$ is an absolute constant, following the similar reasoning in the proof of Lemma 39 in (Zanette et al., 2021):

- $\langle x, \widetilde{\Sigma}x\rangle_{L_2(\mu)} \leq \frac{\lambda\|x\|_{L_2(\mu)}^2}{n}$: It's sufficient to show that $c\sqrt{\lambda\log(2/\delta)} \leq \frac{C\lambda}{2}$ and $\frac{c\log(2/\delta)}{3} \leq \frac{C\lambda}{2}$, which can be achieved by $\lambda = \Omega(\log 1/\delta)$.

- $\langle x, \widetilde{\Sigma}x\rangle_{L_2(\mu)} \geq \frac{\lambda\|x\|_{L_2(\mu)}^2}{n}$: It's sufficient to show that

$$\lambda \geq \frac{c}{C}\log(2/\delta) \quad \text{and} \quad \frac{c}{C}\sqrt{\frac{\|x\|_{L_2(\mu)}^2 \log(2/\delta)}{n}} \leq \sqrt{\langle x, \widetilde{\Sigma}x\rangle_{L_2(\mu)}}.$$

  As $\langle x, \widetilde{\Sigma}x\rangle_{L_2(\mu)} \geq \frac{\lambda\|x\|_{L_2(\mu)}^2}{n}$, when $\lambda \geq \max\left\{\frac{c}{C}, \frac{c^2}{C^2}\right\}\log(2/\delta)$, these two conditions hold simultaneously.

Hence, for any fixed $x$ with $\|x\|_{\mathcal{H}_{\widetilde{k}}} = 1$, we have

$$\left| \frac{1}{n}\sum_{i\in[n]} \left[\langle x, \widetilde{\phi}_i\rangle_{L_2(\mu)}^2\right] - \langle x, \widetilde{\Sigma}x\rangle_{L_2(\mu)} \right| \leq C\left\langle x, \left(\widetilde{\Sigma} + \frac{\lambda}{n}I\right)x \right\rangle_{L_2(\mu)} \leq C\left\langle x, \left(\widetilde{\Sigma} + \frac{\lambda}{n}T_k^{-2}\right)x \right\rangle_{L_2(\mu)}.$$

Now, assume such condition holds for an $\varepsilon$-net $\mathcal{B}_\varepsilon$ of $\mathcal{S}_{\mathcal{H}_{\widetilde{k}}}$, the unit sphere of RKHS $\mathcal{H}_{\widetilde{k}}$ (i.e. $\{x : \|x\|_{\mathcal{H}_{\widetilde{k}}} = 1\}$), under $\|\cdot\|_{L_2(\mu)}$. Then $\forall x$ satisfies $\|x\|_{\mathcal{H}_{\widetilde{k}}} = 1$, let $x'$ be the closest point of $x$ in $\mathcal{B}_\varepsilon$ under $\|\cdot\|_{L_2(\mu)}$ (note that $x' \in \mathcal{H}_{\widetilde{k}}$ by the definition of $\varepsilon$-net). We have that

$$\left| \langle x, \widetilde{\Sigma}x\rangle - \langle x', \widetilde{\Sigma}x'\rangle \right| \leq 2\varepsilon$$

$$\left| \left\langle x, \left(\frac{1}{n}\sum_{i\in[n]}\widetilde{\phi}_i\widetilde{\phi}_i^\top\right)x \right\rangle - \left\langle x', \left(\frac{1}{n}\sum_{i\in[n]}\widetilde{\phi}_i\widetilde{\phi}_i^\top\right)x' \right\rangle \right| \leq 2\varepsilon$$

With a triangle inequality, $\forall n, \forall \|x\|_{\mathcal{H}_k} \leq 1$, we have

$$\left| \left\langle x, \left( \frac{1}{n} \sum_{i \in [n]} \widetilde{\phi}_i \widetilde{\phi}_i^\top + \frac{\lambda}{n} T_k^{-2} \right) x \right\rangle_{L_2(\mu)} - \left\langle x, \left( \widetilde{\Sigma} + \frac{\lambda}{n} T_k^{-2} \right) x \right\rangle_{L_2(\mu)} \right|$$

$$\leq C \left\langle x, \left( \widetilde{\Sigma} + \frac{\lambda}{n} T_k^{-2} \right) x \right\rangle_{L_2(\mu)} + (4 + 2C)\varepsilon$$

Hence, we can choose $\varepsilon = O\left( \frac{\lambda}{n} \right)$, to guarantee that

$$C \left\langle x, \left( \widetilde{\Sigma} + \frac{\lambda}{n} T_k^{-2} \right) x \right\rangle_{L_2(\mu)} + (4 + 2C)\varepsilon \leq C' \left\langle x, \left( \widetilde{\Sigma} + \frac{\lambda}{n} T_k^{-2} \right) x \right\rangle_{L_2(\mu)},$$

where $C' < 1$ is an absolute constant.

Now we consider the covering number $\mathcal{N}(\mathbb{S}_{\mathcal{H}_{\widetilde{k}}}, \|\cdot\|_{L_2(\mu)}, \varepsilon)$. We start from the entropy number $e_i(\mathcal{S}_{\mathcal{H}_{\widetilde{k}}}, \|\cdot\|_{L_2(\mu)})$. From (A.36) in Steinwart & Christmann (2008), we know

$$e_i(\mathcal{B}_{\mathcal{H}_{\widetilde{k}}}, \|\cdot\|_{L_2(\mu)}) \leq e_i(\mathcal{S}_{\mathcal{H}_{\widetilde{k}}}, \|\cdot\|_{L_2(\mu)}) \leq 2e_i(\mathcal{B}_{\mathcal{H}_{\widetilde{k}}}, \|\cdot\|_{L_2(\mu)}),$$

where $\mathcal{B}_{\mathcal{H}_{\widetilde{k}}}$ is the unit ball in RKHS $\mathcal{H}_{\widetilde{k}}$ (i.e. $\{x : \|x\|_{\mathcal{H}_{\widetilde{k}}} \leq 1\}$). With Carl's inequality[1] (Carl & Stephani, 1990) (also see (Steinwart et al., 2009)), $\forall p > 0, m \in \mathbb{N}^+$, we have

$$\sup_{i \in [m]} i^{1/p} e_i(\mathrm{id} : \mathcal{H}_{\widetilde{k}} \to L_2(\mu))$$

$$\leq c_p \sup_{i \in [m]} i^{1/p} \mu_i^{1/2} \left( T_k^2 : L_2(\mu) \to L_2(\mu) \right)$$

$$= c_p \sup_{i \in [m]} i^{1/p} \mu_i \left( T_k : L_2(\mu) \to L_2(\mu) \right)$$

where $c_p = 128(32 + 16/p)^{1/p}$ denotes a constant only depending on $p$. We then consider the entropy number under different eigendecay conditions:

- For $\beta$-finite spectrum, as we have $\sum_{i \in I} \mu_i \leq 1$ from Assumption 3, and $\forall i > \beta, \mu_i(T_k : L_2(\mu) \to L_2(\mu)) = 0$, we know for a fixed $p$,

  $$e_i(\mathrm{id} : \mathcal{H}_{\widetilde{k}} \to L_2(\mu)) \leq 128 \left((32 + 16/p)\right)^{1/p} (\beta/i)^{1/p}.$$

  Take $p = \beta/i$, we know that

  $$e_i(\mathrm{id} : \mathcal{H}_{\widetilde{k}} \to L_2(\mu)) \leq 128(32\beta + 16)^{-i/\beta}.$$

- For $\beta$-polynomial decay, take $p = 2/\beta$ and obtain that

  $$e_i(\mathrm{id} : \mathcal{H}_{\widetilde{k}} \to L_2(\mu)) \leq 128C_0(32 + 8\beta)^{\beta/2} i^{-\beta}.$$

- For $\beta$-exponential decay, note that, for a fixed $p$, direct computation shows the maximum of $i^{1/p} \exp(-C_2 i^\beta)$ is achieved when $i^\beta = \frac{1}{C_2 \beta p}$. Furthermore, $i^{1/p} \exp(-C_2 i^\beta)$ is monotonically increasing with respect to $i$ when $i^\beta < \frac{1}{C_2 \beta p}$, while monotonically decreasing with respect to $i$ when $i^\beta > \frac{1}{C_2 \beta p}$. Hence, for a given $i$, we can choose $p$ such that $i^\beta > \frac{1}{C_2 \beta p}$, and obtain that

  $$e_i(\mathrm{id} : \mathcal{H}_{\widetilde{k}} \to L_2(\mu)) \leq 128(32 + 16/p)^{1/p} C_1 \exp(-C_2 i^\beta).$$

  As we can take $p \to \infty$, we have that

  $$e_i(\mathrm{id} : \mathcal{H}_{\widetilde{k}} \to L_2(\mu)) \leq 128C_1 \exp(-C_2 i^\beta).$$

We now convert the entropy number bound for different eigendecay conditions to the covering number bound accordingly.

---

[1]A more formal claim is on the approximation number of the bounded linear operator, which, as shown in Steinwart & Christmann (2008), is identical to the eigenvalue of the bounded linear operator if the bounded linear operator is compact, self-adjoint and positive.

- For $\beta$-finite spectrum, we fix a $\delta \in (0, 1)$ and an $\varepsilon \in (0, 128]$, and assume the integer $i \geq 1$ satisfies the condition:
$$128(1 + \delta)(32\beta + 16)^{-(i+1)/\beta} \leq \varepsilon \leq 128(1 + \delta)(32\beta + 16)^{-i/\beta}.$$
  By the definition of the entropy number and covering number, we know
  $$\begin{aligned}
  &\log \mathcal{N}(\mathcal{B}_{\mathcal{H}_{\tilde{k}}}, \|\cdot\|_{L_2(\mu)}, \varepsilon) \\
  &\leq \log \mathcal{N}(\mathcal{B}_{\mathcal{H}_{\tilde{k}}}, \|\cdot\|_{L_2(\mu)}, 128(1 + \delta)(32\beta + 16)^{-(i+1)/\beta}) \\
  &\leq i \log(2) \\
  &\leq \beta \log(2) \log\left(\frac{128(1 + \delta)}{\varepsilon}\right) \\
  &\leq \beta \log(2) \log\left(\frac{256}{\varepsilon}\right) = O\left(\beta \log(1/\varepsilon)\right)
  \end{aligned}$$

- For $\beta$-polynomial decay, with Lemma 6.21 in (Steinwart & Christmann, 2008), we have that
$$\log \mathcal{N}(\mathcal{B}_{\mathcal{H}_{\tilde{k}}}, \|\cdot\|_{L_2(\mu)}, \varepsilon) \leq \log(4) \left(\frac{128 C_0 (32 + 8\beta)^{\frac{\beta}{2}}}{\varepsilon}\right)^{1/\beta} = O\left(C_{\text{poly}} \varepsilon^{-1/\beta}\right).$$

- For $\beta$-exponential decay, we fix a $\delta \in (0, 1)$ and an $\varepsilon \in (0, 128 C_1]$, and assume the integer $i \geq 1$ satisfies the condition
$$128 C_1 (1 + \delta) \exp(-C_2 (i + 1)^\beta) \leq \varepsilon \leq 128 C_1 (1 + \delta) \exp(-C_2 i^\beta).$$
  By the definition of the entropy number and covering number, we know
  $$\begin{aligned}
  &\log \mathcal{N}(\mathcal{B}_{\mathcal{H}_{\tilde{k}}}, \|\cdot\|_{L_2(\mu)}, \varepsilon) \\
  &\leq \log \mathcal{N}(\mathcal{B}_{\mathcal{H}_{\tilde{k}}}, \|\cdot\|_{L_2(\mu)}, 128 C_1 (1 + \delta) \exp(-C_2 (i + 1)^\beta)) \\
  &\leq i \log(2) \\
  &\leq \log(2) \left(\frac{\log\left(\frac{128 C_1 (1+\delta)}{\varepsilon}\right)}{C_2}\right)^{1/\beta} \\
  &\leq \log(2) \left(\frac{\log\left(\frac{256 C_1}{\varepsilon}\right)}{C_2}\right)^{1/\beta} = O\left(C_{\exp} \log(1/\varepsilon)^{1/\beta}\right),
  \end{aligned}$$
  where $C_3$ is a constant depends on $C_1$ and $C_2$.

Note that $n \leq N$. Hence, we can choose $\varepsilon$ for different eigendecay conditions and lead to the first claim as follows:

- For $\beta$-finite spectrum: we choose $\varepsilon = \Theta(n^{-1})$, and obtain the first claim with $\lambda = \Theta\left(\beta \log N + \log(N/\delta)\right)$ using a union bound over $\mathcal{B}_\varepsilon$ and $[N]$.

- For $\beta$-polynomial decay: we choose $\varepsilon = \Theta(n^{-\beta/(1+\beta)})$, and obtain the first claim with $\lambda = \Theta\left(C_{\text{poly}} N^{1/(1+\beta)} + \log(N/\delta)\right)$ using a union bound over $\mathcal{B}_\varepsilon$ and $[N]$.

- For $\beta$-exponential decay: we choose $\varepsilon = \Theta(n^{-1})$, and obtain the first claim with $\lambda = \Theta\left(C_{\exp}(\log N)^{1/\beta} + \log(N/\delta)\right)$ using a union bound over $\mathcal{B}_\varepsilon$ and $[N]$.

For the second claim, note that,
$$\begin{aligned}
\left\langle x, \left(n\widetilde{\Sigma}_n + \lambda T_k^{-1}\right) x \right\rangle &= \left\langle T_k^{-1/2} x, T_k^{1/2} \left(n\widetilde{\Sigma}_n T_k^{1/2} + \lambda T_k^{-1}\right) T_k^{1/2} T_k^{-1/2} x \right\rangle, \\
\left\langle x, \left(\sum_{i=1}^n \phi_i \phi_i^\top + \lambda T_k^{-1}\right) x \right\rangle &= \left\langle T_k^{-1/2} x, T_k^{1/2} \left(n \sum_{i=1}^n \phi_i \phi_i^\top + \lambda T_k^{-1}\right) T_k^{1/2} T_k^{-1/2} x \right\rangle.
\end{aligned}$$

Note that, $\{T_k^{-1/2}x, x \in \mathcal{H}_k\}$ spans the $L_2(\mu)$, when the first claim holds, we have that, $\forall x' \in L_2(\mu)$, $\forall n \in [N]$

$$\frac{1}{c_2} \left\langle x', T_k^{-1/2}\left(n\Sigma + \lambda T_k\right)^{-1} T_k^{-1/2}x' \right\rangle_{L_2(\mu)} \le \left\langle x', T_k^{-1/2}\left(\sum_{i\in[n]} \phi_i\phi_i^\top + \lambda T_k^{-1}\right)^{-1} T_k^{-1/2}x' \right\rangle_{L_2(\mu)},$$

and

$$\left\langle x', T_k^{-1/2}\left(\sum_{i\in[n]} \phi_i\phi_i^\top + \lambda T_k^{-1}\right)^{-1} T_k^{-1/2}x' \right\rangle_{L_2(\mu)} \le \frac{1}{c_1} \left\langle x', \left(n\Sigma + \lambda T_k^{-1}\right)^{-1} T_k^{-1/2}x' \right\rangle_{L_2(\mu)}.$$

As $\forall x \in \mathcal{H}_k$, $T_k x \in L_2(\mu)$ and we can choose $x' = T_k x$, which shows the second claim holds when the first claim holds. $\qquad\square$

*Remark* 9. Here we follow the idea of Zanette et al. (2021, Lemma 45) and present a less involved proof. However, it is also possible to use the Bernstein inequality for matrix martingale with intrinsic dimension (e.g. Minsker, 2017) to prove the similar results.

**Lemma 18** (Simulation Lemma). *Suppose we have two MDP instances* $\mathcal{M} = (\mathcal{S}, \mathcal{A}, P, r, d_0, \gamma)$ *and* $\mathcal{M}' = (\mathcal{S}, \mathcal{A}, P', r + b, d_0, \gamma)$. *Then for any policy* $\pi$, *we have that*

$$V_{P',r+b}^\pi - V_{T,r}^\pi = \frac{1}{1-\gamma} \mathbb{E}_{(s,a)\sim d_P^\pi}\left[b(s,a) + \gamma\left[\mathbb{E}_{P'(s'|s,a)}[V_{P',r+b}^\pi(s')] - \mathbb{E}_{P(s'|s,a)}[V_{P',r+b}^\pi(s')]\right]\right],$$

$$V_{P',r+b}^\pi - V_{T,r}^\pi = \frac{1}{1-\gamma} \mathbb{E}_{(s,a)\sim d_{P'}^\pi}\left[b(s,a) + \gamma\left[\mathbb{E}_{P'(s'|s,a)}[V_{T,r}^\pi(s')] - \mathbb{E}_{P(s'|s,a)}[V_{T,r}^\pi(s')]\right]\right].$$

*Proof.* See Uehara et al. (2022, Lemma 20). $\qquad\square$

**Lemma 19** (Potential Function Lemma for RKHS). *If* $\alpha = \Omega(1)$, *then for any distribution* $\nu$ *supported on the unit ball of* $\mathcal{H}_{\tilde{k}}$, *we have that,*

- *For* $\beta$-*finite spectrum:*

$$\log\det\left(\alpha\mathbb{E}_\nu[\phi\phi^\top] + I\right) = O\left(\beta\log\left(1 + \frac{\alpha}{\beta}\right)\right).$$

- *For* $\beta$-*polynomial decay:*

$$\log\det\left(\alpha\mathbb{E}_\nu[\phi\phi^\top] + I\right) = O\left(C_{\text{poly}}\alpha^{1/(2\beta)}\log\alpha\right).$$

- *For* $\beta$-*exponential decay:*

$$\log\det\left(\alpha\mathbb{E}_\nu[\phi\phi^\top] + I\right) = O\left(C_{\text{exp}}(\log\alpha)^{1+1/\beta}\right).$$

*where operators are in the space of* $L_2(\mu) \to L_2(\mu)$.

*Meanwhile, when* $\alpha = O(1)$, *for any eigendecay conditions, we have that*

$$\log\det\left(\alpha\mathbb{E}_\nu[\phi\phi^\top] + I\right) = O(1).$$

*Proof.* We consider the optimization problem:

$$\sup_\nu \log\det\left(I + \alpha\mathbb{E}_{\phi\sim\nu}\left[\phi\phi^\top\right]\right).$$

We first consider the optimality condition of $\nu$. Note that, $\log\det(X)$ is concave with respect to positive definite $X$ and $\mathbb{E}_{\phi\sim\nu}\left[\phi\phi^\top\right]$ is linear with respect to $\nu$. Direct computation shows that

$$\frac{d\log\det\left(I + \alpha\mathbb{E}_{\phi\sim\nu}\left[\phi\phi^\top\right]\right)}{d\nu(\phi')} = \text{Tr}\left(\alpha\left(I + \alpha\mathbb{E}_{\phi\sim\nu}\left[\phi\phi^\top\right]\right)^{-1}\phi\phi^\top\right)$$

$$= \left\langle \phi', \left(\alpha^{-1}I + \mathbb{E}_{\phi\sim\nu}\left[\phi\phi^\top\right]\right)^{-1}\phi' \right\rangle_{L_2(\mu)}.$$

Note that, $\left(\alpha^{-1}I + \mathbb{E}_{\phi \sim \nu}\left[\phi\phi^{\top}\right]\right)^{-1} \preceq \alpha I$. Hence, the inner product is well-defined. As $\nu$ is a probability measure over the $\mathcal{B}_{\mathcal{H}_{\widetilde{k}}}$, and if $c \geq 1$,

$$\left\langle c\phi', \left(\alpha^{-1}I + \mathbb{E}_{\phi \sim \nu}\left[\phi\phi^{\top}\right]\right)^{-1} c\phi'\right\rangle_{L_2(\mu)}$$

$$= c^2 \left\langle \phi', \left(\alpha^{-1}I + \mathbb{E}_{\phi \sim \nu}\left[\phi\phi^{\top}\right]\right)^{-1} \phi'\right\rangle_{L_2(\mu)}$$

$$\geq \left\langle \phi', \left(\alpha^{-1}I + \mathbb{E}_{\phi \sim \nu}\left[\phi\phi^{\top}\right]\right)^{-1} \phi'\right\rangle_{L_2(\mu)}.$$

Hence, we can focus on the $\nu$ supported on $\mathcal{S}_{\mathcal{H}_{\widetilde{k}}}$. Furthermore, with the optimality condition of the probability measure, we know the optimal $\nu$ should satisfy that $\forall \phi' \in \operatorname{supp}(\nu)$,

$$\left\langle \phi', \left(\alpha^{-1}I + \mathbb{E}_{\phi \sim \nu}\left[\phi\phi^{\top}\right]\right)^{-1} \phi'\right\rangle_{L_2(\mu)} = C,$$

and $\forall \phi' \in \mathcal{S}_{\mathcal{H}_{\widetilde{k}}}$, we have

$$\left\langle \phi', \left(\alpha^{-1}I + \mathbb{E}_{\phi \sim \nu}\left[\phi\phi^{\top}\right]\right)^{-1} \phi'\right\rangle_{L_2(\mu)} \leq C,$$

where $C$ is some constant.

We first show that, $C \geq \frac{\alpha\mu_1(T_k)^2}{\alpha\mu_1^2(T_k)+1}$, which can be shown by consider the following constraint optimization problem

$$\inf_{\nu} \left\langle \phi', \left(\frac{\lambda}{n}I + \mathbb{E}_{\phi \sim \nu}\left[\phi\phi^{\top}\right]\right)^{-1} \phi'\right\rangle_{L_2(\mu)},$$

where $\nu$ is from the space of probability measure supported on $\mathcal{S}_{\mathcal{H}_{\widetilde{k}}}$. As $\langle x, A^{-1}x\rangle$ is convex with respect to positive definite $A$ and $\mathbb{E}_{\phi \sim \nu}[\phi\phi^{\top}]$ is linear with respect to $\nu$, straightforward computation shows that

$$\frac{d\left\langle \phi', \left(\frac{\lambda}{n}I + \mathbb{E}_{\phi \sim \nu}\left[\phi\phi^{\top}\right]\right)^{-1} \phi'\right\rangle_{L_2(\mu)}}{d\nu\left(\widetilde{\phi}\right)} = -\left\langle \phi', \left(\frac{\lambda}{n}I + \mathbb{E}_{\phi \sim \nu}\left[\phi\phi^{\top}\right]\right)^{-1} \widetilde{\phi}\right\rangle_{L_2(\mu)}^2.$$

With the optimality condition, the optimal $\nu$ should satisfy that $\forall \phi \in \operatorname{supp}(\nu)$,

$$\left\langle \phi', \left(\alpha^{-1}I + \mathbb{E}_{\phi \sim \nu}\left[\phi\phi^{\top}\right]\right)^{-1} \widetilde{\phi}\right\rangle_{L_2(\mu)}^2 = C'.$$

and $\forall \widetilde{\phi} \in \mathcal{S}_{\mathcal{H}_{\widetilde{k}}}$, we have

$$\left\langle \phi', \left(\alpha^{-1}I + \mathbb{E}_{\phi \sim \nu}\left[\phi\phi^{\top}\right]\right)^{-1} \widetilde{\phi}\right\rangle_{L_2(\mu)}^2 \leq C',$$

where $C'$ is an absolute constant. With Cauchy-Schwartz inequality, we have

$$\left\langle \phi', \left(\alpha^{-1}I + \mathbb{E}_{\phi \sim \nu}\left[\phi\phi^{\top}\right]\right)^{-1} \widetilde{\phi}\right\rangle_{L_2(\mu)}^2$$

$$\leq \left\|T_k \left(\alpha^{-1}I + \mathbb{E}_{\phi \sim \nu}\left[\phi\phi^{\top}\right]\right)^{-1} \phi'\right\|_{L_2(\mu)} \|T_k^{-1}\widetilde{\phi}\|_{L_2(\mu)}$$

$$= \left\|T_k \left(\alpha^{-1}I + \mathbb{E}_{\phi \sim \nu}\left[\phi\phi^{\top}\right]\right)^{-1} \phi'\right\|_{L_2(\mu)},$$

where the maximum only achieves when

$$\phi' = c' \left(\alpha^{-1}I + \mathbb{E}_{\phi \sim \nu}[\phi\phi^{\top}]\right) T_k^{-2}\widetilde{\phi}.$$

where $c'$ is an absolute constant to make sure $\|\widetilde{\phi}\|_{\mathcal{H}_{\widetilde{k}}} = 1$. Hence, the optimal $\nu$ is a point measure supported on $\widetilde{\phi}$, which further leads to

$$\left(\alpha^{-1}I + \widetilde{\phi}\widetilde{\phi}^{\top}\right) T_k^{-2}\widetilde{\phi} = \left(\alpha^{-1}T_k^{-2} + I\right) \widetilde{\phi},$$

Take $\phi' = \mu_1(T_k)e_1$, as $e_i$ is the eigenfunction of $T_k^{-2}$, we know $\nu$ should only support on $\mu_1(T_k)e_1$, and

$$\inf_{\nu} \left\langle \phi', \left(\alpha^{-1}I + \mathbb{E}_{\phi \sim \nu}\left[\phi\phi^{\top}\right]\right)^{-1} \phi'\right\rangle_{L_2(\mu)} = \frac{\alpha\mu_1^2(T_k)}{\alpha\mu_1^2(T_k)+1},$$

which means $C \geq \frac{\alpha\mu_1(T_k)^2}{\alpha\mu_1^2(T_k)+1}$.

Now we consider the constraint optimization problem

$$\max_{\phi'} \left\langle \phi', \left(\alpha^{-1}I + \mathbb{E}_{\phi \sim \nu}\left[\phi\phi^\top\right]\right)^{-1}\phi' \right\rangle, \quad \text{s.t.} \quad \|\phi'\|_{\mathcal{H}_{\tilde{k}}} \leq 1.$$

With the method of Lagrange multiplier, we know that $CT_k^{-2} - \left(\alpha^{-1}I - \mathbb{E}_{\phi \sim \nu}\left[\phi\phi^\top\right]\right)^{-1} \succeq 0$, and for all $\phi$ in the support of $\nu$, we have $\left(CT_k^{-2} - \left(\alpha^{-1}I + \mathbb{E}_{\phi \sim \nu}\left[\phi\phi^\top\right]\right)^{-1}\right)\phi = 0$. Note that $\left\|\left(\alpha^{-1}I + \mathbb{E}_{\phi \sim \nu}\left[\phi\phi^\top\right]\right)^{-1}\right\|_{\text{op}} \leq \alpha$. With Weyl's inequality, we know that,

$$\mu_i\left(CT_k^{-2} - \left(\alpha^{-1}I - \mathbb{E}_{\phi \sim \nu}\left[\phi\phi^\top\right]\right)^{-1}\right) \geq C\mu_i(T_k)^{-2} - \alpha \geq \alpha\left(\frac{\mu_1^2(T_k)\mu_i^{-2}(T_k)}{\alpha\mu_1^2(T_k) + 1} - 1\right),$$

which means the support of $\nu$ is at most $i_0$ dimension, where $i_0$ is the largest integer that $\mu_1^2(T_k)\mu_i^{-2}(T_k) \leq \alpha\mu_1^2(T_k) + 1$.

When $\alpha = O(1)$, with Assumption 3, we know $\mu_i(T_k) \leq 1$ and $i_0 = O(1)$. Combined with Jensen's inequality, we finish the proof of the second claim.

We then consider the case when $\alpha = \Omega(1)$ under different eigendecay conditions:

- $\beta$-finite spectrum: we know $i_0 \leq \beta$. As $\|\phi\|_{L_2(\mu)} \leq \|\phi\|_{\mathcal{H}_{\tilde{k}}} = 1$, we have $\left\|\mathbb{E}_{\phi \sim \nu}\left[\phi\phi^\top\right]\right\|_{\text{op}} \leq 1$. With Jensen's inequality, we have

$$\log\det\left(I + \alpha\mathbb{E}_{\phi \sim \nu}\left[\phi\phi^\top\right]\right) = O\left(\beta\log\left(1 + \frac{\alpha}{\beta}\right)\right).$$

- $\beta$-polynomial decay: we know $i_0 = O\left(C_{\text{poly}}\alpha^{1/(2\beta)}\right)$ dimension. As $\|\phi\|_{L_2(\mu)} \leq \|\phi\|_{\mathcal{H}_{\tilde{k}}} = 1$, we have $\left\|\mathbb{E}_{\phi \sim \nu}\left[\phi\phi^\top\right]\right\|_{\text{op}} \leq 1$. With Jensen's inequality, we have

$$\log\det\left(I + \alpha\mathbb{E}_{\phi \sim \nu}\left[\phi\phi^\top\right]\right) = O\left(C_{\text{poly}}\alpha^{1/(2\beta)}\log\alpha\right).$$

- $\beta$-exponential decay: we know the support of $\nu$ is at most $O\left(C_{\text{exp}}(\log\alpha)^{1/\beta}\right)$ dimension. As $\|\phi\|_{L_2(\mu)} \leq \|\phi\|_{\mathcal{H}_{\tilde{k}}} = 1$, we have $\left\|\mathbb{E}_{\phi \sim \nu}\left[\phi\phi^\top\right]\right\|_{\text{op}} \leq 1$, With Jensen's inequality, we have

$$\log\det\left(I + \alpha\mathbb{E}_{\phi \sim \nu}\left[\phi\phi^\top\right]\right) = O\left(C_{\text{exp}}(\log\alpha)^{1+1/\beta}\right).$$

Hence, we obtain the desired results. $\qquad\square$

## G ADDITIONAL EXPERIMENT RESULT

### G.1 TRAINING WITH 1M STEPS

This section provides the learning curves with 1M training steps compared to SAC in four Mujoco control problems. We only tune the *feature-updates-per-step* parameter from $\{1, 3, 5\}$ and report the best result to save computations and running time. The results clearly demonstrate that LV-Rep also achieves significantly better performance in the long run.

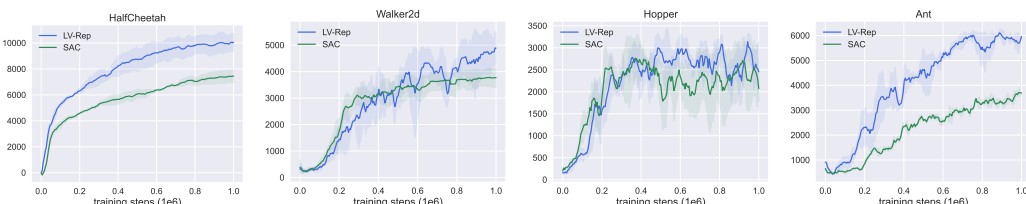

Figure 3: We show the learning curves in Mujoco control compared to SAC. The $x$-axis shows the training iterations and $y$-axis shows the performance. All plots are averaged over 4 random seeds. The shaded area shows the standard error.

## G.2 ABLATION STUDY ON LATENT REPRESENTATION SIZE

In this section, we provide an ablation study on the latent representation dimension to show this parameter affects the performance of LV-Rep. In all our experiments the latent feature dimension is set to 256. We compare to latent feature dimension 64 and 128 in HalfCheetah. The results are reported in Figure 4.

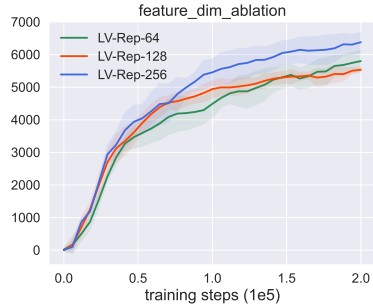

Figure 4: Ablation study on the dimension of latent representations.

