# OpenReview forum: "Latent Variable Representation for Reinforcement Learning"
_ICLR.cc/2023/Conference — ICLR 2023 poster_

### Official Review · Reviewer_9PEm · 2022-10-24

**Confidence:** 2
**Correctness:** 2
**Technical Novelty And Significance:** 3
**Empirical Novelty And Significance:** 2
**Recommendation:** 3

**Clarity, Quality, Novelty And Reproducibility:**

Clarity (minor points):
- Please add a citation when mentioning MBBL for the first time
- How is the unsupervised pre-training phase for Proto-RL handled in the experiments?

Reproducibility:
- A source code release would be great; based on the claimed contributions, easy reproducibility and re-use would be very beneficial to the RL community.

**Strength And Weaknesses:**

Honestly, I had a hard time reading this paper and believe that my understanding is somewhat limited (cf confidence score) due to scarce exposure to relevant prior work. I'll hence focus my review on the experimental section and am happy to adjust my score in the rebuttal if necessary.

The paper opens with a (quite bold) claim that the proposed algorithm remedies the computation difficulties of RL; in particular, the authors claim sample complexity (I assume this is what is referred to with "statistical complexity") independent of the number of states. While the theoretical analysis suggests that this is the case, I would have liked to see this demonstrated in experiments more clearly.

The experiments themselves are performed with common MuJoCo tasks, either as defined in gym or in dm_control. Generally, I think that learning curves for up to 1M steps should at least be shown in the Appendix; the plotted learning curves clearly indicate that further learning progress can be anticipated (according to the MBBL paper). The MBBL paper mentions that they limited their experiments to 200k steps since model-based methods are often slow to run and converge much earlier than model-free algorithms. What is the wall-clock time in your case? Besides, assuming that the authors ran the Dreamer and Proto-RL baselines themselves, could you add them to plotted the learning curves?

For Table 1, maybe the authors could provide more exhaustive evaluation by training on both ET and non-ET variants, for the sake of completeness?

Re the dm_control experiments: my understanding is that `hopper_hop` and `humanoid_run` are not sparse-reward; the writing should be adjusted accordingly.

**Summary Of The Paper:**

This submission proposes a new model-based RL method grounded in representation learning. It seems like the central idea is to impose linearity on the transition function, which is formulated as the dot product between $p(z|s.a)$ and $p(s'|z)$, which are both parameterized as neural networks.

**Summary Of The Review:**

As admitted above, this is a low-confidence review. However, I don't see the claims made in the introduction that suggest a fundamentally more efficient RL algorithm supported by the evidence presented in the experimental section. Yes, the learning curves are equal or better than SAC, but evaluation is limited to the first 200k environment steps. I think that the paper could definitely benefit from more exhaustive experimental analysis, even if it's placed in the Appendix.

---

> ### Author Response · Authors · 2022-11-16
> **Author Response to Reviewer 9PEm**
>
> We thank reviewer 9PEm for the suggestions on the experiments. We would like to address the concerns in the following:
>
> * **Sample complexity independent w.r.t. number of states**
>
>   As we show in Theorem 1, the sample complexity (the number of samples needed to be PAC-optimal) only depends on the effective dimension of the latent variable space. This is especially important towards better understanding the successes of RL in practical problems with extremely large or even infinite state space. Concretely, the vanilla statistical bound, which depends on the number of states (potentially infinite), would suggest vacuous sample complexity upper bounds, and thus, fails to explain any empirical success of RL algorithms.
>
>   To this end, our contributions indeed are coherent in both theoretical and empirical aspects: the proposed algorithm performs significantly better in Mujoco locomotion control problems, where the number of states is infinite due to continuous state space, and our theory already provides a good explanation of why the algorithm can work in practice.
>
> * **Experimental suggestion with 1M steps, ET and non-ET**
>
>   Thanks for the suggestion. We follow the **exact** experimental setup with MBBL [1] baselines for a fair comparison, with the same number of steps and termination criteria. As we can see, even with a few interactions, our algorithm already demonstrates the significant advantages w.r.t. the existing SoTA model-based and model-free RL algorithms, which emphasizes the efficiency of our algorithm in terms of both sample complexity.
>
>   Finally, we also provide the learning curves for up to 1M steps in the revision (Figure 3 in Appendix G.1), in which our algorithm shows even more advantages.
>
> * **Adding Dreamer and Proto-RL curves**
>
>   Thanks for the suggestion. We have added them to the curves. Please see Figure 2 for the detail.
>
> * **Sparse reward for Deepmind Control Suite environments**
>
>   We thank the reviewer for pointing this out. For these two environments, although the reward is not strictly sparse (e.g., the agent gets the feedback at the end of an episode), the reward itself is relatively sparse. The agent often will not receive any reward at most steps during the episode. There are some prior works that also categorize these environments as sparse-reward [2]. That’s the reason we call it a sparse environment.
>
> * **On the unsupervised pre-training phase of Proto-RL**:
>
>   The original Proto-RL paper uses the pretraining phase for 500k steps, in a total of 1M steps. In our experiments, since we only ran 200k steps in total, we ran the pretraining phase for 100k steps. We have added the details in our revision.
>
> > [1] Wang, Tingwu, et al. "Benchmarking model-based reinforcement learning." arXiv preprint arXiv:1907.02057 (2019). \
> [2] Zhang, Tianjun, et al. "Making linear mdps practical via contrastive representation learning." International Conference on Machine Learning. PMLR, 2022.

---

> ### Author Response · Authors · 2022-11-18
> **Look forward to your feedback.**
>
> Dear Reviewer 9PEm,
>
> We have addressed your concerns in our author response and updated our manuscript accordingly. We would like to hear your feedback. If there are any further concerns, please let us know and we will try our best to address them during the discussion period. If there are no further concerns, we would like to sincerely ask for a re-evaluation for our manuscript. Thanks!
>
> Best Regards,
>
> Authors of Paper 3443

---

### Official Review · Reviewer_F81s · 2022-10-25

**Confidence:** 3
**Correctness:** 3
**Technical Novelty And Significance:** 3
**Empirical Novelty And Significance:** 3
**Recommendation:** 6

**Clarity, Quality, Novelty And Reproducibility:**

+ This paper is well clarified.
+ To my knowledge, such a theoretical and methodological connection between latent variable models and linear MDPs seems new. This would be of interest to the community.
+ The current version might be not sufficient to reproduce the results presented in the paper.

**Strength And Weaknesses:**

**Strength:**

+ The paper is well organised and well written.
+ The idea is reasonable and also well supported both theoretically and empirically.

**Weaknesses:**

I only have some minor concerns:

+ Theorem 1 shows that the sample complexity largely depends on $|\mathcal{A}|$. This seemingly implies that the proposed approach might struggle in the environments with continuous action space. Is it right?
+ I understand that the factorization in (3) is consistent with the linear MDP setting. I am wondering how this could be extended to the nonlinear MDP setting? In my opinion, a more straightforward and general factorization should be $T^*(s'|s, a)=\int_{\mathcal{Z}}\int_{\mathcal{Z}} p^*(z|s)p^*(z'|z,a)p^*(s'|z')dzdz'$, which explicitly models an MDP over the latent space. Is there any insight over this?

**Summary Of The Paper:**

This paper provides a representation view of latent variable models in linear MDPs, and further propose a computationally efficient algorithm to implement it for both online and offline RL. The authors also theoretically and empirically demonstrate the proposed approach.

**Summary Of The Review:**

The paper builds a nice bridge between latent variable models and linear MDPs, with well supports on both theory and practice.

---

> ### Author Response · Authors · 2022-11-16
> **Author Response to Reviewer F81s**
>
> We thank reviewer F81s for the positive feedback. We address the concerns in the following:
>
> * **Dependency on $|\mathcal{A}|$**: This is still an open problem in the theoretical community. In fact, *all of the existing theoretical analysis* has this dependency on $|\mathcal{A}|$. For continuous action space, we can replace the $|\mathcal{A}|$ with the volume of the action space, and the theoretical analysis still follows. Empirically, our algorithm can still work on benchmarks with continuous action space. We leave the theoretical analysis without explicit dependency on the cardinality of action space as a future work.
>
> * **Alternative formulation**: It is indeed possible to model the environment with the latent MDP model such that $T^*(s^\prime|s, a) = \int_{\mathcal{Z}\times \mathcal{Z}} p^*(z|s) p(z^\prime|z, a) p(s^\prime|z^\prime) ds ds^\prime$ where $\mathcal{Z}$ is a latent state space. However, even with this factorization, the effective representation is still $p(z^\prime|s, a)$ obtained by integrating out $z$, upon which we can represent the $Q$ function as a linear function.

---

### Official Review · Reviewer_Y1QJ · 2022-10-25

**Confidence:** 3
**Clarity, Quality, Novelty And Reproducibility:** (see general comments)
**Correctness:** 4
**Technical Novelty And Significance:** 3
**Empirical Novelty And Significance:** 3
**Recommendation:** 8

**Strength And Weaknesses:**

First of all, the paper is very dense, providing both theoretical results and practical algorithms. As a consequence, it is sometimes difficult to understand exactly what the contributions presented are -- for instance in Section 4, the theoretical analysis is quite short (regarding the dozens of appendix pages) and it is hard to understand what to conclude exactly on the approach from the provided theorem which is very hard to catch without providing more explanations. Similarly, when describing the algorithms, many different aspects are still unclear to me (and not better detailed in the appendix). So I have mixed feelings about the article, and I think that the authors have to update the writing to make the paper more pedagogical and to clearly state the critical aspects of their approach. This is the type of paper that better fits in a journal than in a conference.

Here are some concrete questions:
Page 4: Connection to Ren: many different notions arise in this section like representation complexity, We understand that it is related to sample complexity (and Section 4). The last sentence "The LV-Rep with....and learning" is completely unclear at this moment in the article. I would suggest moving this paragraph into Section 4 instead since, at this point, it does not really bring relevant information to the reader, and make the message unclear.
Concerning the ELBO approach which defines a variational encoder for the transitions, this approach has already been proposed in different papers like "Temporal Difference Variational Auto-Encoder - Karol Gregor et al." for instance. Can you discuss the differences between this paper for instance ?  Is it really new ?
On page 5, the w^pi(z) notation appears without any clear definition. If we understand that it is a rewriting of the value function by using the latent representation, it would be nice to give a clear definition, and to comment a little bit more.
The use of random feature representation (described in the appendix only) is crucial to understand how you handle continuous representations. And the paper does not give clear definitions of this particular aspect. I would recommend providing more explanations about this point directly in the paper. It is not clear for instance how you choose the value of 'm', the distribution P(xsi_i) and what is the impact of these choices (Note that, in the experimental section, these choices are not detailed).
Equation 8: the bellman equation is rewritten with an exploration bonus. This equation is connected to the notion of 'planning' in the paper and in the algorithm where (line 7) a new policy is obtained by planning using the exploration bonus. I don't understand exactly how this step is done. Moreover, in Section 6, you state that SAC is used as your planner in LV-Rep. Maybe it is just a matter of semantics, but in my mind, SAC is not a planning algorithm. Can you make this point more clear ?  I think that depending on the background of the reader, the use of the term planning can be misleading.
Section 4 is actually too short to capture the theoretical contribution you are making in the paper. For a reader which is not specialized in theory, I would suggest better explain the content of Theorem 1, and providing a small insights about the steps to obtain such a theorem
On the experimental side, I would be happy to have more information about the impact of the representation size on the performance. Experiments with a high-dimensional space would also be interesting to evaluate if the approach is also able to capture a relevant representation space when observations are in high dimension -- this is not a critical remark but would strengthen the paper.


**Summary Of The Paper:**

The article proposes a new RL algorithm where the idea is to model the dynamics of the system by using a latent representation space. Using a latent representation to capture the dynamics, it allows one to use
variational methods (and ELBO optimization) for doing representation learning over the MDP (Section 3). This is done by connecting latent representation models to Linear MDPs. In Section 3.1, the authors then use the maximization of ELBO approach to learning representations while simultaneously learning the policy (see Algorithm 1). The proposed approach is to alternate between representation learning steps and policy updates, these policy updates being made by planning on the learned model. One of the main points in the algorithm is to add an exploration bonus similar to the one proposed in REP-UCB. Section 4 provides a theoretical analysis of the algorithm. Section 5 describes experimental results obtained in the online RL setting over different classical control benchmarks. In that section, the proposed approach (with both discrete and continuous representations) is compared to classical RL (model-free and model-based) methods. It demonstrates a high efficiency on many of the tasks.


**Summary Of The Review:**

In conclusion, the paper explores the use of a VAE as a representation model learned together with the policy. One of the interests of the VAE is that it allows the definition of an exploration bonus to guide the learning, which is certainly the critical reason for the good performance of the approach. The paper is interesting to the community, with theoretical and practical contributions (I did not check the proofs), but the writing makes the paper quite hard to follow.

---

> ### Author Response · Authors · 2022-11-16
> **Author Response to Reviewer Y1QJ**
>
> We thank reviewer Y1QJ for the detailed feedback and suggestions on the paper organization. We address the main concerns below:
>
> * **On the connection to Ren et al. 2022**: Here we would like to provide a concrete example as the special case for the proposed LV-Rep, to demonstrate the generality and versatility of the proposed representation. We have revised our manuscript to make this purpose more clear.
>
> * **Variational approaches and the difference between the proposed methods and Gregor et al. 2018**: Thanks for pointing out the references. We want to remark that the variational approaches for (conditional) density estimation have been thoroughly studied in the literature under different probabilistic models, and we didn’t claim the variational approaches used for the estimation of LV-Rep is novel. Instead, we want to show that LV-Rep can be easily estimated through the variational approaches, which is in sharp contrast to the previous representation learning approaches, which rely on the intractable MLE.
>
>   Comparing to Gregor et al. 2018, there are also substantial differences. Specifically, we considered the sequential decision making problems under certain MDP assumptions while Gregor et al. focused on the sequential data modeling. In our model, given $(s_t, a_t)$, $z_t$ is independent of $z_{t-1}$ while in Gregor et al., $z_t$ still depends on $z_{t-1}$. We focused on the representation view of latent variable models in reinforcement learning and how we can leverage this view for provably efficient exploration and exploitation, while Gregor et al. focused on the sequential data generation without these considerations.
>
> * **The definition of $w^\pi$**: Sorry for the confusion here. With Equation 1, we define $w^\pi(z) = \int p^*(s^\prime|z) V_{T^*, r}^\pi(s^\prime) ds^\prime$. It can be viewed as the value function on the latent state. We have added the definition in our revised version.
>
> * **On the random feature approximation**: We thank the reviewer for the suggestion on improving the clarity. We have a preliminary introduction to the random Fourier feature (RFF) included in Appendix, due to the space limitation. We will reorganize the manuscript  and use the one additional page for more details about RFF after it gets accepted. For more detailed information, please refer to (Rahimi and Recht, 2009, Dai et al, 2014).
>
>   For your question,  the probability density $P(\xi)$ used in the random feature is determined by the kernel from the model. For example, the Gaussian kernel will induce $P(\xi)$ as Gaussian.
>
>   In general, the larger $m$ (number of the random feature), the smaller approximation error. In the experiments we use $m = 256$.
>
> * **Planning in MDP**: Here we refer to planning as the procedure of finding the optimal policy upon the MDP (https://rltheory.github.io/lecture-notes/planning-in-mdps/lec1/#:~:text=Computing%20an%20optimal%20policy%20can,the%20algorithmic%20question%20becomes%20interesting!), which is what the SAC algorithm is doing. Empirically, in our implementation, we run SAC with the Q function represented by the learned feature upon the collected data.
>
> * **Presentation of the theoretical results**: Thanks for the suggestion.Theorem 1 shows our algorithm can obtain an $\epsilon$-optimal policy with high probability with the number of samples we collect having at most polynomial dependency with the terms of interest.The sample complexity only depends on the complexity of the representation space, without an explicit dependency on the size of the state space, which justifies the benefits of representation learning.   We will add more discussion using the extra page once the paper is accepted.
>
> * **Ablation on representation size**: An ablation study on representation size is provided in Appendix G.2. We compare LV-Rep using latent representation size 256 (our default setting) to LV-Rep with representation size 64 and 128. The results show that using representations with higher dimensions improves the learning performance in general.
>
> * **High-dimension Extension**: Thanks for the suggestion. The focus of the current paper is to demonstrate the benefits of latent variable representation for MDP setting from both theoretical and empirical aspects, as the first step. We believe the POMDP is more appropriate for modeling environments with high-dimension images, and how to extend the LV-Rep for POMDP is our on-going work.
>
> > [1] Rahimi, Ali, and Benjamin Recht. "Random features for large-scale kernel machines." Advances in neural information processing systems 2007. \
> [2] Dai, Bo, et al. "Scalable kernel methods via doubly stochastic gradients." Advances in neural information processing systems 27 (2014).

---

### Official Review · Reviewer_F7os · 2022-10-30

**Confidence:** 4
**Correctness:** 3
**Technical Novelty And Significance:** 3
**Empirical Novelty And Significance:** 3
**Recommendation:** 6

**Clarity, Quality, Novelty And Reproducibility:**

The paper is well-written and the difference from past work is quite nicely explained. A pseudocode that more closely resembles the code that was run for the experiments, would significantly improve the clarity.

The paper is closely related to previous work on low-rank MDPs. In particular, the paper is very related to REP-UCB and FLAMBE algorithms in past work, that the paper cites and describes. The proof strategy is also closely related to REP-UCB. Therefore, the novelty is somewhat limited. However, this is fine as the impact of results is more important than the novelty of techniques. A key difference, however, is that the paper presents a nice set of experiments.


**Strength And Weaknesses:**

Strength:

1. Generalization to infinite-dimensional $z$ adds more flexibility, however, there is also constraints on the decomposition restricted to probability measures.
2. PAC guarantees
3. Experimental results showing comparable results

Weakness:

1. The paper seems to make two wrong assertions regarding prior work:

- _"Our definition of LV-Rep is more general than the original definition in Agarwal et al. (2020), which assumes |Z| is finite."_ This seems wrong. While FLAMBE assumes a finite rank, the features are not restricted to probability measures, in particular, features can take negative values. They also argue that the non-negative rank can be much larger than the rank of $T(s' \mid s, a)$. I would like to understand how this trade-off works.

- Regarding block MDP _"p∗(z|s, a) is a deterministic measure supported on the latent state that generates the next observation."_. This is not correct. The block MDP algorithms mentioned also handle stochastic transitions. I believe this is a typo and $p(z \mid s, a)$ is a stochastic measure that generates the next state and $p(s' \mid z)$ is the emission distribution for generating an observation $s'$ based on latent state $z$.

2. It is not clear to me when is the ELBO approach more computationally-efficient than the MLE approach in FLAMBE, etc. that the paper criticizes. Doesn't the ELBO approach also require maximizing over $q$? What is the empirical optimization that is being performed in the code? If this can be described clearly in a pseudocode and the advantage over MLE is more clearly described, then it would be helpful.

3. Representing the Q function requires approximation using kernel or neural network. This requires certain conditions to be satisfied that I am not sure, how easy, it will be to satisfy in practice. Experiments, however, show that this is possible at times which makes this less concerning.


**Summary Of The Paper:**

This paper provides a provable reinforcement learning algorithm for low-rank MDPs. The key difference from past work on this literature is that the transition model $T(s' \mid s, a)$ has a decomposition $<\mu(s'), \phi(s, a)>$ which can be an infinite-dimensional inner product, however, it is required that $\phi$ and $\mu$ are both probability measures. This is viewed as a latent variable model where $\phi(s, a)$ denotes the vector $(p(z \mid s, a))_{z}$ and $\phi(s')$ denotes the vector $(\mu(s' \mid z))_z$. The transition dynamics can then be expressed as $\int p(s' \mid z) p(z \mid s, a) dz.

It is claimed that the above decomposition, in addition to more flexibility due to infinite-dimensional $z$, offers a flexibility of computationally-efficient maximum-likelihood estimation through ELBO and a simple-sampling scheme for $T$ obtained via $z \sim p(\cdot \mid, s, a)$ and $s' \sim p(\cdot \mid z)$. However, infinite-dimensionality of $z$ complicates things as well, in particular, requiring $Q$ function to be approximated using a kernel (or a two-layer neural network).

The presented algorithm LV-REP follows previous work (REP-UCB, FLAMBE), and estimates the model based on data collected so far, and then plans with an optimism-based elliptic bonus to find the next policy to collect data from. PAC results are presented for online learning (and a similar result for the offline case). Experiments are presented on Mujoco tasks showing comparable performance to prior work, while the presented approach also has theoretical guarantees.

**Summary Of The Review:**

I lean towards a weak acceptance. My two main concerns are (i) clarity on how and when is LV-REP MLE more efficient than FLAMBE, (ii) how the trade-off work with FLAMBE that assumes finite-rank but allows the decomposition into $\mu, \phi$ to take negative values, and (iii) the two wrong assertions stated above.

---

> ### Author Response · Authors · 2022-11-16
> **Author Response to Reviewer F7os**
>
> We thank the detailed comments from reviewer F7os. We would like to address the concerns in the following:
>
> * **About the two "wrong" claims:**
> > * **On the comparison of the definition of LV-Rep with Agarwal et al. 2020**: This misunderstanding is caused by a confusion of the terminology. We are referring to the Definition 2 in Agarwal et al. 2020, which is named as LV-Rep in their paper. For general linear/low-rank MDP as defined in Definition 1 in Agarwal et al. 2020, features can take negative values and are not necessarily probabilistic measures, but for the latent variable representation in Definition 2, it indeed requires these two properties. We have revised our manuscript to clarify that.
> > * **On the block MDP instance**: Your understanding is correct and we are sorry for the confusion here. For block MDP, we can view $P(z^\prime|s, a)$ as a composition of deterministic $P(z|s)$ and a potentially stochastic $T(z^\prime|z, a)$ by the definition of block MDP. We have clarified this in the latest manuscript.
>
> * **On the computational efficiency comparison between LV-Rep vs. FLAMBE and RepUCB**:
> For MDP with LV-Rep, by definition, we always have the partition function equal to $1$. Moreover, the latent variable parametrization naturally induces the computational friendly variational ELBO. We would like to kindly remark that variational approaches have been comprehensively investigated in the Bayesian community as a computational tractable alternative for the computational intractable MLE.
> For the original low-rank MDP, as we discussed in Section 2, MLE requires solving a constrained optimization problem, which can be infeasible in practice.  The major difficulty lies in estimating the partition function to the desired accuracy, which can be difficult when the state space is not finite.
>
> * **The empirical optimization performed in the code**: The empirical optimization problem for MLE can be written as $\max_{p(z|s, a), p(s^\prime|z)} \sum_{(s_i, a_i, s_i^\prime)\in \mathcal{D}} \log T(s_i^\prime|s_i, a_i)$. With the variational method, we can perform the empirical optimization via maximizing the following empirical ELBO $ \max_{p(z|s, a), p(s^\prime|z), q(z|s, a, s^\prime)} \sum_{(s_i, a_i, s_i^\prime \in \mathcal{D})} \mathbb{E}_{z\sim q(s_i, a_i, s_i^\prime)} [\log p(s_i^\prime|z)] -  KL(q(z|s_i, a_i, s_i^\prime) \\| p(z|s_i, a_i))$.
>
> * **On the trade-off between rank and non-negative rank**: Agarwal et al. 2020 discussed the trade-off between rank and non-negative rank when both of the terms are **finite**. Specifically, they show that there exist examples that have finite rank and non-negative rank, for which the non-negative rank can be exponentially larger than the rank, which shows a separation between these two terms. However, it is not clear how to generalize this claim for infinite-dimension features with special eigen-decay structures, as the rank counting is no longer applicable. Our focus is on demonstrating the effectiveness of the (potentially infinite) LV-Rep from both theoretical (with novel complexity measure without dependency on the finite rank) and empirical (with comprehensive comparison) sides. The detailed investigation on such infinite structural assumptions, e.g., the expressiveness ability difference with/without non-negativity, is an interesting but separate research topic beyond the scope of our manuscript.
>
> * **On the required condition for representing $Q$ function**:  When $|\mathcal{Z}| < \infty$, Assumption 2 is identical to the standard normalization condition considered in e.g. Agarwal et al. 2020, Jin et al. 2020, with $C = d$. When $\mathcal{Z}$ is continuous, from a learning theoretical point of view, we still require some conditions to guarantee $w^\pi$ is learnable.  As we have discussed in our manuscript, if Assumption 2 holds, we have $w^\pi \in \mathcal{H}_k$, which guarantees that we can represent $Q$ accordingly. This is a common assumption that is widely used in the community. How to extend Assumption 2 to more general cases is left as an open problem.
>
> * **On the novelty of theoretical analysis**: As we have discussed in Remark 3 & 4 in Section 4, although our proof strategy is similar to RepUCB, there are several substantial difficulties when dealing with the *infinite dimension* representations. Specifically, for Lemma 17 and Lemma 19, we cannot directly bound the term of interest with the dimension of the representation like the previous work did (as the dimension can be infinite), and we need to deal with this issue with several additional tools in the functional analysis, for which we can be also of independent interest.

---

### Decision · Program_Chairs · 2023-01-20

**Decision:**

Accept: poster

**Justification For Why Not Higher Score:**

The paper proposes an interesting approach with interesting theoretical analyses but modest empirical results.

**Justification For Why Not Lower Score:**

The paper is borderline, but the proposed approach is quite novel and could be of interest for future research.

**Metareview: Summary, Strengths And Weaknesses:**

The paper provides a representation view of the latent variable models for state-action value functions, which is efficient and easily incorporates optimistic exploration bonuses. The authors provide both theoretical and empirical analyses of their work.
After reading each others' reviews and the authors' feedback, the reviewers discussed their concerns: the results are overclaimed and not very impressive. On the other hand, during the discussion, it emerged that the strong points (novelty, connections with other approaches, nice theoretical results, clear presentation) outweigh the weak points, and the paper deserves to be published.
The authors must carefully consider the reviewers' suggestions in preparing the final version of their paper.


**Note From Pc:**

if the above contains the word "oral" or "spotlight" please see: "oral" presentation means -> notable-top-5% and "spotlight" means -> notable-top-25%. As stated in our emails, we are disassociating presentation type from AC recommendations

**Summary Of Ac-Reviewer Meeting:**

All the reviewers participated in the virtual meeting.
The discussion was mainly focused on the concerns of the most critical reviewer, who felt the authors overclaim some results and the paper is too dense to properly verify all the contributions.
At the end of the discussion, the reviewers agree that, despite some drawbacks, the paper matches the bar and deserves publication.